# Comparisons of DOM and its Optical Characteristics in Small Low and High Arctic catchments

Caroline Coch[1,2], Bennet Juhls[3], Scott F. Lamoureux[4], Melissa J. Lafrenière[4], Michael Fritz[1], Birgit Heim[1], and Hugues Lantuit[1,2]

[1] Alfred-Wegener-Institute Helmholtz Centre for Polar and Marine Research, Telegrafenberg A45, 14473 Potsdam, Germany
[2] University of Potsdam, Institute of Earth and Environmental Science, Karl-Liebknecht-Straße 24/25, 14476 Potsdam, Germany
[3] Freie Universität Berlin, Institute for Space Sciences, Department of Earth Sciences, 12165 Berlin, Germany.
[4] Queen's University, Department of Geography and Planning, Mackintosh-Corry Hall, Kingston, Ontario K7L 3N6, Canada

*Correspondence to*: Caroline Coch (coch.caroline@gmail.com)

**Abstract.** Climate change is affecting the rate of carbon cycling, particularly in the Arctic. Permafrost degradation through deeper thaw and physical disturbances result in the release of carbon dioxide and methane to the atmosphere and to an increase in lateral dissolved organic matter (DOM) fluxes. Whereas riverine DOM fluxes of the large Arctic rivers are well assessed, knowledge is limited with regard to small catchments that cover more than 40 % of the Arctic drainage basin. Here, we use absorption measurements to characterize changes in DOM quantity and quality in a Low Arctic (Herschel Island, Yukon, Canada) and a High Arctic (Cape Bounty, Melville Island, Nunavut, Canada) setting with regard to geographical differences, impacts of permafrost degradation and rainfall events. We find that DOM quantity and quality is controlled by differences in vegetation cover and soil organic carbon content (SOCC). The Low Arctic site has higher SOCC and greater abundance of plant material resulting in higher chromophoric dissolved organic matter (cDOM) and dissolved organic carbon (DOC) than in the High Arctic. DOC concentration and cDOM in surface waters at both sites show strong linear relationships similar to the one for the great Arctic rivers. We used the optical characteristics of DOM such as cDOM absorption, Specific UltraViolet Absorbance (SUVA), UltraViolet (UV) spectral slopes (S275-295) and slope ratio (SR) for assessing quality changes downstream, at baseflow and stormflow conditions and in relation to permafrost disturbance. DOM in streams at both sites demonstrated optical signatures indicative of photodegradation downstream processes, even over short distances of 2000 m. Flow pathways and the connected hydrological residence time control DOM quality. Deeper flow pathways allow the export of permafrost-derived DOM (i.e. from deeper in the active layer), whereas shallow pathways with shorter residence times lead to the export of fresh surface and near-surface derived DOM. Compared to the large Arctic rivers, DOM quality exported from the small catchments studied here is much fresher and therefore prone to degradation. Assessing optical properties of DOM and linking them to catchment properties will be a useful tool for understanding changing DOM fluxes and quality at a pan-Arctic scale.

**1 Introduction**

Climate change has important impacts on carbon cycling, particularly in the Arctic. Approximately 1300 Gt of organic carbon are stored in permafrost soils in the northern hemisphere (Hugelius et al., 2014), which is 40 % more than currently circulating in the atmosphere. Thawing permafrost and deepening of the active layer leads to the mobilization of this carbon (Osterkamp,

2007; Woo et al., 2008), the release of carbon dioxide ($CO_2$) and methane ($CH_4$) to the atmosphere (Schaefer et al., 2014), and an increase in riverine dissolved organic carbon (DOC) fluxes (Frey and Smith, 2005; Le Fouest et al., 2018). Also associated with warming is the development of surface (physical) disturbances such as active layer detachments or retrogressive thaw slumps (Lacelle et al., 2010; Lamoureux and Lafrenière, 2009; Lewkowicz, 2007; Ramage et al., 2018), and thermal perturbation of the subsurface (Lafrenière and Lamoureux, 2013). As these processes influence freshwater systems, they

ultimately have impacts on the biological production and the biogeochemistry of the Arctic Ocean. The six largest Arctic rivers (Mackenzie, Yukon, Ob, Yenisey, Lena, Kolyma) drain 53 % of the Arctic Ocean drainage basin (Holmes et al., 2012) and transport huge amounts of nutrients and dissolved organic matter (DOM) to the ocean. However, there are limited flux estimates and information on DOM quality available for the remaining 47 %, which are sourced by smaller watersheds. "Small" in this context refers to smaller than the large Arctic rivers, as the actual size distribution of these watersheds remains unknown.

Terrigenous DOM is an important source of DOC originating from allochthonous (terrestrial such as soil and plants) and autochthonous (in situ production) sources (Aiken, 2014), and is modified by biotic and abiotic processes during its lateral transport to the ocean (Tank et al., 2018; Vonk et al., 2015a; Vonk et al., 2015b). Yet little is known about the transformation of DOM along short distances in small catchments. The composition and the vulnerability of riverine DOM to transformation is influenced by several factors such as soil organic matter and vegetation, sorption processes in the mineral layers, and

biodegradation and photodegradation processes (Cory et al., 2014; Mann et al., 2012; Vonk et al., 2015b; Ward and Cory, 2015; Ward et al., 2017). Chromophoric or colored dissolved organic matter (cDOM) is a fraction of DOM, which absorbs light in the ultraviolet and visible wavelengths (Green and Blough, 1994). Optical characteristics of cDOM such as absorption coefficients and spectral slopes can serve as proxies for DOM molecular weight and aromaticity, which in turn can help to characterize the lability of DOM (Helms et al., 2008; Neff et al., 2006; Spencer et al., 2009; Striegl et al., 2005; Weishaar et

al., 2003). High SUVA values (UV specific absorbance at 254 nm) in combination with low S275-295 (spectral cDOM slope between 275 and 295 nm) values indicate "fresh" DOM, or systems of shorter residence time receiving a greater input of fresh DOM from the catchment area. In contrast, low SUVA and high S275-195 are considered indicators of limited inputs of fresh DOM, a higher relative contribution of autochthonous DOM, greater exposure to photobleaching and longer residence time (Anderson and Stedmon, 2007; Fichot and Benner, 2012; Fichot et al., 2013; Helms et al., 2008; Whitehead et al., 2000).

Previous studies have focused on characterizing the cDOM-DOC relationship for the large Arctic rivers and coastal shelf areas, which exhibits a strong seasonality (Spencer et al., 2008; Stedmon et al., 2011; Walker et al., 2013). Some studies have investigated cDOM-DOC relationships in smaller Arctic catchments: Dvornikov et al. (2018) examined cDOM characteristics in surface waters of the Yamal Peninsula and cDOM-DOC relationships were examined for Subarctic catchments (Balcarczyk

et al., 2009; Cory et al., 2015; Larouche et al., 2015; O'Donnell et al., 2014) and the High Arctic (Fouché et al., 2017; Wang et al., 2018). Optical parameters have also been used to assess the impact of permafrost disturbance on stream geochemistry in Alaska (Abbott et al., 2014; Larouche et al., 2015) and the Northwest Territories (NWT), Canada (Littlefair et al., 2017). As most studies focused on downstream reaches, knowledge on the spatial variability across catchments is limited. To our knowledge, no study has examined this relationship in a Low Arctic setting or attempted to resolve geographic differences between the Low and High Arctic.

Here, we investigate cDOM and DOC in surface waters in the Low Arctic (Herschel Island, Yukon, Canada) and the High Arctic (Cape Bounty, Melville Island, Nunavut, Canada). The aim of this study is to (1) compare the variability and relation of DOC concentration and cDOM in Low and High Arctic surface water environments, (2) to investigate changes in DOM composition along longitudinal stream profiles and with regard to permafrost disturbances, and (3) examine changes in DOM concentration and composition throughout the summer season with occasional rainfall events.

## 2 Study Area

This study was carried out in two Arctic locations, Herschel Island in the Low Arctic and at the Cape Bounty Arctic Watershed Observatory (CBAWO), Melville Island in the High Arctic (Fig. 1a). Herschel Island (Yukon, Canada) is located at 69°35' N and 139°05' W in the Beaufort Sea off the Yukon coast. The island is composed of unconsolidated and fine-grained marine and glaciogenic sediments as it was formed by the Laurentide Ice Sheet (Rampton, 1982). The island is situated in the zone of continuous permafrost with ground ice content between 30 and 60 % for the entire island. Physical permafrost degradation typically occurs in the form of retrogressive thaw slumps (Lantuit and Pollard, 2008) and active layer detachments (Coch et al., in review). Ramage et al. (2019) reported mean active layer depths of 52.2 ± 20.2 cm. Soil organic carbon content (SOCC) for valleys on the eastern side of Herschel Island was estimated to be 11.4 ± 3.7 kg m$^2$ at 0 - 30 cm depth and 26.4 ± 8.9 kg m$^2$ at 0 - 100 cm depth with a C:N ratio of 12.9 ± 2.2 in 0 - 100 cm depth (Ramage et al., 2019). The dominant vegetation type is lowland tundra (Myers-Smith et al., 2011; Smith et al., 1989) and can be classified into subzone E (CAVM, 2003), which corresponds to the Low Arctic. The mean annual air temperature and yearly precipitation between 1971 and 2000 at Komakuk Beach, the nearest long-term meteorological station ~40 km away from our study site, are -11 °C and 161.3 mm respectively. The mean July temperature is 7.8 °C and average precipitation is 27.3 mm (Environment and Climate Change Canada, 2018). Snowmelt is the largest hydrological event of the year occurring in May to early June. Summer baseflow after mid-June is controlled by rainfall events (Coch et al., 2018). The active layer freezes up by mid-November (Burn, 2012). The studied catchments unofficially named Ice Creek West (1.4 km$^2$) and Ice Creek East (1.6 km$^2$) are adjacent to each other and merge before draining into the Beaufort Sea (Fig. 1b, Table 1). Both sampled ponds in Ice Creek West are < 1 ha, and there are degrading ice-wedge polygons present in the headwaters of Ice Creek West (Coch et al., in review).

The CBAWO is situated on the south coast of Melville Island (Nunavut, Canada) at 74° 55' N and 109° 35' W. The geology is characterized by Devonian sandstone and siltstone bedrock overlain by Quaternary marine and glacial sediments (Hodgson et

al., 1984). The soils are categorized as cryosols with a thin organic horizon. The site is situated in the zone of continuous permafrost, and active layer depths typically range from 50 to 70 cm (Lafrenière et al., 2013). Permafrost degradation such as deep thaw and physical disturbances have altered hydrochemical fluxes of the rivers (Lamoureux and Lafrenière, 2017). The vegetation cover is patchy with polar semi-desert, mesic tundra and wet sedge meadows (Edwards and Treitz, 2018), and falls into subzones B and C (CAVM, 2003). Soil organic carbon is estimated to be 3.0 kg m$^2$ in 0 - 30 cm depth, and 10.2 kg m$^2$ in 0 - 100 cm depth (Hugelius et al., 2013), with a C:N ratio of 10.0 in 0 - 100 cm depth (ADAPT, 2014). The nearest long-term meteorological station is located ~ 300 km away, at Mould Bay (NWT). Between 1971 and 2000, the mean annual air temperature and precipitation were -17.5°C and 111 mm, respectively. The mean July temperature is 4.0 °C and precipitation is 13.5 mm. Snowmelt and nival runoff start in early to mid-June with baseflow establishing around early to mid-July. Refreezing of the active layer starts late August or early September (Lamoureux and Lafrenière, 2017; Lewis et al., 2012). Samples were taken downstream in Boundary River (152.5 km$^2$), its sub-catchment Robin Creek (14.8 km$^2$), and the neighboring watersheds West River (8.6 km$^2$) and East River (12.0 km$^2$) (all unofficial names). There is an active retrogressive thaw slump in the Robin Creek watershed, and a number of recent (since 2007) active layer detachments and other disturbances in the other watersheds. The sampled lakes and ponds cover a range of sizes from below 1 ha in West River, to the larger downstream West and East lakes (~140-160 ha).

## 3 Methods

### 3.1 Field methods and hydrochemistry

To explore downstream changes in DOM across regions, we used a transect approach in this study. Samples were taken along longitudinal stream profiles in catchments and additionally samples were collected from standing water bodies (ponds and lakes) (Fig. 1). We also obtained discharge records and water samples from the outflow of both catchments at Herschel Island over the course of the 2016 season as detailed below.

Field work on Herschel was carried out in July-August 2016. We measured discharge using a cutthroat flume equipped with a U20 Onset Hobo level logger in Ice Creek West. Discharge data at 30 minute intervals is available from 15 May 2016 and at 5 minute intervals after 22 July 2016 (Coch et al., 2018; Coch et al., in review). In Ice Creek East, discharge was determined using the area velocity method in combination with a U20 Onset Hobo level logger (see Coch et al. in review for a detailed description). Data in Ice Creek East is available at 5-minute intervals after 25 July 2016. Weather data is available from the local Environment and Climate Change Canada station, and from an additional station deployed in Ice Creek West during the summer. Water samples were collected in bottles triple rinsed with sample at the outflow of both streams between 20 July and 10 August. At the outflow of Ice Creek West, water samples were collected using an automatic water sampler (ISCO 3700) at a 12-hour interval between 25 July and 10 August and more frequently during rainfall events (between 1-3 hours). Prior to the automatic sampling, and also in Ice Creek East, water samples were taken manually once per day. We collected water samples along longitudinal profiles of the channels (11 in Ice Creek West, 12 in Ice Creek East) starting in the headwaters (~ 2000 m

distance from the outflow) and following the channel downstream (Fig. 1). Longitudinal profiles were sampled 3 times in Ice Creek West (20, 25 and 30 July) and once in Ice Creek East (30 July). Samples of flowing water are available from Ice Creek West (n=90), Ice Creek East (n=32) and the alluvial fan (n=8). Standing water samples (n=4) were collected from 2 ponds in the Ice Creek West catchment.

The field work at CBAWO took place during August 2017. All samples were collected manually after triple rinsing the bottle with sample water. Similar to the Herschel field work, we collected samples along longitudinal stream profiles. Robin Creek is a subcatchment of Boundary River (Fig. 1), where stream samples were collected at six locations downstream of a retrogressive thaw slump. Three lakes were also sampled in the Boundary river catchment, and 2 samples from the main channel of the Boundary River. A total of 21 stream samples and 9 samples from lakes and ponds are available from the West

River catchment, some of which were collected after the rainfall event on 12 August 2017. In East River, 4 samples are available from the stream and 8 samples from standing water bodies.

Within 24 hours of sampling, electrical conductivity (EC) and pH were measured in the field laboratory. After collection, water samples were filtered through pre-rinsed 0.7 µm GF/F syringe filters and were then stored cool and dark for transport to the Alfred Wegener Institute, University of Hamburg and Geoscience Research Centre GFZ, Germany, where analysis for DOC

and cDOM were carried out. Water samples for absorbance measurements were kept in brown glass bottles without acidifying. Samples for DOC analyses were acidified with HCl (30 % suprapur) prior to the measurements. In 2016, DOC measurements were performed on a Shimadzu TOC-L analyzer with a TNM-L module (University of Hamburg), whereas a Shimadzu TOC-VCPH analyzer was used in 2017 (AWI). The error for these measurements is below 10 %.

Inorganic carbon was sparged out using synthetic air prior to the measurement. As we had a shortage of HCl in the field in

2016, 82 of the samples were frozen and acidified upon return to Germany. After new acid was acquired later in the summer, sample duplicates (n=47) were processed directly in the field and also frozen. The frozen duplicate was thawed and acidified upon return to determine the effect of different sample treatment (Coch et al. 2018). There is a significant linear relationship ($p < 0.05$, $n = 47$, $R^2 = 0.87$) between DOC concentrations of unfrozen and frozen sample duplicates. Samples that were frozen in the field, and subsequently thawed and acidified upon return to Germany showed lower DOC concentrations (by 13%) than

samples that were acidified directly in the field and kept unfrozen. We corrected the frozen samples for this offset (Supplementary S1). In both years, deionized water used in the field was also analyzed as blank following the same procedure. The absorbance was measured for the wavelengths from 200 to 800 nm with 1 nm increment using a LAMBDA 950 UV/VIS Spectrophotometer (GFZ Potsdam). The measurements were made in duplicates using a 5 cm cuvette and Milli-Q water as a blank. Some of the water samples showed fine particles precipitated in the sample bottle. They appeared in the form of small

thin flakes, which partly remained in suspension or accumulated at the bottom of the flask. This precipitation occurred after the samples were filtered through 0.7 µm glass fiber filters, transported to the laboratory for storage of about 4 weeks. This information was documented, and although absorbance spectra were measured, they were not further analyzed in this study as interference of the spectral characteristics by the particles might have occurred. This was the case for 25 (out of 55) samples at Cape Bounty and for 8 samples (out of 134) at Herschel Island.

The Naperian spectral absorption coefficient of cDOM ($a_{cDOM}(\lambda)$) was calculated out of the mean of each duplicate with

$$a_{CDOM}(\lambda)(m^{-1}) = 2.303\frac{(Asample(\lambda) - Areference_{(\lambda)}}{L}, \tag{1}$$

where $A_{sample}$ is the absorbance of the sample, $A_{reference}$ the absorbance of the Milli-Q reference and L the optical path length of the used cuvette in the spectrophotometer (L=0.05 m). The decadal absorption is multiplied by 2.303 that is the conversion factor of base 10 to base e logarithm to derive the Naperian absorption coefficient that is used for cDOM.

$$cDOM(\lambda)(m^{-1})\frac{2.303*(Asample(\lambda) - Areference_{(\lambda)}}{L}, \tag{2}$$

The absorption was corrected for scatter using a baseline correction by subtracting $a_{cDOM}(700)$ (Hancke et al., 2014; Helms et al., 2008). At that wavelength, absorption by cDOM is assumed negligible (Mitchell et al, 2002). Spectral slope values of $a_{cDOM}$ for wavelength ranges from 275 to 295 nm (S275-295) and 350 to 400 nm (S350-400) were calculated using Eq. (2) and a non-linear fit. These spectral slope values indicate photochemical or microbial alteration of DOM (Helms 2008). The ratio of both spectral slopes (S275-295: S350-400) defines the slope ratio (SR). The SUVA (mg L$^{-1}$ m$^{-1}$) was calculated by dividing the decadal absorption ($A_{254}$ / L) at 254 nm (m$^{-1}$) where $A_{254}$ is the absorbance at 254 nm and L the optical path length of the used cuvette in the spectrophotometer by DOC (mg l$^{-1}$). Both parameters, SR and SUVA, have been related to the relative molecular weight and aromaticity of DOM (Helms et al., 2008; Weishaar et al., 2003).

A cDOM absorption spectrum, aCDOM($\lambda$), is generally expressed as an exponential function

$$a_{CDOM}(\lambda)(\lambda) = a_{CDOM}(\lambda\ ) * e^{-S(\lambda-\lambda_0)}, \tag{2}$$

where $\lambda_0$ is the absorption coefficient at reference wavelength and $S$ is the spectral slope of $a_{CDOM}(\lambda)$ for the chosen wavelength range. To compare our data with different studies we converted absorption coefficient values reported in various studies to $a_{cDOM}350$ using an interpolation method developed by Massicotte et al. (2017). Throughout the manuscript all data is reported as mean ± standard deviation.

## 3.2 Statistical Analyses

We used RStudio (Version 1.0.153) to perform statistical tests (RStudio Team, 2016). Normality was tested using the Shapiro-Wilk normality test. To determine the difference in means of two populations, we applied the Welch's two sample t-test if the data was normally distributed with unequal variances. In the case where data was not normally distributed, we used the Wilcoxon-Mann-Whitney test. To measure the relationship between two variables, we used the Pearson correlation coefficient for normally distributed data and the Spearman rank correlation if the data was not normally distributed.

## 4 Results

### 4.1 DOM characteristics and relationships in and across Low Arctic (Herschel Island) and High Arctic (Cape Bounty) catchments

Comparing DOC concentrations and cDOM absorption between both study sites, significant differences can be found. The cDOM absorption is significantly higher ($p < 0.05$) in samples from Herschel Island compared with Cape Bounty across the entire spectrum (Figure 2a) with $a_{cDOM}350$ of $14.5 \pm 5.1$ m$^{-1}$ and $5.5 \pm 4.9$ m$^{-1}$, respectively (Fig. 2b). DOC concentrations show a similar pattern with significantly higher values ($p < 0.05$) on Herschel Island ($10.0 \pm 1.6$ mg l$^{-1}$) compared to Cape Bounty ($2.5 \pm 2.0$ mg l$^{-1}$) (Figure 2c, Table 2).

Comparing the streams on Herschel Island (Fig. 3b), the highest DOC and $a_{cDOM}350$ values are found in the headwaters of Ice Creek West. Ice Creek West has significantly higher ($p < 0.05$) values in DOC ($10.4 \pm 1.5$ mg l$^{-1}$) and $a_{cDOM}350$ ($16.1 \pm 5.4$ m$^{-1}$) than Ice Creek East, which are $8.7 \pm 1.1$ mg l$^{-1}$ and $11.1 \pm 1.8$ m$^{-1}$, respectively. At Cape Bounty, West River shows highest DOC ($2.5 \pm 1.7$ mg l$^{-1}$) and $a_{cDOM}350$ ($8.5 \pm 5.2$ m$^{-1}$) compared to other sampled streams. The highest values of DOC and $a_{cDOM}350$ values of flowing are are recorded in West River after the August 8 rainfall event.

For both study areas, DOC concentrations are substantially higher in standing waters compared to flowing water (Fig. 2c). Furthermore, on Herschel Island, generally upstream waters show higher DOC concentrations and $a_{cDOM}350$ compared to downstream waters. At Cape Bounty, samples from standing waters in the East River catchment show the highest DOC concentrations and $a_{cDOM}350$ values.

Mean S275-295 on Herschel Island is generally higher ($16.4 \pm 1.5 \times 10^{-1}$ nm$^{-1}$) compared to Cape Bounty ($14.8 \pm 3.2 \times 10^{-1}$ nm$^{-1}$), whereas SUVA values show a broader range at Cape Bounty (from 1.35 to 5.16 mg L$^{-1}$ m$^{-1}$) compared to Herschel Island (from 2.0 to 4.3 mg L$^{-1}$ m$^{-1}$) (Fig. 2d, Table 2). At Cape Bounty, standing water samples show significantly larger spectral slopes and slope ratios ($p < 0.05$) and mostly smaller SUVA ($p < 0.05$) than flowing water samples (Fig. 2d-f).We observed a significant positive relationship (rho = 0.78, $p < 0.05$) between $a_{cDOM}350$ and DOC concentration for all samples at both sites (Fig. 3a). Whereas on Herschel Island, the relationship between $a_{cDOM}350$ and DOC follows one linear trend (rho = 0.72, $p < 0.05$) the relationship at Cape Bounty is broadly separated into two groups, which we schematically indicate in by solid ellipsoids that correspond to flowing and standing water. Correlations for both groups are significant (< 0.05) and show different spectral slopes. One sample identified as standing water falls into the group of flowing water. We identify this sample as an outlier likely affected by a local DOM source.

We found a strong negative relationship between SUVA and S275-295 among all water samples from both locations (rho = -0.64, $p < 0.05$, Fig. 4a). This relationship is even stronger when only samples from Herschel are considered (rho = -0.72, $p < 0.05$), whereas samples from Cape Bounty show higher deviations. Reported SUVA and S275-295 from Walker et al. (2013) indicate a similar relationship among the large Arctic Rivers (Ob, Lena, Yenisei, Kolyma and Mackenzie). SUVA and S275 of the large Arctic Rivers are located in the center of our reported values from Herschel Island and Cape Bounty (highlighted in Fig. 4a).

Whereas no difference was found between standing and flowing water on Herschel Island, pH and EC values were significantly higher in standing water at Cape Bounty than in flowing water (Table 2). Robin Creek showed highest EC values and the largest variability ($145 \pm 213$ µS cm$^{-1}$) of the Cape Bounty rivers, whereas West River showed the overall lowest EC values ($60 \pm 17$ µS cm$^{-1}$). On Herschel Island, both adjacent streams show EC and pH values in the same order of magnitude with a slight decrease at the alluvial fan outflows.

## 4.2 Hydrochemical and DOM patterns along longitudinal stream transects

The studied streams followed different hydrochemical patterns from upstream to downstream (Fig. 5). EC and pH are significantly higher ($p < 0.05$) in surface waters on Herschel Island ($1050 \pm 370$ µS cm$^{-1}$ and $8.2 \pm 0.2$ µS cm$^{-1}$) than at Cape Bounty ($137 \pm 136$ µS cm$^{-1}$ and $7.2 \pm 0.5$ µS cm$^{-1}$). On Herschel Island EC increased from upstream to downstream in Ice Creek West, whereas it varied less in Ice Creek East (Fig. 5e). At Cape Bounty, river samples exhibit no clear visible trends, except for Robin Creek, where an active retrogressive thaw slump is hydrologically connected to the stream. Here, we observed a substantial downstream decrease in EC.

At Herschel Island, overall, DOC concentration and $a_{cDOM}350$ (Fig. 5a) decreased downstream at all sampling periods. However, we observed a stronger decrease in Ice Creek West compared to Ice Creek East. Ice Creek West shows an increase in DOC and $a_{cDOM}350$ concentration at ~1300 m, where a tributary joins the main stem. On 30 July 2016, when both streams on Herschel Island were sampled simultaneously, Ice Creek East showed significantly lower ($p < 0.05$) DOC concentrations and $a_{cDOM}350$ than Ice Creek West throughout the entire profile. At Cape Bounty, DOC concentrations and $a_{cDOM}350$ do not show clear downstream trends but are at a rather low $< 2$ mg l$^{-1}$ in all streams. One exception is West River after the rainfall event on 12 August 2017, where we found a slight downstream increase of DOC and $a_{cDOM}350$ with generally higher levels of DOC and $a_{cDOM}350$ compared to the period before rainfall. In Robin Creek, we observed an increase in DOC from 1.3 mg l$^{-1}$ to 1.7 mg l$^{-1}$ as the stream gets impacted by a retrogressive thaw slump, and then DOC decreases thereafter. Boundary River shows similar concentrations to Robin Creek. Generally, for Cape Bounty rivers other than West River the number of samples is likely too low to allow clear statements about downstream trends.

SUVA values on Herschel Island showed a similar, however, weaker decreasing downstream trend as DOC and $a_{cDOM}350$ (Fig. 5c). Different to DOC and $a_{cDOM}350$, SUVA values of both Herschel Island streams are very similar. In contrast, at Cape Bounty, West River (sampled after rainfall) showed higher SUVA than the remaining rivers (sampled before the rainfall).

Spectral slope values (S275-295) at Herschel Island showed an increase downstream (Fig. 5d). When sampled on the same day, Ice Creek West showed only slightly smaller spectral slope values along the stream profile compared to Ice Creek East on 30 July 2016. Significant differences were observed between different sampling periods in the Ice Creek West. Spectral slope values were smallest after the first rainfall event (19 July) and increase progressively over the course of the season. The Cape Bounty rivers showed highest spectral slopes for East River ($16.1 \pm 1.6$ x $10^{-3}$ nm$^{-1}$) and the lowest for West River ($11.9 \pm 0.8$ x $10^{-3}$ nm$^{-1}$. A slight downstream increase in spectral slope was recorded in West River.

## 4.3. Temporal changes of DOM under different meteorological conditions

The mean annual air temperature on Herschel Island was -6.3 °C in 2016 with mean temperatures of 9.4 °C in July and 7.7 °C in August. During the monitoring period, rainfall events of 33.9 mm (19 July), 9.3 mm (30 July) and 12.7 mm (5 August) were recorded. Cape Bounty had a mean annual air temperature of -15.3 °C in 2017, with mean air temperatures of 4.5 °C in July and 1.6 °C in August. During the monitoring period, two rainfall events of 0.2 mm (4 August) and 1.2 mm (8 August) occurred. Changes in discharge, DOM composition, and conductivity over the summer season were observed for both streams at Herschel Island. Rainfall response is direct with steep rising hydrographs and elongated falling limbs (Fig. 6a) in both streams (detailed presentation of rainfall response in Coch et al. 2018). In both streams, DOC, $a_{cDOM}350$, and SUVA were highest following the 33.9 mm rainfall event (Event-1). Following the rainfall event, declining DOC accompanied by a decline in $a_{cDOM}350$, SUVA, and an increase in S275-295 (Fig. 6b-e). EC is steadily increasing after peak flow in both streams (Fig. 6f).

The subsequent rainfall event (Event-2, 9.3 mm) led to an increase of DOC, $a_{cDOM}350$ and S275-295, and a decrease in SUVA (Fig. 6b-e) in Ice Creek West on 30 July. This dynamic was only captured to some extent in Ice Creek East, which was sampled at a longer time interval. Baseflow increased after this rainfall event (Fig. 6a). EC in both streams is dropping with peak flow and increasing thereafter.

The hydrochemical response to rainfall Event-3 (12.7 mm) was different from the response to Event-2. An initial decrease in DOC, $a_{cDOM}350$, and S275-295 is followed by a sharp increase of these parameters in Ice Creek West. SUVA shows two peaks on 3 August followed by a general decreasing trend until the end of our sampling period 10 August). Ice Creek East had a different response showing an increase in DOC and $a_{cDOM}350$ and a distinct decrease in SUVA.

## 5 Discussion

### 5.1. Catchment processes and DOM alteration

#### 5.1.1 Regional catchment properties of DOM

Our study sites show strong differences in DOM quantity and quality related to their geographic location and environmental setting. DOM characteristics, such as SUVA and S275-295, provide insights into potential sources, degradation state and properties of the water bodies. Herschel Island (Low Arctic) shows on average significantly higher values in DOC, $a_{cDOM}350$, and SUVA than Cape Bounty (High Arctic). Catchment topography (Connolly et al., 2018), vegetation type and soil characteristics (Harms et al., 2016) are important drivers of DOC concentrations in catchments. The greater abundance of vegetation in the Low Arctic (Fig. 1) delivers more organic material resulting in high amounts of lignin introduced into the aquatic system (Sulzberger and Durisch-Kaiser, 2009) compared to the High Arctic. The Herschel Island tundra catchments have thick organic moss mats and a dense layer of vascular plant cover which delivers plant detritus that is continuously decomposed resulting in DOC and $a_{cDOM}350$ values higher than in the high Arctic catchment at Cape Bounty.

In addition to higher DOM concentrations, we observed that the sampled surface waters in the Herschel Island catchments contain DOM with specific optical characteristics for high aromaticity and high molecular weight (Guéguen et al., 2007; Guo et al., 2007) that are indicative of fresh organic matter (Neff et al., 2006; Stedmon et al., 2011). High value ranges and variability of SUVA and S275-295 in the surface waters of the Cape Bounty catchments point towards a broad spectra of different DOM sources and quality. The sampled surface waters throughout the 2017 summer season contain DOM with high spectral slope values (S275-295) and low SUVA as well as surface waters with low S275-295 and high SUVA. This is due to different flow pathways, residence time and permafrost disturbance delivering DOM of different quality.

Cape Bounty, two different water types were identified based on the $a_{cDOM}350$ to DOC ratios (Fig. 3c). The group of surface waters with lower $a_{cDOM}350$ to DOC ratios is dominated by standing water bodies. High residence times in standing waters make photodegradation of DOM a dominant process (Vonk et al., 2015b) and result in an increase of S275-295, and decrease of the cDOM-DOC ratio. The group of surface waters with higher $a_{cDOM}350$ to DOC ratios, higher SUVA and lower S275-295 is, in contrast, dominated by flowing water. Higher turbidity in flowing waters potentially limits photodegradation processes (Cory et al., 2015; Cory et al., 2014) preserving low S275-295. Within the catchments, there may be also more import of fresh organic material to the flowing water bodies.

### 5.1.2 Downstream patterns of DOM and impact of permafrost disturbance

Transport and degradation of DOM is a dynamic process. Vonk et al. (2015b) showed that the microbial and photo-degradability decreased from small streams towards larger rivers within the continuous permafrost zone. The fate of DOM along lateral flow pathways from headwater streams through lakes and large rivers to the ocean is altered by photochemical and biological oxidation (Cory et al., 2015; Cory et al., 2014). Studies show the importance of headwater systems where photodegradation (Cory et al., 2014) and bacterial respiration of ancient permafrost-derived DOC are prevalent (Mann et al., 2015). Our sampling strategy along the rivers in combination with detailed mapping of the catchments with a focus on permafrost disturbances, provide insights into upstream to downstream patterns in small coastal catchments in both the Low and High Arctic.

At Herschel Island, we found a high variability of DOC, SUVA, and S275-295 in the headwaters of Ice Creek West. The locations at 2000 m and 1300 m distance from the outflow show distinct high values of DOC and $a_{cDOM}350$ compared to the other locations downstream of them. These high concentrations are a result of degrading ice-wedge polygons, which heavily influence DOM in the headwaters of the stream (Coch et al. in review). The location at 1300 m marks the inflow of another headwater tributary impacted by degrading ice-wedge polygons. Thus, the main expected sources for fresh mobilized DOM, from deeper permafrost soil horizons, are the headwaters and tributary water. This is supported by high SUVA and low S275-295 indicating high molecular weight.

DOC and $a_{cDOM}350$ are highest in the headwaters and decrease downstream. Combined with increasing S275-295 along both streams, our results are indicative of a progressive photochemical degradation of DOM. S275-295 has been found a good indicator for photodegradation of DOM (Fichot and Benner, 2012; Fichot et al., 2013; Helms et al., 2008), and also been

observed along a flow-path continuum of the Kolyma River (Frey et al., 2016). They found a relatively constant proportion of bioavailable DOM along the entire flow path, indicating an acclimatization of aquatic microorganisms to downstream DOM changes and/or the generation of labile DOM for microbial processing through photodegradation. Cory et al. (2014, 2015) show at a Subarctic site that DOC in headwater streams, which are directly sourced by soil water, have low prior exposure to
light and is therefore prone to photodegradation to $CO_2$.

At Cape Bounty, optical data of downstream patterns is more limited (see section 5.3). West River shows an increase in DOC concentration downstream (3 August 2017), which is also reflected in an increase of $a_{cDOM}350$. As discussed by Fouché et al. (2017) and Wang et al. (2018), the West River is characterized by a downstream increase in autochthonous DOM. SUVA and S275-295 do not show strong differences downstream in the West River suggesting little modification of DOM through
microbial and/or photodegradation processes. A retrogressive thaw slump at Robin Creek heavily impacted DOM quality. At ~2100 m distance from the outflow, closest to the slump, we see the highest DOC, S275-295 and EC values and lowest SUVA, indicative of low aromaticity and lower molecular weight.

Abbott et al. (2014) found that DOM is most biodegradable during active disturbance at sites in the Subarctic. SUVA values at thermokarst outflows in that study were half in the magnitude of the undisturbed reference waters, indicating less aromatic
DOC. High S275-295 and SR were observed in conjunction with geomorphic disturbance in headwater streams of West River by Fouché et al. (2017). Impact of retrogressive thaw slumps on DOM quality was also studied in the Subarctic Peel Plateau by Littlefair et al. (2017). They reported similar dynamics at modestly sized slumps as we observed at Robin Creek: DOC concentration is highest directly at the slump outflow and is lower downstream compared to undisturbed upstream conditions. The authors attribute low SUVA and high S275-295 within the disturbed site to deep permafrost flow pathways. SUVA values
of surface waters in Lake Bounty catchments within the slump and downstream are very similar to the ones reported by Littlefair et al. (2017). Overall, DOM characteristics in both study areas are affected by local permafrost disturbances. In sampling transects which are not affected by permafrost disturbances, gradual degradation was observed.

### 5.1.3 Rainfall event impacts on DOM

Rain magnitude, intensity, and antecedent conditions play an important role for mobilizing DOM in permafrost catchments.
At Herschel Island, we captured the response to three different rainfall events through continuous sampling at the outflow and repeated sampling along the longitudinal stream profile in Ice Creek West.

Rainfall Event-1 (33.9 mm) was captured only at the receding hydrograph at the outflow (Fig. 6), but along the stream profile in Ice Creek West (Fig. 5). This event of high magnitude and intensity led to high SUVA and low S275-295 values indicating "fresh" plant derived DOM that is prone to degradation processes, both, microbial and photodegradation. After this event, the
hydrograph recedes, and the DOM signature during the "post rain" conditions suggests a source from deeper in the active layer that contains potentially older carbon (decreased SUVA and increased S275-295 at the outflow and throughout the profile) than surface soils with mostly recently fixed carbon from the vegetation cover. The contrasting response of Ice Creek West to rainfall events 2 and 3, suggests different sourcing of DOM and controlling factors.

During the second rainfall event (9.3 mm), as DOC concentration increased, we found a decrease in SUVA accompanied by an increase in S275-295. This indicates a decrease in aromaticity and a lower molecular weight, indicative of more decomposed material. The following event of 12.7 mm led to a decrease in DOC concentration and S275-295 and an increase in SUVA indicating an increase in aromaticity and a higher molecular weight – suggesting fresher and lignin-rich plant derived DOM.

A change in water sources for these two rainfall events was examined by Coch et al. (in review). Whereas runoff during the 9.3 mm rainfall event showed the signature of supra-permafrost water, which was forced out during that rainfall event, runoff during the subsequent 12.7 mm rainfall event reflected the isotopic signature of rain (Coch et al. (in review). Thus, the DOM was first sourced from the surface and through the entire active layer and had a longer residence time than the rain event after. These results indicate that antecedent (pre-rainfall) soil water conditions play a crucial role for the sourcing of DOM. The

second rainfall event (9.3 mm) occurred about 10 days later, whereas the time difference between the second and the third one was less than 4 days (i.e. the soil was saturated mobilizing surface OM). In addition to the antecedent conditions, the magnitude and intensity of the rainfall event might also play an important role here. The 9.3 mm rainfall event occurred over a period of 3 days. Thus, the flow pathways during this event might be deeper in the active layer mobilizing more decomposed OM (Marín-Spiotta et al., 2014). In contrast, the subsequent 12.7 mm event occurred within 1 day, which presumably led to increased

overland flow and the mobilization of surface OM. Baseflow in this catchment is increasing with summer rainfall and as the summer season progresses (Coch et al., 2018). The authors also reported a linear increase of DOC export with increasing runoff. Our dataset shows that the quality of exported DOC depends on the intensity of rainfall and the antecedent conditions, which in turn determine hydrological flow pathways and sourcing of DOM.

When sampling before and after rainfall on Cape Bounty (West River), we found a substantial increase in DOC concentration compared to the pre-rainfall concentrations. Fouché et al. (2017) conducted an extensive study of DOM quality in four

headwater streams of West River (Cape Bounty) and also reported an increase in DOC concentrations and fluxes during stormflow. They observed a change in DOM quality: enrichment in fresh low molecular weight (LMW), microbially-derived, components as indicated by an increase in S275-295 and a decrease in SUVA during rainfall. Although we do not have data on the optical properties for West River before the rainfall event, similar concentrations of DOC in the West and East rivers point towards similar optical characteristics at that time. Baseflow in undisturbed High Arctic headwater streams seems

therefore characterized by more high molecular weight (HMW) humic-like components with high aromaticity (low spectral slope and increase in SUVA) relative to stormflow DOM. In turn, stormflow leads to an export of DOM characterized by lower molecular weight and decreased aromaticity (high spectral slope, decreased SUVA). Fouché et al. (2017) explain this pattern by a change in flow pathways from shallow active layer soils (baseflow) to subsurface runoff (rainfall), where soluble

components from mineral soils deeper in the active layer are mobilized. Associated with the change in DOM quality, they also found an increase in total dissolved solids (TDS) supporting this hypothesis. Impacts of changing flow pathways on DOM quality are also reported from a subarctic setting by Balcarczyk et al. (2009). The increased residence time of percolating water through the active layer leads to a selective sorption of compounds to mineral soil particles. The authors describe that hydrophobic compounds are absorbed, while hydrophilic compounds remain in the solution, and are therefore exported from

the catchment (Balcarczyk et al., 2009). Further, an increased residence time and subsurface flow mobilizes DOC that is more microbially degraded (Striegl et al., 2005; Ward and Cory, 2015).

Several studies anticipate a shift towards deeper flow pathways as active layer depths increase with climate change (Drake et al., 2018; Liljedahl et al., 2016; Mann et al., 2015; O'Donnell et al., 2014; Ward and Cory, 2015). These studies found that permafrost-derived DOM is more labile and thus easily used by bacteria compared to surface (organic mat) DOM. As described above, we show that different flow pathways are activated during stormflow conditions at the Low and High Arctic locations, which influences the quality of DOM exported. At the Low Arctic setting our data suggests that more permafrost-derived DOM is exported with increasing baseflow during the season and during a rainfall event of smaller magnitude and lower intensity. Based on the optical properties, this material shows low molecular weight and aromaticity (i.e. it is already altered). In contrast, high magnitude and intensity rainfall events that act on saturated soil lead to shorter residence time in the flow path and thus export more fresh (less altered due to different degradation processes), near-surface-derived DOM (higher SUVA and lower S275-295). As summer rainfall is projected to increase across the Arctic (Bintanja, 2018; Bintanja and Andry, 2017) an increase in DOC export is expected (Coch et al. 2018). Small catchments in the subarctic Canadian Shield have already shifted towards a nival-pluvial flow regime leading to substantial increases in organic matter fluxes during fall and winter (Spence et al., 2011; Spence et al., 2015). The DOM quality will depend on the residence time and thus, flow pathways within the catchment, which in turn is controlled by the frequency and magnitude of the rainfall events and the thaw depth of the active layer.

## 5.2 DOM dynamics of small and large Arctic catchments

The knowledge of ecosystem responses to external disturbances is necessary for predictive models. Flux of DOM is a significant input into Arctic coastal oligotrophic marine environments. The major Arctic catchments cover approximately half of the Arctic drainage basin, whereas the remainder is covered by the complex network of smaller catchments. In the Arctic, most historical data and studies on riverine DOM dynamics are from the major Arctic catchments. Research on small catchments may yield different information to riverine DOM of the major Arctic catchments.

We linked DOC and $a_{cDOM}350$ from this study and the literature (Table 3; Supplementary Table S1, S2) to latitude and the soil organic carbon content (SOCC) in 0-30 cm and 0-100 cm depth as retrieved from Hugelius et al. (2013). We found a positive correlation (rho = 0.53/0.51, $p < 0.05$) between SOCC and DOC concentration, indicating that vegetation coverage and the connected SOCC are influencing DOM. The relationship between $a_{cDOM}350$ and SOCC is also significant, although weaker (rho = 0.26 / 0.34, $p < 0.05$). It is important to bear in mind that the northern circumpolar soil carbon database is a product of upscaling and will most likely not cover the spatial variability reported in the studies. Nevertheless, the data shows a decrease of SOCC at higher latitudes, influenced by climate and accordingly vegetation cover and related soil cover. DOM and SOCC are further influenced by watershed topography. Longer residence times in low relief terrain and high hydrologic connectivity facilitate leaching and export of DOM from soil organic matter (Connolly et al., 2018; Harms et al., 2016). However, DOM characteristics are not always influenced by the subcatchmentes of the investigated waterbodies. Several of the studies contain

data on waterbodies with small subcatchments but located within the large Arctic river floodplains with very large catchment sizes. For example, Dvornilov et al. (2018) report untypically high cDOM for tundra lakes in Yamal (Western Siberia) and Skorospekhova et al. (2016) for tundra lakes in the Lena Delta (Central Siberia), which are all influenced by the spring flood of the large rivers seasonally flashing high amounts of organic material into these lakes.

5 Strong positive correlations between DOC and $a_{cDOM}350$ found in this study (Fig. 3a) is also characteristic for riverine DOM of the large Arctic rivers (Walker et al., 2013), across the land-ocean continuum in the Eastern Arctic (Juhls et al., 2019; Mann et al., 2016) and globally (Massicotte et al., 2017). We used DOC and cDOM data available from surface waters in northeastern Canada (Breton et al., 2009), Scandinavia (Forsström et al., 2015; Kellerman et al., 2015) and Alaska (Cory et al., 2015; Larouche et al., 2015). Comparing our data from the low Arctic and High Arctic sites to those found in the literature confirms

10 the strong positive relationship (rho = 0.85, p < 0.05) between DOC and $a_{cDOM}350$ (Fig. S3), indicating the robustness for using the optical parameter $a_{cDOM}350$ as a proxy for DOC concentration in terrestrial freshwater systems. Compared to DOC concentrations and cDOM magnitudes of other small Arctic catchments, some of the samples from Cape Bounty show extremely low values, which reflects the low supply of organic matter due to the low plant abundance in the High Arctic. DOC and cDOM from Herschel Island are within the range of most studies on Arctic catchments whereas studies at High Arctic

15 sites with low vegetation cover are underrepresented. However, compared to other reported studies with DOC and cDOM in the same value range, $a_{cDOM}350$ is slightly depleted that is visible in a lower cDOM to DOC ratio. This can be a result of stronger photodegradation compared to other sites with eventually more turbid water types, but information on turbidity or suspended matter is frequently not provided in those studies focusing on DOM dynamics and properties.

Due to snow melt dynamics and active layer development throughout the summer season in Arctic catchments there is also a

20 strong seasonal influence on the DOC to cDOM relationship leading to variability throughout the season and regions (Mannino et al., 2008; Vantrepotte et al., 2015). Walker et al. (2013) report SUVA for three different flow regimes of the large Arctic rivers: peakflow (spring freshet), midflow (summer) and baseflow (winter). The SUVA values reported in this study ($2.9 \pm 0.4$ L mg$^{-1}$ m$^{-1}$ for Herschel Island and $2.8 \pm 1.1$ L mg$^{-1}$ m$^{-1}$ for Cape Bounty) are higher than the mean mid-flow SUVA for the five large Arctic rivers (2.4 L mg$^{-1}$ m$^{-1}$), which ranges between 2.0 L mg$^{-1}$ m$^{-1}$ in the Mackenzie River and 2.7 L mg$^{-1}$ m$^{-1}$ in

25 the Ob'. This supports the model proposed by Vonk et al. (2015b), that DOM exported from smaller rivers has a higher aromaticity, which suggests that the material is fresh and less altered by different degradation processes. However, our results also show that a broad range of SUVA as well as S275-295 can be found in small Arctic catchments. This highlights the importance of small rivers and streams for the magnitude of potential modification of DOM before waters are exported in the large rivers or directly to the Arctic Ocean. The ratio of SUVA versus S275-295 values from the large Arctic rivers (Fig. 4a)

30 falls in the same value range like the ratio values from the catchments of Cape Bounty and Herschel Island. However, SUVA values of the large Arctic rivers are in the low value range only reflecting the higher degradation status of the transported DOM. The results of this study suggest that small Arctic catchments potentially deliver material that is fresher and more prone to degradation compared to DOM of the large Arctic rivers.

## 5.3. Limitations of cDOM measurements from terrestrial sources

There are some constraints to optical DOM measurements and the nature of the samples themselves that we encountered in this study. As described in the methods section, some samples formed precipitates inside the bottles in the form of small thin flakes, which partly remained in suspension or accumulated at the bottom. All samples were filtered in the field through 0.7

μm glass fiber filters, and the precipitation occurred after filtration during storage. At Cape Bounty, these problematic samples had very high $a_{cDOM}350$ values of $13.9 \pm 13.8$ m$^{-1}$ with a maximum of 75.8 m$^{-1}$, and SUVA values of $10.1 \pm 11.5$ L mg$^{-1}$ m$^{-1}$ with a maximum of 59.5 L mg$^{-1}$ m$^{-1}$. Those values are significantly higher ($p < 0.05$) than the mean values reported in Table 2 and are not realistic for natural surface waters. At Herschel Island, $a_{cDOM}350$ and SUVA did not differ significantly from the mean ($11.8 \pm 0.8$ m$^{-1}$ and $3.5 \pm 0.4$ L mg$^{-1}$ m$^{-1}$ respectively).

As described in the methods section (3.1), samples showing precipitates in the laboratory were excluded from the study, even if the absorption values were plausible when compared to the corresponding DOC concentration (Fig 7). At Cape Bounty, this was the case for 25 out of 55 samples. We assume that absorbance measurements are high because of scattering by newly formed colloid complexes and precipitates and absorbance from other absorbing dissolved constituents. Also, Hansen et al. (2016) and Weishaar et al. (2003) report that SUVA values above 6.0 L mg$^{-1}$ m$^{-1}$ are indicative for an optical disturbance due

to other constituents in the sample (Hansen et al., 2016; Weishaar et al., 2003).

We suggest that the optical interference could be due to polymeric iron (hydr)oxides or high concentrations of dissolved iron and changing pH conditions of the sample. Dissolved iron in terrestrially dominated waters is dominantly complexed with humic and fulvic acids. With changing temperature and changing pH of the sample filtrates, redox reaction can result in colloid formation and phase changes, which then strongly affect the optical properties of the sample filtrate by scattering. Poulin et al

(2014) describe how iron (Fe(II,III)) is known to interfere with the absorption of cDOM with a linear dependency of increasing $a_{cDOM}$ with increasing Fe(III) concentration in the water. Poulin et al. (2014) suggest to correct cDOM absorption coefficients according to the iron concentrations using correction coefficients. Coch et al. (2018) report total aqueous dissolved iron concentrations from Herschel Island. High total iron concentration is found to occur in high $a_{cDOM}350$ (Fig. S2), which indicates a potential influence of iron concentration on the absorption. Fraction of Fe(II) and Fe(III) on the total iron concentration was

not measured as a standard hydrochemistry measurement, thus the correction could not be performed. However, Figure 7 clearly shows that the samples that were removed fell into the problematic group (circled), where cDOM was overestimated compared to DOC concentration. This conservative approach removed also other samples with reasonable cDOM to DOC ratios.

Poulin et al. (2014) also showed that in samples with low pH the dominant fraction of iron is Fe(II) which then potentially can

precipitate as Fe(III) with increasing pH during transport and storage. The Cape Bounty samples that showed a substantially lower pH, likely caused by low vegetation, are therefore more prone to precipitate Fe(III) colloids that affect the optical absorption measurements and lead to the high absorption values at 700 nm (Fig. S4). Herschel Island samples originally already had a higher pH compared to Cape Bounty. Thus, we expect that the dominant fraction of iron on Herschel Island was Fe(III)

that leads to a lower potential of Fe(III) precipitation compared to Cape Bounty. Catchment properties that influence riverine pH such as the local lithology may play an important role. In case of alpine and high Arctic catchments with thin or no soil cover, a bed rock composition of acid rocks in the catchments will lead to lower pH values in surface waters such as it is the case for Cape Bounty. Whereas surface waters from Herschel Island catchments on glacial moraines and marine sediments are

characterized by higher alkalinity.

## 6 Conclusion

This study investigates DOM optical properties in Low and High Arctic surface water environments and downstream patterns with regard to permafrost disturbance and rainfall events. We find that both Arctic locations exhibit a distinct signature of DOC concentration and $a_{cDOM}350$ linked to the differences in vegetation cover and SOCC content. Compared to the High

Arctic (Cape Bounty), DOC and cDOM in the Low Arctic (Herschel Island) is higher due to the greater abundance of plant material and higher SOCC. In both regions, the strong terrestrial signature of DOM is apparent in the optical properties, which is typical for small headwater catchments. The relationship between $a_{cDOM}350$ and DOC is very strong across both regions and including data from the literature, proving the applicability of cDOM as a tracer for DOC throughout different aquatic Arctic environments (rivers, streams and lakes).

Comparing DOM optical characteristics (SUVA, S275-295, and SR) from large Arctic rivers to the surface waters in our study, we find that the low and high Arctic small catchments potentially deliver fresh, less altered DOM. However, the results also show that DOM characteristics indicate organic matter modification and degradation cover a broad spectrum and can be highly variable in small catchments in space (along longitudinal transects) and time (throughout the season). Degrading ice-wedge polygons and retrogressive thaw slumps impact DOM quantity and quality in the catchments. This underlines the importance

of small catchments for potential DOM modification and degradation before waters are entering bigger rivers or coastal waters of the Arctic Ocean.

The optical characteristics of DOM prove to be useful for assessing downstream patterns in the studied streams. The downstream increase of S275-295 is indicative for photodegradation processes, which is apparent in most of the streams. Although the temporal resolution of data at Cape Bounty is limited, we found a similar response to rainfall events like in the

Herschel Island study. Rainfall leading to runoff with a short residence time (rainfall of high magnitude and intensity, dry antecedent conditions in the catchment) leads to the export of fresh near-surface-derived DOM (higher SUVA, lower S275-295). In contrast, baseflow conditions and long residence times (including low magnitude rainfall events and a saturated catchment) favors the export of permafrost-derived DOM that has undergone microbial processing in the soil. Examining flow pathways and residence time will be crucial to assess the impacts of projected increasing summer rainfall across the Arctic.

Monitoring optical properties of DOM in combination with a mapping of permafrost disturbances across river catchments, will be a useful tool for assessing DOM fluxes and DOM quality changes at a pan-Arctic scale.

**Data Availability**

Data has been made available through PANGAEA:

Coch, Caroline; Juhls, Bennet; Lamoureux, Scott; Lafrenière, Melissa; Fritz, Michael; Heim, Birgit; Lantuit, Hugues (2019): Colored dissolved organic matter (cDOM) absorption measurements in terrestrial waters on Herschel Island (Low Arctic) and Melville Island (High Arctic) in 2016 and 2017. PANGAEA, https://doi.pangaea.de/10.1594/PANGAEA.897289

**Author Contributions**

C.C., H.L., and S.L. developed the study design. Field work was conducted by C.C. in 2016, and by C.C., S.L., M.L. in 2017. M.F. partly funded and supervised laboratory analyses for cDOM measurements. B.J. processed the absorbance spectra and contributed to developing the manuscript. C.C. ran laboratory analyses and processed and interpreted the data with input from B.H., M.L., S.L., and H.L. and prepared the manuscript with editorial contributions from all co-authors.

**Competing interests**

The authors declare that they have no conflict of interest.

**Acknowledgements**

Thanks to the two anonymous reviewers, who helped to improve the manuscript. We are grateful to the Yukon Territorial Government, Yukon Parks (Herschel Island Qikiqtaryuk Territorial Park), and the Aurora Research Institute for their support during this project. This work was funded by the Helmholtz Association (grant no. VH-NG-801 to Hugues Lantuit), and it has received funding under the European Union's Horizon 2020 research and innovation programme under grant agreement No 773421. C. Coch received a scholarship and travel support from the Studienstiftung des deutschen Volkes. This work was also financially supported by Geo.X, the Research Network for Geosciences in Berlin and Potsdam (Grant/Project-number: SO_087_GeoX). Research at CBAWO is supported by the Canadian Natural Sciences and Engineering Research Council (NSERC) and ArcticNet National Centres of Excellence. Polar Continental Shelf Program provided field logistical support. We thank the support of the Hamlet of Resolute and the Nunavut Research Institute. The authors wish to thank Antje Eulenburg, Christian Knoblauch, Birgit Grabellus, Justus Gimsa, Marek Jaskólski, Jennifer Krutzke, Nicole Mätzing, Paul Overduin, Julian Schneider and Samuel Stettner for their help in the instrumentation set up, data collection in the field and laboratory analyses. We acknowledge Saskia Foerster and Sabine Chabrillat (GFZ Potsdam) for providing access to the laboratory spectrometer. A special thanks to Cameron Eckert, Richard Gordon, Ricky Joe, Paden Lennie, Edward McLeod and Samuel McLeod for their support and helpful insights in the field. Many thanks also to the CBAWO 2017 field team.

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

**Figures**

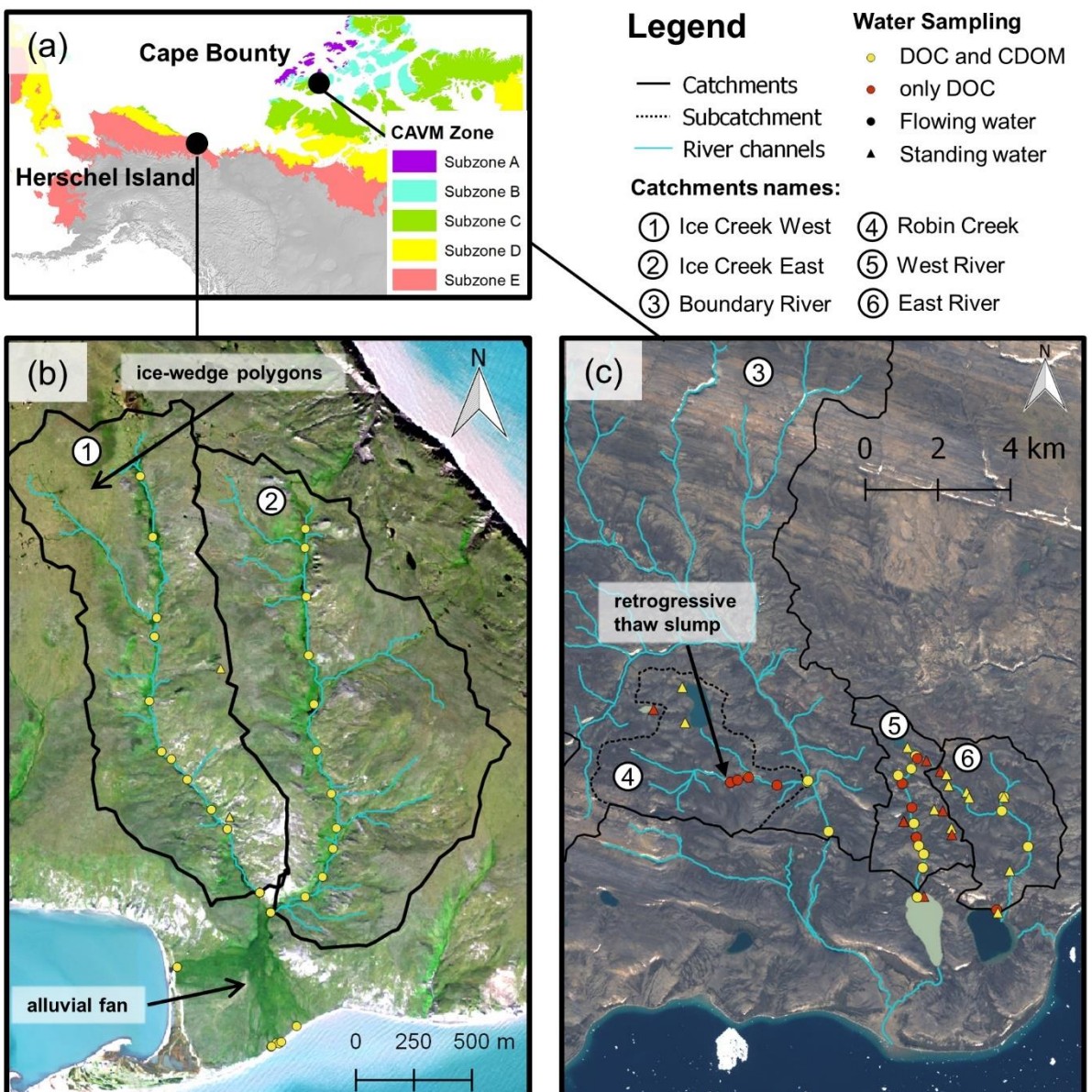

Figure 1. Maps of the study area showing (a) the location of Herschel Island and Cape Bounty in the Canadian Arctic including the Circumpolar Arctic Vegetation Map (CAVM) bioclimatic zones (Walker & Raymond 2016), (b) the studied catchments Ice Creek West and Ice Creek East on Herschel Island and (c) the studied catchments Boundary River with its subcatchment Robin Creek (dashed watershed), West River, and East River. The watershed names are indicated with numbers, and the general flow direction is southwards towards the ocean. Note that samples from flowing water (rivers and streams) are indicated by circles, whereas samples from standing water (ponds and lakes) are indicated by triangles. Yellow colors mark locations where DOC concentration and cDOM measurements are available, while only DOC concentrations are available at red locations. The background images are true color mosaics (Herschel Island: WorldView-3 quasi-true color RGB composite, acquired on 8 August 2015; Cape Bounty: Sentinel-2 quasi-true color RGB composite, acquired on 7 August 2016).

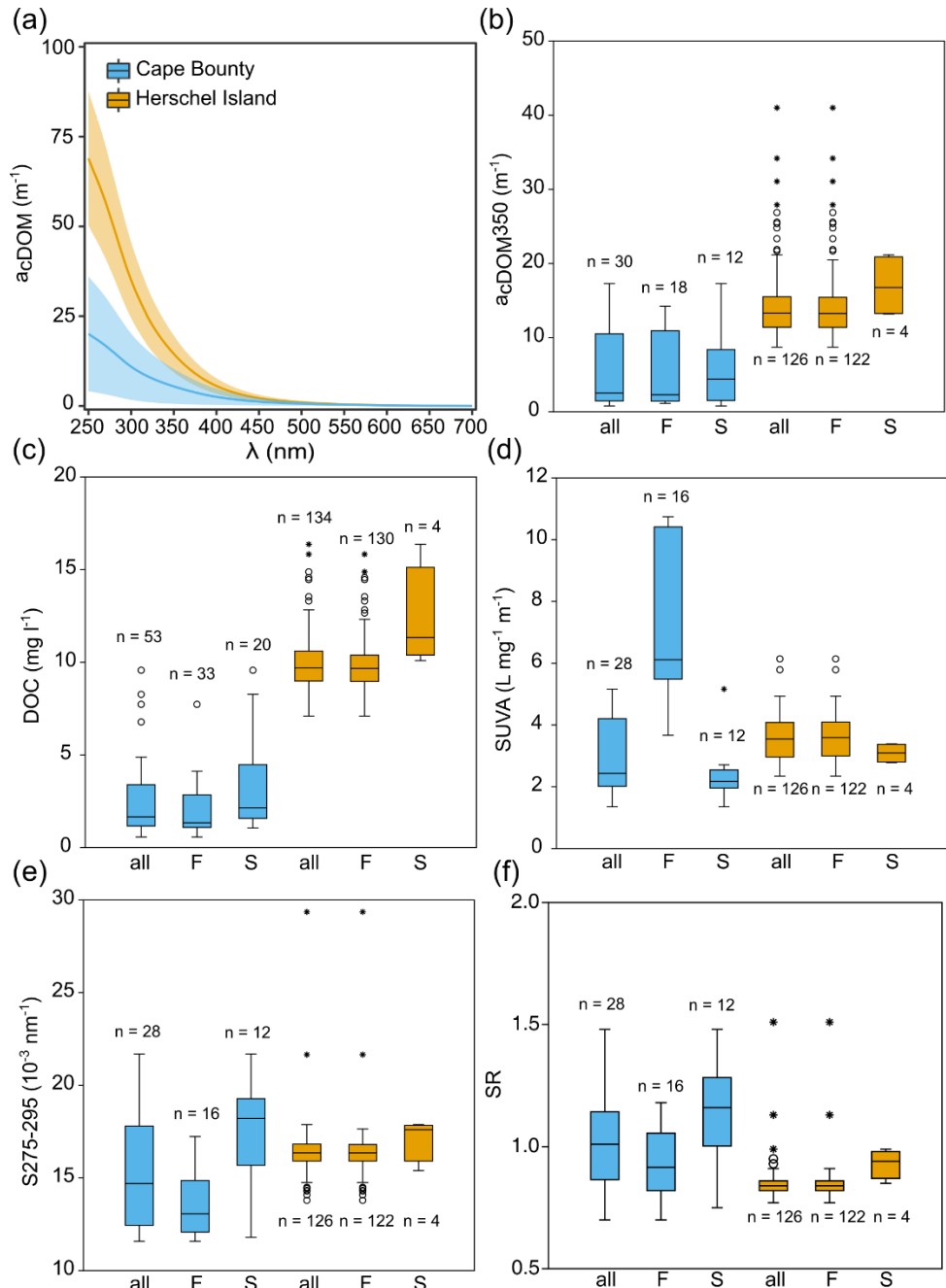

**Figure 2. Dissolved organic matter (DOM) quality and quantity for the sites from Herschel Island (HE, in orange) and Cape Bounty (CB, in blue). (a) Average cDOM absorption (m$^{-1}$) for the wavelengths ($\lambda$) between 250 and 700 nm. The colored shaded areas represent the standard deviation from the mean (solid line). Boxplots of (b) colored dissolved organic matter (cDOM) absorption at 350 nm a$_{cDOM}$350 (m$^{-1}$), (c) dissolved organic carbon, DOC (mg l$^{-1}$ (d) specific ultraviolet absorbance, SUVA (L mg$^{-1}$ m$^{-1}$), (e) cDOM Slope S275-295 (10$^{-3}$ nm$^{-1}$ and slope ratio SR. The plots depict distributions for all samples (all) and subsets of flowing water (F) and standing water (S) for each of the site.**

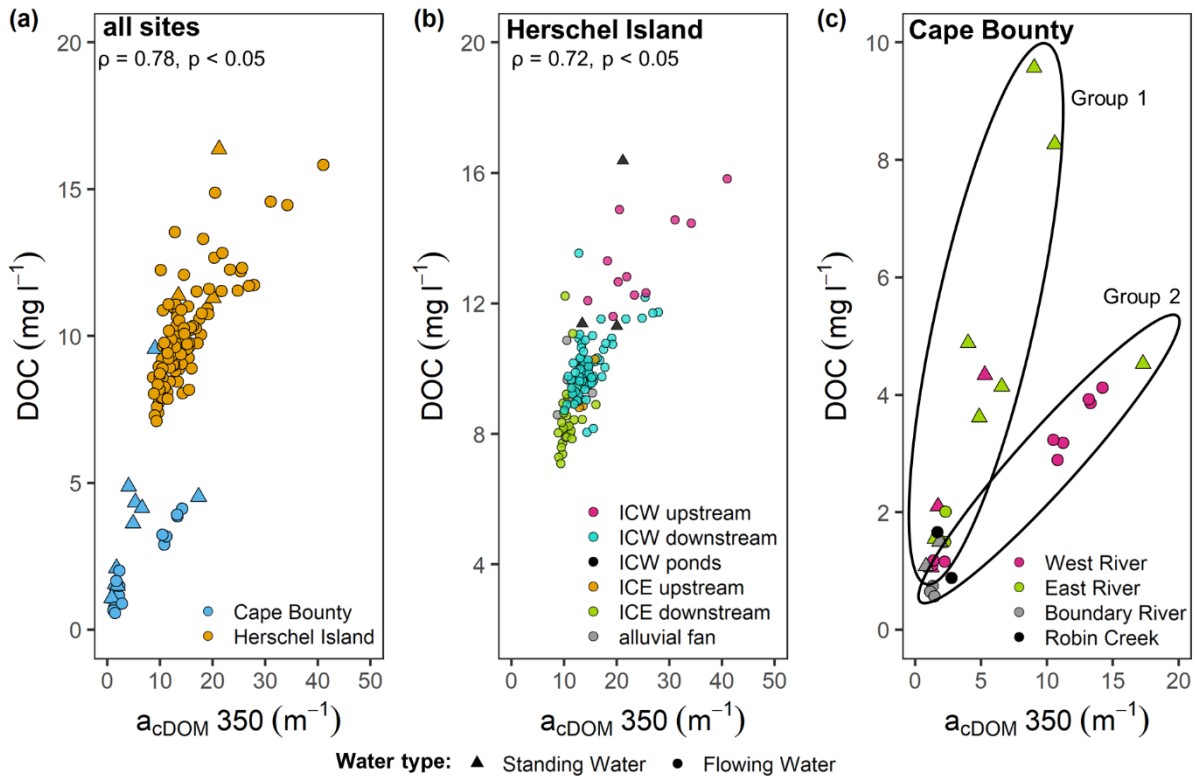

**Figure 3.** Absorption of colored dissolved organic matter (cDOM) at 350 nm (m⁻¹) versus dissolved organic carbon (DOC) concentration (mg l⁻¹) for (a) all sites, (b) sites on Herschel Island depicting the sampling locations Ice Creek West (ICW) upstream, downstream and ponds, Ice Creek East (ICE) upstream and downstream, and alluvial fan, and (c) sites at Cape Bounty (West River, East River, Boundary River, Robin Creek). Note that flowing water is indicated by a circle while standing water such as lakes or ponds is indicated by a triangle. The cDOM to DOC relationships are divided in two different groups (c).

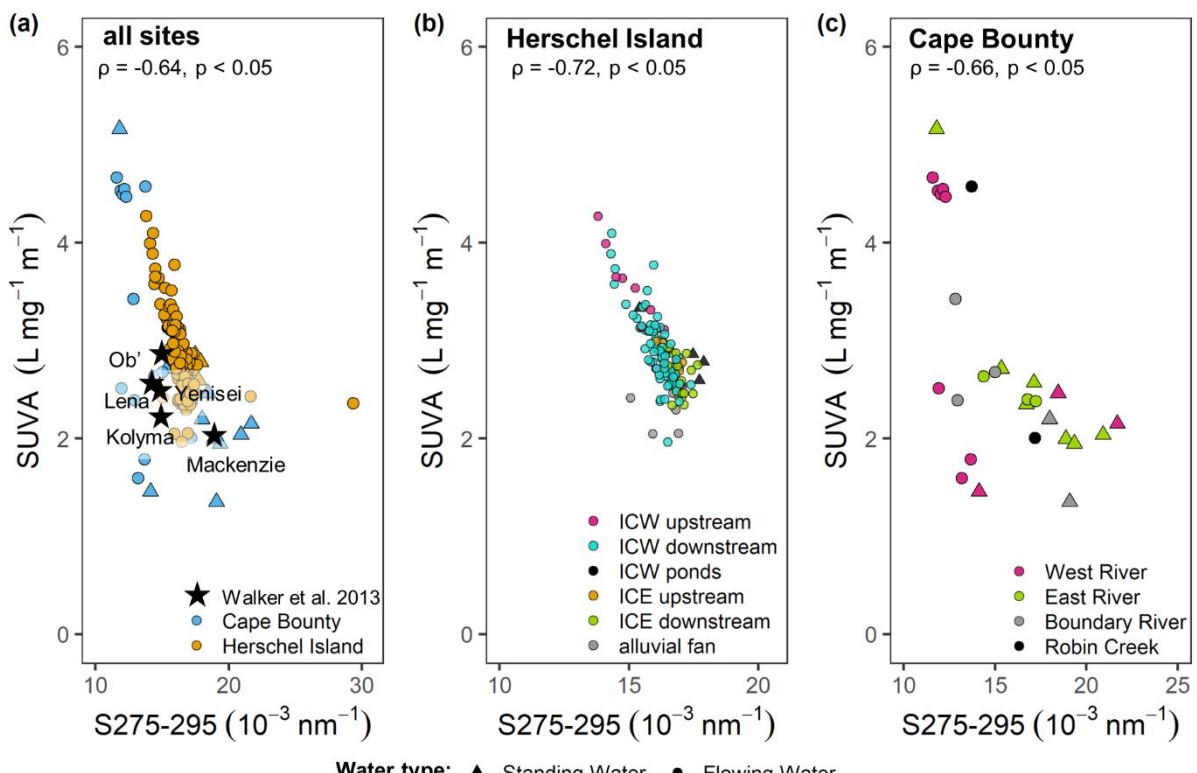

**Figure 4. Slope of colored dissolved organic matter ultraviolet cDOM UV absorption 275-295 ($10^{-3}$ nm$^{-1}$) versus specific ultraviolet absorbance SUVA (L mg$^{-1}$ m$^{-1}$) for (a) all sites, (b) sites on Herschel Island depicting the sampling locations Ice Creek West (ICW) upstream, downstream and ponds, Ice Creek East (ICE) upstream and downstream and alluvial fan, and (c) sites at Cape Bounty (West River, East River, Boundary River, Robin Creek). Note that flowing water is indicated by a circle while standing water such as lakes or ponds is indicated by a triangle. Midflow SUVA and S275-295 values for the large Arctic Rivers (Walker et al. 2013) are added to a).**

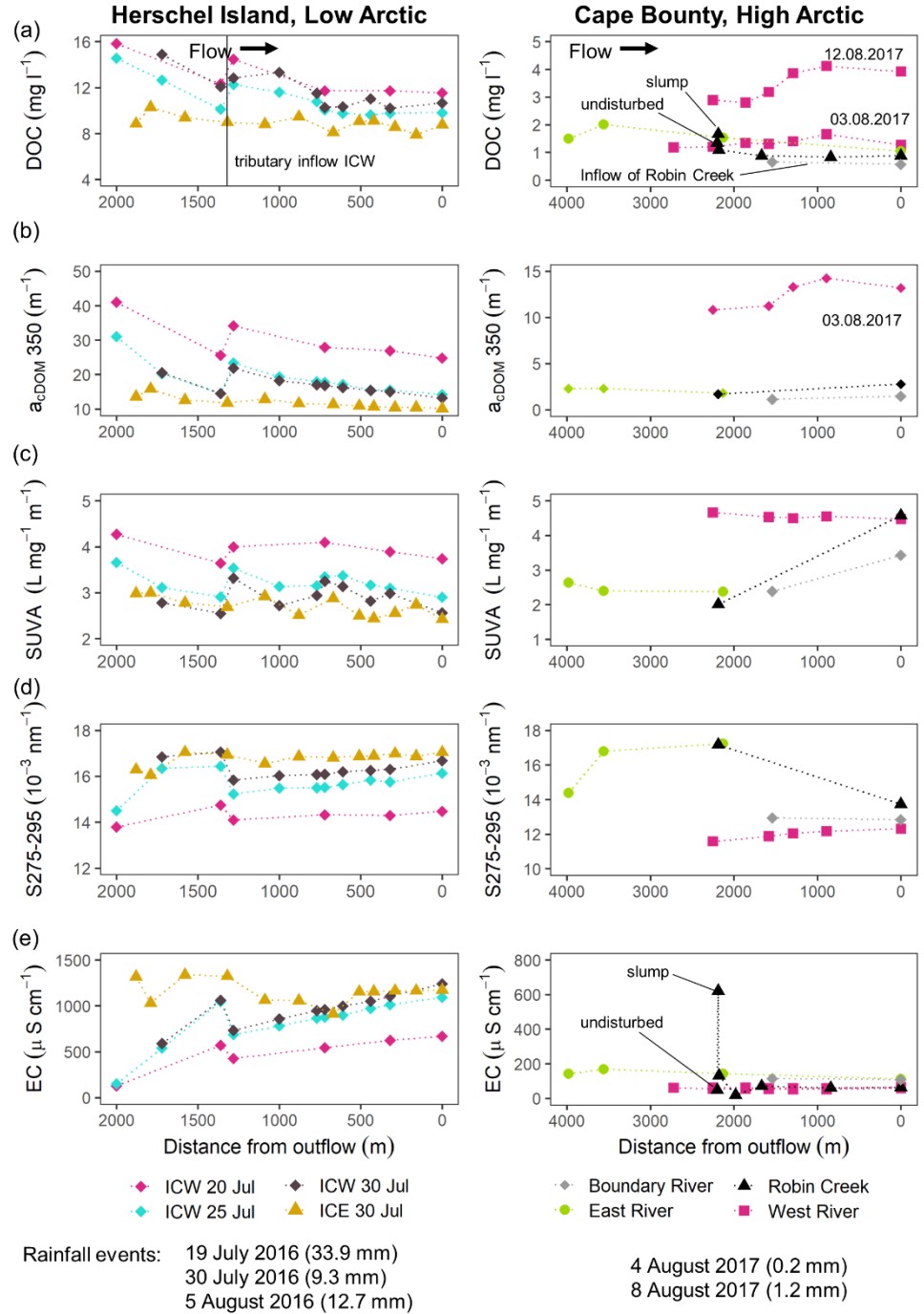

**Figure 5. Stream transects showing values of (a) dissolved organic carbon (DOC) concentration (mg l$^{-1}$), (b) absorption of colored dissolved organic matter (cDOM) at 350 nm, a$_{cDOM}$350 (m$^{-1}$), (c) specific ultraviolet absorbance SUVA (L mg$^{-1}$ m$^{-1}$), (d) cDOM Slope S275-295 (10$^{-3}$ nm$^{-1}$), (e) electrical conductivity (EC) for streams on Herschel Island (left) and Cape Bounty (right). Note that Ice Creek West (ICW) and Ice Creek East (ICE) on Herschel Island were sampled at different dates as indicated in the legend.**

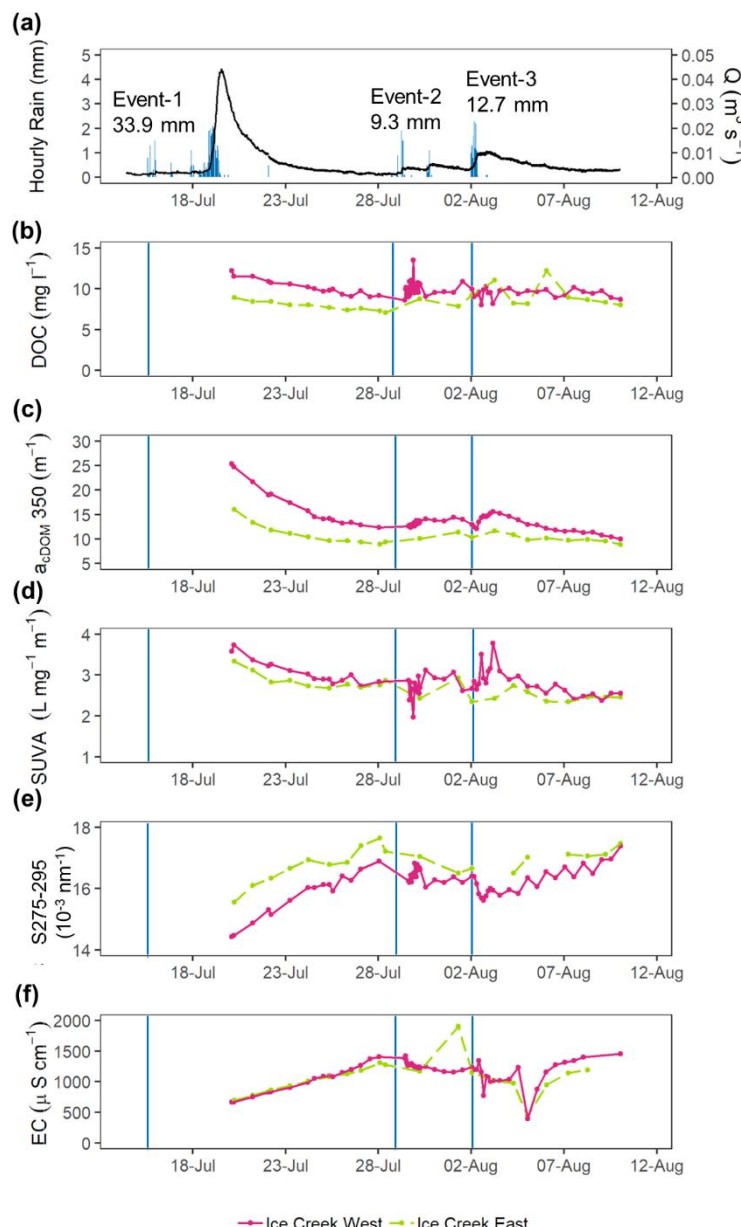

**Figure 6.** Time series from Herschel Island in 2016 showing (a) Discharge (m$^3$ s$^{-1}$) and hourly rainfall (mm) from Ice Creek West, (b) dissolved organic carbon (DOC) concentration (mg l$^{-1}$), (c) colored dissolved organic matter absorption at 350 nm, a$_{cDOM}$350 (m$^{-1}$), (d) specific ultraviolet absorbance SUVA (L mg$^{-1}$ m$^{-1}$) and (e) the cDOM slope S275-295 (10$^{-3}$ nm$^{-1}$) over the summer season 2016 for Ice Creek West (magenta) and Ice Creek East (green) respectively. The onset of rainfall events is marked with vertical blue lines. As described in the methods, DOC concentrations were corrected between 30 July and 7 August. The scale depicts only S275-295 values below 18 x 10$^{-3}$ nm$^{-1}$ to capture the variability, hence the two outliers in Ice Creek East (Fig. 2) are not displayed.

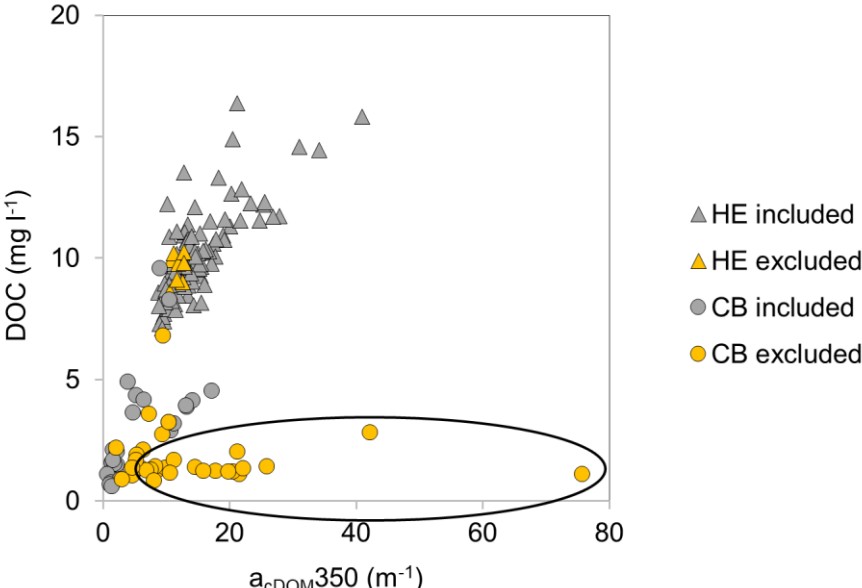

**Figure 7. Relationship between colored dissolved organic matter absorption $a_{cDOM}350$ (m$^{-1}$) and dissolved organic matter concentration DOC (mg l$^{-1}$) at Herschel Island (HE) displayed as triangle and Cape Bounty (CB) shown as circle. Samples marked in orange were excluded from the study due to flocculation after filtration (section 3.1). The samples circled in black show disproportionately high absorption values in relation to the DOC concentration.**

# Tables

**Table 1. Characteristics of studied watersheds on Herschel Island (Low Arctic) and Cape Bounty (High Arctic) showing catchment size (km$^2$), channel length (km), circumarctic vegetation map (CAVM) bioclimatic zone (CAVM, 2003), soil organic carbon content (SOCC) (Hugelius et al., 2013; Ramage et al., 2019) and maximum catchment elevation above sea level (m).**

| Site | Catchment size (km$^2$) | Channel length (km) | Vegetation zone (CAVM) | SOCC 0-30cm/0-100cm (kg m$^2$) | Maximum catchment elevation (m above sea level) |
|---|---|---|---|---|---|
| **Herschel Island, Low Arctic** | | | | | |
| Ice Creek West | 1.4 | 2.2 | Subzone D | | 88 |
| Ice Creek East | 1.6 | 1.9 | Subzone D | 11.4 / 26.4 | 95 |
| **Cape Bounty, High Arctic** | | | | | |
| Boundary River | 152.5 | 22.7 | Subzone B/C | | 213 |
| Robin Creek | 14.8 | 5.1 | Subzone B/C | | 151 |
| West River | 8.6 | 4.2 | Subzone B/C | 3.0 / 10.2 | 94 |
| East River | 12.0 | 5.2 | Subzone B/C | | 103 |

**Table 2.** Descriptive statistics (mean ± standard deviation) of dissolved organic carbon, DOC (mg l$^{-1}$), specific ultraviolet absorbance, SUVA (L mg$^{-1}$ m$^{-1}$), colored dissolved organic matter (cDOM) absorption at 350 nm, a$_{cDOM}$350 (m$^{-1}$), cDOM Slope S275-295 (10$^{-3}$ nm$^{-1}$), slope ratio SR, electrical conductivity EC (µS cm$^{-1}$), pH, and the number (n) of all samples/samples with cDOM absorption measurements. The statistics are given for specific streams, samples from flowing waters, standing waters, and all samples on Herschel Island (HE) and Cape Bounty (CB) respectively. The symbols ">" and "<" indicate significant inter-group differences at the alpha = 0.95 level. When the inter-group differences are significantly different at the alpha = 0.99 level, then they are underlined. When the difference is not significant, "≈" is used.

| Site | EC µS cm$^{-1}$ | pH | DOC mg l$^{-1}$ | SUVA L mg$^{-1}$ m$^{-1}$ | a$_{cDOM}$350 m$^{-1}$ | Slope 275-295 10$^{-3}$ nm$^{-1}$ | SR | n |
|---|---|---|---|---|---|---|---|---|
| **Herschel Island, Low Arctic** | | | | | | | | |
| Ice Creek West | 1050 ± 310 | 8.2 ± 0.2 | 10.4 ± 1.5 | 3.0 ± 0.4 | 16.1 ± 5.4 | 16.0 ± 0.7 | 0.83 ± 0.02 | 90/82 |
| Ice Creek East | 1030 ± 340 | 8.2 ± 0.2 | 8.7 ± 1.1 | 2.7 ± 0.2 | 11.1 ± 1.8 | 17.3 ± 2.4 | 0.90 ± 0.12 | 32/32 |
| Alluvial fan | 970 ± 170 | 7.8 ± 0.2 | 9.2 ± 0.9 | 2.5 ± 0.4 | 11.1 ± 1.9 | 16.3 ± 0.7 | 0.84 ± 0.02 | 8/8 |
| Flowing Water (all) | 1040 ± 310 | 8.2 ± 0.2 | 9.9 ± 1.5 | 2.9 ± 0.4 | 14.5 ± 5.1 | 16.4 ± 1.5 | 0.85 ± 0.07 | 130/122 |
| Standing Water (all) | 1440 ± 1300 | 8.3 ± 0.1 | 12.3 ± 2.8 | 2.9 ± 0.3 | 17.0 ± 4.2 | 17.1 ± 1.2 | 0.93 ± 0.06 | 4/4 |
| All samples | 1050 ± 370 | 8.2 ± 0.2 | 10.0 ± 1.6 | 2.9 ± 0.4 | 14.5 ± 5.1 | 16.4 ± 1.5 | 0.85 ± 0.07 | 134/126 |
| Standing (S) vs. Flowing (F) | S ≈ F | S ≈ F | S > F | S ≈ F | S ≈ F | S ≈ F | S > F | n.a. |
| **Cape Bounty, High Arctic** | | | | | | | | |
| Boundary River | 110 ± 3 | 7.1 ± 0.0 | 0.7 ± 0.1 | 2.8 ± 0.5 | 1.3 ± 0.2 | 13.1 ± 1.4 | 1.13 ± 0.06 | 3/3 |
| Robin Creek | 145 ± 213 | 7.3 ±0.6 | 1.1 ± 0.3 | 3.3 ± 1.8 | 2.2 ± 0.8 | 15.1 ± 2.3 | 1.03 ± 0.12 | 7/2 |
| West River | 60 ± 17 | 6.9 ± 0.3 | 2.5 ± 1.7 | 3.6 ± 1.4 | 8.5 ± 5.2 | 11.9 ± 0.8 | 0.86 ± 0.10 | 19/8 |
| East River | 141 ± 22 | 7.3 ± 0.1 | 1.5 ± 0.4 | 2.5 ± 0.1 | 2.1 ± 0.3 | 16.1 ± 1.6 | 0.95 ± 0.06 | 4/3 |
| Flowing Water (all) | 92 ± 101 | 7.0 ± 0.4 | 1.9 ± 1.5 | 3.2 ± 1.2 | 5.5 ± 5.1 | 13.1 ± 2.0 | 0.94 ± 0.13 | 33/16 |
| Standing Water (all) | 210 ± 160 | 7.5 ± 0.6 | 3.4 ± 2.4 | 2.4 ± 1.0 | 5.4 ± 4.9 | 17.4 ± 2.9 | 1.14 ± 0.20 | 20/12 |
| All samples | 137 ± 136 | 7.2 ± 0.5 | 2.5 ± 2.0 | 2.8 ± 1.1 | 5.5 ± 4.9 | 14.8 ± 3.2 | 1.02 ± 0.19 | 53/28 |
| Standing (S) vs. Flowing (F) | S > F | S > F | S > F | S < F | S ≈ F | S > F | S > F | n.a. |
| He_all vs. CB_all | HE > CB | HE > CB | HE > CB | HE > CB | HE > CB | n.a. | HE > CB | n.a. |

**Table 3. Correlation matrix using the Spearman's rho correlation coefficient between latitude, dissolved organic carbon concentration (DOC), colored dissolved organic carbon absorption at 350 nm (a$_{cDOM}$350), soil organic carbon content (SOCC) in 0-30 cm and 0-100 cm depth (Hugelius et al. 2014). Significance levels of p < 0.05 and p ≤ 0.01 are indicated.**

| | Latitude | a$_{cDOM}$350 | DOC | SOCC 0-30cm | SOCC 0-100cm |
|---|---|---|---|---|---|
| Latitude | 1.00 | **-0.22** | **-0.13** | **-0.19** | **-0.26** |
| a$_{cDOM}$350 | | 1.00 | **0.85** | **0.26** | **0.34** |
| DOC | | | 1.00 | **0.53** | **0.51** |
| SOCC 30cm | | | | 1.00 | **0.71** |
| SOCC 100cm | | | | | 1.00 |

