# Peer review of "Comparisons of DOM and its Optical Characteristics in Small Low and High Arctic catchments"

_Biogeosciences, 2019_

## Referee Comment (RC1) · Anonymous Referee #1 · 7 Feb 2019

The manuscript, "Characterizing organic matter composition in small Low and High Arctic catchments using terrestrial colored dissolved organic matter (cDOM)," presents a good body of work collected in vastly different Arctic catchments. The original data is strong and is mostly presented in a well-structured manner. Comparisons between the two sample locations show very different patterns with vegetation, latitude, rainfall events, and permafrost disturbance. Where the work requires attention is in the language used, sentence structure, some figure reorganizations/enhancements, and section reorganization. Following the major and minor revision suggestions below will greatly strengthen the manuscript.

[Figure]

Scientific significance: Does the manuscript represent a substantial contribution to scientific progress within the scope of Biogeosciences (substantial new concepts, ideas, methods, or data)? EXCELLENT

Scientific quality: Are the scientific approach and applied methods valid? Are the results discussed in an appropriate and balanced way (consideration of related work, including appropriate references)? BETWEEN GOOD AND FAIR The scientific approach and applied methods are valid – GOOD. The results are not discussed in a very balanced way – FAIR

Presentation quality: Are the scientific results and conclusions presented in a clear, concise, and well-structured way (number and quality of figures/tables, appropriate use of English language)? GOOD AND FAIR See comments regarding strong language, reevaluation, and reorganization that will improve the results and discussion sections, and the conclusion section should be reevaluated upon the completion of the rest of the edited sections. The number of Figures should be reevaluated based on the restructuring of the discussion section.

Major revisions points include: Adjusting weak language to strong scientific language. Examples are provided in the Line by Line revision points. The introduction is written well, but the results and discussion sections are written in a different style, with a narrative tone, that reads a bit too casually. Narrative writing styles are being encouraged in a great many manuscripts as long as the main messages of each sentence, section, and manuscript aren't lost. The recommendation under this point is to adjust the sentence structure to improve clarity, remove redundancy, and provide stronger scientific language. Briefly, language such as "There was, there are, we saw an initial drop in values..." should be replaced with less wordy narrative components where stating what was observed can lead to more clearer understanding. See comments below.

Some figures require extra attention to improve readability and understanding. Those comments can be found below the main text comments. Some figures are included

and not very well discussed in the text. Check through your figure list, decide which are very important for this work (including the possibility of moving figure S2 into the main text), and write appropriately about them. Figures that are included with little to no discussion should be deleted or moved to the supplemental section. Colors/symbols on most figures need to be improved (detailed comments below). Also Figures and Tables require all terms to be defined in the captions to avoid confusion. All acronyms and abbreviations should be written out as standalone text in the manuscript.

Some results regarding understanding the composition of DOM from SUVA, etc. are written inconsistently. Please re-read the results and discussion section to make sure this information is accurate and not just typos. Also, the flow of the discussion section will benefit from reorganizing the sections. Some important sections are listed last with figures as well, which doesn't strengthen the work. Think about the main message of the manuscript, adjust the title and flow of ideas throughout the manuscript to match the main message. Important points should be made up front (earlier in the discussion section) and even within paragraphs, not at the end. Consider making the important points first in the text, and then support the findings or contrast the findings with the literature information afterward.

Title What is terrestrial colored dissolved organic matter? Using the word terrestrial in the title is misleading. Consider a revision that highlights the strength of the conclusions. Rainfall events? Permafrost disturbances? Suggestion: Comparisons of chromophoric dissolved organic matter composition in small low and high Arctic catchments OR Comparisons of cDOM composition with permafrost disturbance in small low and high Arctic catchments

Line by Line revision points (major and minor included). The comments are organized by sections of the manuscript, including Figures, Tables, and Supplemental material. Page numbers and Line number are provided.

Note: Check the manuscript for fluctuating usage of colored and coloured.

Abstract Line 20 Please define SOCC

Line 28-29 How are permafrost-derived DOM vs fresh derived DOM being defined in this study?

Line 30 What does fresher DOM prone to degradation mean? Photo? Microbial? Combination? The abstract does not describe the composition of the DOM. Stronger color of DOM does not describe more aromatic and/or lignin-type constituents. What are the absorption results besides "things change downstream"? Consider a more specific details.

Line 31 "This work shows that optical properties of DOM will be a useful tool for understanding DOM sources and quality at a pan-Arctic scale" Yes, the work does, but the abstract doesn't. Consider blending the ideas together so that the abstract matches the measurements made, chemical interpretations, and conclusions from the work.

Introduction The introduction is nicely written and sets the stage very well. Consider a stronger ending so that it will tie in well with the discussion points and the relevance of small watershed importance with global carbon budgets and vulnerable environments with climate change.

Page 2 Line 21 Typo CDOM instead of cDOM. Please check.

Page 2 Line 25 What is cDOM-DOC? Concentration?

Page 3 Line 6-7 This sentence could be improved by describing the importance of this contribution to global carbon budgets as the climate warms. Consider ending with a stronger contribution statement.

Study Area Page 3 Line 15 Add SOCC here.

Methods Page 5 Line 28 Typo CDOM instead of cDOM. Please check.

Results Page 6 Line 23 Consider revising the subheading to DOC concentration and cDOM absorption characteristics. Usually with a heading that lists specific items, they

then appear in that order in the text. Think about this heading and whether it makes more sense to report DOC concentration before the absorbance data.

Page 6 Line 23, 24, and 27 Typo CDOM instead of cDOM. Please check.

Page 6 Line 27 Will CDOM slope or spectral slope be used? Also, the spectral slope of both are within the same boundaries if accounting for the standard deviation. Will this similarity be discussed? The sentence as currently written suggests that they are significantly different. Please clarify.

Page 7 Line 2 Revise the sentence to list concentration at the beginning of the sentence for improved sentence structure. Then that word can be deleted in the next line.

Page 7 Line 3 It appears as though the lowercase L and the number one are very near identical or identical looking to read. Consider using a capital L for Liters.

Page 7 Line 5 Consider using the word significantly in this results section when a significance value has been calculated. In this sentence it makes sense and it also makes sense in Line 1, however, this word is used in every sentence thus far on this page. Edit the results section accordingly to use the word significantly or significant when it is appropriate. Also, in this line, an open parenthesis is missing.

Page 7 Line 9 refers to different slopes in Figure 3c. Might adding the slope line/trend line or some kind of calculated slope help readers visualize this difference? This relationship is not clear from the data in Figure 3c with overlapping flowing water and standing water symbols. The overlapping data is at low DOC concentration and low absorbance at 350nm. When those values are increased, there may be a change in the grouping. Can that be reported and highlighted in the figure more clearly? Consider a reevaluation of the data and how those results will be reported.

Page 7 Line 13-14 This information is already reported in the first paragraph and commented on above.

Page 7 Line 14-15 Consider reporting in the text that this is a negative relationship.

Page 7 Line 16 We jumped from Figure 3c to Figure 4b. Please correct.

Page 7 Line 16-17 Redundant sentence, please delete. What outliers?

Page 7 Line 21 Same comment as the subheading for 4.2 Consider inserting the word concentration after DOC and a descriptive term for the cDOM measurement reported. This also comes up in Section 4.4 and the reason why it's misleading is because the DOC measurement is a quantitative value and the cDOM measurement involves both qualitative information and some quantitative normalization. Is the usage of cDOM all the time in these headings the best idea? What about using DOM and then describing the quantitative and qualitative information below? For example, 4.3 DOM patterns along longitudinal transects AND 4.4 DOM temporal trends with rainfall

Page 7 Line 23-24 This information seems to be in correct based on the figure for both the characterization of DOC concentration in Ice Creek East and West.

Page 7 Line 26-27 The usage of the word "low" in this sentence is misleading. Please describe the data more accurately. Yes, it is lower than Herschel Island, which is what it is assumed to be compared to, but the wording is weak. Describe the trends of the DOC concentration in Cape Bounty first, then make comparisons. Plenty of streams and rivers have DOC concentrations below 1 mg/L, so think about specific word usage when reporting the results.

Page 7 Line 27 Consider this revision to improve clarity, ". . .levels of DOC concentration compared to other Cape Bounty rivers. . ." Also, this is the same trend as the other West River DOC concentration data (without the rainfall event) and that is important to note.

Page 7 Line 28-29 How is that information supported from the figure shown? It looks like three data points are right on top of each other, which suggests they are not longitudinally or hydrologically separated. This information should be clarified.

Page 7 Line 30 No clear pattern was detected in Boundary River? The figure shows two data points here which suggests that a pattern would be tough to determine. Perhaps

report the similarities between concentrations of Boundary River and Robin Creek?

Page 8 Line 1 Good – we should hope so given the positive relationship. Consider strengthening this sentence by noting the strong relationship between these two parameters. For example, "This confirms the strong positive relationship between both parameters."

Page 8 Line 2-3 Same comment regarding the usage of the word low. Describe the data as remaining constant or with very little variation. Using the word low assumes a comparison. If the intention is to make a comparison, then describe it clearly.

Page 8 Line 4 Didn't DOC and absorbance also show different trends between these river systems? Consider revising this sentence to flow better with the previous text.

Page 8 Line 4-6 Why isn't the increasing trend at ∼1300m reported and discussed for DOC concentration, absorbance, and SUVA in the Herschel Island system?

Page 8 Line 5-6 This is a clear sentence highlighting a comparison between rainfall events. The manuscript can be strengthened by making this point clearer throughout the results and discussion sections with these types of comparisons highlighted on the figures. Use this as a strength moving forward.

Page 8 Line 8 Certainly this could be due to some inputs?

Page 8 Line 9-14 Slope values? Spectral slopes? Or flow gradients? A notation of which figure is being discussed here should be included.

Page 8 Line 16 "Electrical conductivity was found to increase..." This is weak scientific writing. Consider using less words to be clearer and strengthen the main message, e.g., "Electrical conductivity increased from..."

Page 8 Line 16-18 A notation of which figure is being discussed here should be included.

Page 8 Line 22-24 Please reference the specific part of the figure.
Page 8 Line 24 The word "drop" is weak scientific language. Consider using "decrease" in this sentence.

Page 8 Line 20-26 This section describes the results organized in Figure 6 a-f, yet the results are written a bit out of order ending with information seen in 6a. Annotate the text with the specific parts of the figure that is being discussed.

Page 8 Line 27 "The hydrochemical response to the following rainfall event (Event-3, 12.7 mm) was different to the previous one." This is a weak opening sentence. Be more specific to hold the reader's attention. The response of Event 3 was different than the response of Event 2, correct? State that using stronger scientific language. This type of writing continues on in "Here, we saw an initial drop in. . .". DOC, absorbance, and Spectral slope decreased after the event, followed by a sharp increase. . . This section is difficult to follow with the events only listed on one part of the figure. Consider marking all a-f figures with a vertical line highlighting the rainfall events.

Page 8 Line 28-29 What does this mean? "SUVA shows an increase with two positive peaks." All the values are positive, so please describe increases and spikes in the data to higher values using stronger scientific language.

Page 8 Line 30 "No continuous slope records are available for this event as two outliers occurred in this event" This can't be evaluated without seeing the data or reading about how outliers were determined. Consider showing the data in the SI or discussing how outliers were calculated and extracted.

Discussion Page 9 Line 3 Typo measurement should be measurements. Also, limitations in the measurement itself or the sample? The next sentence discusses precipitates. Clarify the limitation because certainly there are limitations in absorbance measurements to infer biogeochemical relationships.

Page 9 Line 3-5 Redundant language and weak writing. Consider stronger language, for example, "Some samples formed small precipitates, which partly remained in suspension or accumulated at the bottom of the bottles."

Page 9 Line 5-6 Consider being more specific with the end of this sentence. Precipitation occurred after filtration during storage, correct? Note the time of storage and any other conditions that are relevant. The way the sentence currently reads assumes immediate precipitation, which probably did not happen.

Page 9 Line 6 "In the absorption spectra, these samples showed extraordinarily high acDOM values..." This is redundant. Consider this revision, "These samples had very high absorbance values at 350nm..." and consider reporting those values. None of this data can be evaluated so "extraordinarily high" holds no water for the reader. A comparison to the DOC concentration level – what is meant by this? Were the samples settled before running the absorbance measurements? Or were the precipitates blocking, filtering, or absorbing some of the light?

Page 9 Line 6-8 "As described in the methods section (3.1), they were therefore excluded from the study based on the laboratory notes." This type of writing is redundant and without understanding what the values were before exclusion or any of these laboratory notes, the reader cannot evaluate or confirm any of this information.

Page 9 Line 8 "At Cape Bounty, this was the case for 25 out of 55 samples." This is very disappointing. Was no redissolution or shaking attempted? This is practically half the data set!

Page 9 Line 13 Meaning absorbance interferences due to the sample and not the method?

Page 9 Line 14 "The cut off between solid and dissolved fraction in a solution is normally made..." Use caution here. Dissolved organic matter is operationally defined as material that can pass through a $1.0\mu m$ filter poresize. What is listed here is just a few examples of filter poresize used commonly in the DOM aquatic community. Please revise this language.

Page 9 Line 15 Please add a comma after e.g.

Page 9 Line 18 Please add a reference to this statement.

Page 9 Line 21-22 For what environments? 12% cannot be evaluated without an environmental reference and ties to comparisons of the percentage range or difference in the outlier values.

Page 9 Line 24 There is no filter difference in this study, correct? What is meant with this statement?

Page 9 Line 26-27 "Dissolved iron in terrestrially dominated waters is dominantly complexed with humic and fulvic acids" Wouldn't this suggest that the "outliers" could also have been influenced by this effect? Was iron measured in this study? Are there any references to iron concentrations in this region?

Page 9 Line 27-28 Did pH and temp change? The reference to Table 2 this late in the manuscript seems a bit out of place. This is good information that should be known before the discussion. Consider moving this table to the results section.

Page 10 Line 5-6 This is the first mention of iron concentrations being measured in this study. Please revise Table 1 or Table 2 to include this important information. Add a methods section describing these measurements. Also, the iron concentration figure in the supplemental (S2) is great and should be added to the main text.

Page 10 Line 9 "Therefore, all problematic samples were removed from this study." Understandable, but the work would be strengthened if the reader could see all these data and relationships, and then this discussion section would make a lot more sense. This section defines limitations regarding data that isn't presented.

Page 10 Line 12 Is this a typo? "Our both study sites..."? Use Our or Both our to start this sentence.

Page 10 Line 17-20 Is the 195 a typo?

Page 10 Line 20 Please insert a reference.

Page 11 Line 4 What is a full response of a rainfall event? This sentence is very confusing.

Page 11 Line 6-7 Consider revising this sentence to improve clarity. This indicates a decrease in aromaticity and a shift to lower molecular weight, which suggests. . .

Also, please define what is meant by labile material. Labile from a microbial perspective?

Page 11 Line 8 What is a clear increase?

Page 11 Line 10-11 Confusing sentence. The meaning is meant to be about the rain itself or the river? During the event or after?

Page 11 Line 16 The duration of the rainfall event seems very important. This point should be included earlier in the text, added into a table, or gray shading can indicate the duration in Figure 6.

Page 11 Line 23-24 A tremendous increase? Compared to what?

Page 12 Line 16 Redundant portion of the sentence - delete "across the Arctic"

Page 13 Line 1 ". . .constant proportion of bioavailable DOC. . ." Meaning concentration or qualitative nature? The meaning of DOC is dissolved organic carbon and doesn't inherently imply concentration so the usage of DOC in this manuscript should be clarified where appropriate and this section of the discussion needs to include more descriptive qualitative or quantitative language.

Page 13 Line 4-6 Example of weak language and very confusing ideas. How was the influence of ice wedge polygons assessed? The information is provided after the confusing sentence. Please reorganize and use concise language.

Page 13 Line 6-7 But not upstream? This sentence does not make sense as written.

Page 13 Line 8-9 How does this make sense from the previous statements? The flow of this paragraph is very confusing.

Page 13 Line 10-11 Why is rainfall discussed again in this permafrost impact section? Is that the disturbance? Clearer ideas need to be presented.

Page 13 Line 13 "SUVA and S275-295 do not show strong differences downstream in the West River." This is a result. Why is this? Is this discussed?

Page 13 Line 18 What does a shallow S275-295 mean?

Page 13 Line 20 In this sentence, low aromaticity is linked with SUVA increases, yet a few sentences ago it is linked with decreases in SUVA. This is very confusing. Greater SUVA values mean???

Page 13 Line 20-30 This section was difficult to read and understand the flow of the main discussion points. Please reorganize and put main discussion points up front in the section, then provide supporting evidence throughout the paragraph.

Page 13 Line 31 What is cDOM-DOC? And this section seems really important. Can it be reorganized earlier in the discussion section. If the figures are being kept in this section, then they will appear earlier. A reference to Figure1 might also assist in the terrestrial/nature argument of the different catchments.

Page 13 Line 32-33 This is another example of weak language. Consider this revision to improve scientific language and flow of ideas. "Strong positive correlations between DOC and acDOM350 were previously reported in similar Arctic rivers and globally (insert references).

Page 13-14 Line 1-2. This information stops the flow of the discussion. Consider removing the sentence, keep the references, and reorganize the next sentence to include them.

Page 14 Line 4 "This means that...." is an example of weak language. Consider

revising these two thoughts into one sentence with a connecting word like "indicating" so that unnecessary words are removed, and the main messages are clear.

Page 14 Line 4-6 Why is the point of stating this?

Page 14 Line 1-6 Is the point of this section to state the good correlation and proxy for DOC concentration using absorbance? Figure inclusion and discussion should be an important component of the manuscript. Why is it important in this work? Think about the distributions of the data and the relationships to the other work. Does the other comparative work have similar geographical features? Ice-wedges? Etc.?

Page 14 Line 7-8 Very confusing sentence. Another example of weak language. Is this referring to concentration and a directional trend?

Page 14 Line 9 Delete "where a large range of absorption values is covered". This is redundant. Check each sentence for repetition and redundant ideas.

Page 14 Line 11 Going back to Figure 7? Consider keeping Figure 7 discussion in the same section.

Page 14 Line 17 "higher aromaticity, which suggests that the material is fresh and prone to degradation" and fresh material? Fresh from what? Fresh as considered by what? Light? Microbes? Terrestrial soils?

Page 14 Line 18-23 This seems like important information to put in the results section. Then it can be discussed in this section. Consider reorganizing this section.

Page 14 Line 26-27 This point was just made in the discussion section and not fully developed to be included yet in this conclusion section. How are the linkages supported?

Page 14 Line 29-31 These points needs to be clearer in the results and discussion section. Please reorganize.

Page 15 Line 2-3 Redundant sentence. Please delete.

Page 15 Line 4 Fresher DOM prone to degradation means what? How is fresh defined? What type of degradation?

Page 15 Line 7 This idea needs further development in this work and cannot be a standalone conclusion. The same comment can be applied for the remaining conclusion statements.

Figure 1 In (a) it is a little confusing that ocean and glaciers are white? Is that correct? Where are the glaciers? Consider using line and dotted line symbols in the legend for catchment and subcatchment areas so that readers don't look for boxed regions. Can river flow direction be added to these (b) and (c) figures? The legend is written well and was easy to read. Consider two revisions to include the word "concentration" when referring to DOC measurements and define CAVM in the caption.

Figure 2 Very aesthetically pleasing, well done. In the caption, please revise the opening statement to "Dissolved organic matter (DOM) absorption characteristics from Herschel..." so that all the terms are defined.

Figure 3 Great ideas here, just need slight improvements to enhance understanding and readability. Define the terms in the caption, DOM, DOC, ICE, ICW, etc. Next, the symbols of circles and triangles indicating flowing and standing water are good, but too small in all these figures. Also, triangles and circles overlapping each other look like blobs. Consider open and closed symbols to improve readability. The data blobs are hardest to read in (a). The choice of pink and red or purple and pink colors are too close together to visualize clearly in (b) and (c). Consider using light green and dark green (or some similar color tone gradient) for upstream and downstream to keep that data grouped together aesthetically. Add trend lines for (c) to show the different slopes or box/circle the two different groups to help visualize the differences discussed in the text.

Figure 4 Same comments as Figure 3 with caption definitions, data point size, circles and triangles, and color tones. Also, is the variability of Cape Bounty discussed in the

text? These figures should really tight groupings for Herschel but not Cape Bounty.

Figure 5 Similar comments to Figure 3 and 4. Same sites should use colors that fall in the same family with different gradients so they can be linked visually on the figures. Shades of green IC East on different sample dates will help. Keep acronyms similar among figures, e.g., IC East vs. ICE. These figures have a lot of gridlines on them which makes the dotted line hard to follow. Consider removing the gridlines or thickening the dotted lines. Also, consider using different symbols for different river samples. Define the terms in the caption.

Figure 6 The gridlines wash out the green data points and lines. Consider changing the color scheme and increasing the size of the connected lines and data points. Adding a vertical line through all figures for each rain event will improve these figures. Missing data should be notated in the caption. If all the data was collected in 2016, please remove the 2016 date indicator on the x-axis because it is very crowded. The legend also includes IC west and IC East. Please make this consistent with the other notations in the previous figures and define all the terms.

Figure 7 Good figure. If it is needed still in the manuscript, since only two to three sentences discuss parts of it, then keep it with some improvements. Keep the reference on the figure but put the regions for the samples it is referring to in the figure caption and remove these words from the figure (it crowds the data). Define the terms in the figure caption. There are multiple data sets with the same color assigned to them. Please select different colors to see the different groups represented on this figure. Also, consider including the slope calculated from this work as a comparison to the literature calculated slope.

Figure 8 Good figure. If it is needed still in the manuscript, since only two to three sentences discuss parts of it, then keep it with some improvements. Define the terms in the caption. Is it possible to put black outlines around the Permafrost extent legend colors? The isolated patches color is very difficult to read in the legend. Also, mark the

color of the ocean, since it is nearly identical to the isolated patches color, or change the ocean color to something darker?

Table 1 Define the term CAVM in the caption. Some formatting of this table is confusing like the dark thick line near the top and then a defining line combining ICW, ICE, and CB. Consider using indents for the sample names under the low and high Arctic categories, using another horizontal row divider (as in Table 2), or separate columns.

Table 2 Define the terms in the caption (all abbreviations and acronyms) and provide an explanation for underlining as a useful tool for these statistical comparisons. Typo at the bottom line "He" should read "HE"

Table 3 Define the terms in the caption.

Supplemental Information Define the terms in the figure and table captions. Consider moving S2 to the main manuscript.
* * *

---

## Referee Comment (RC2) · Anonymous Referee #2 · 18 Feb 2019

In this study, Coch et al. undertake a comparative assessment of dissolved organic carbon and optical (as a350, SUVA, and spectral slopes) measurements. The authors use measurements from catchments on Herschel Island and Cape Bounty, using a transect design to explore changes in DOC concentration and DOM character across regions and with movement downstream. As described below (and, as discussed by the authors) there are some issues with the optical data that appear to be still outstanding. As described in greater detail in the overarching comments, it would also be nice to see the authors more clearly elucidate how their study represents a step forward in DOM dynamics in sub-Arctic and high Arctic regions.

[Figure]

Overarching comments

My most significant comment on the manuscript is the concerns related to Fe interference. Given the large scatter in SUVA and slope results for Cape Bounty, even across the stream sites that appear to not be affected by particularly long residence times, I think that this is an issue that must be dealt with before these data can be interpreted soundly. I don't have great confidence that the authors' approach of discarding samples that had evidence of flocculation was able to fully ameliorate this issue. Perhaps there is some residual sample that could be analyzed for Fe? A high level of confidence in the optical measurements is quite critical for the integrity of the manuscript.

A second high level comment is that I would like to see the authors do a better job of putting their work in the context of what has been done previously in the arena of DOC and cDOM in Arctic stream networks and elsewhere. At some points (see specific comments below) it seemed as if the text was focusing more on re-iterating previous findings, and less on carving out how the results from this study advance our knowledge. Ideally, a revised manuscript would have a much clearer emphasis on the latter.

Some editing for English grammar is also needed throughout the manuscript. I certainly have sympathy for non-native English speakers who are having to write in a second language! Perhaps some of the co-authors could assist with this sort of an edit.

Figure quality could be improved, particularly for figures 3 – 6.

Lake residence time: there is some discussion on effects of lakes vs. streams on cDOM in the two regions. Presumably photobleaching is more prevalent at Cape Bounty. Knowing something about the residence time (or, even rough volume / mean depth of these systems) early on in the manuscript would help greatly with this interpretation. My understanding is that the 'lakes' on Cape Bounty are relatively large: perhaps the Herschel systems have a very low residence time by comparison? I see that this information is provided towards the end of the paper, but it would be helpful to have it

presented as part of the study site description.

Specific comments (as page / line number):

1/16, 2/13: What is small? The catchments being studied are very small indeed, for catchments discharging straight to the Arctic Ocean. It is not correct to state that direct-export catchments of this size cover 40

1/20: you don't test for variation with SOCC and vegetation cover: "consistent with variation in vegetation cover and SOCC between the two sites"?

1/21: I would keep lignin out of the abstract, seeing as you don't measure this at all

2/20-21: cDOM, or CDOM? I'm not a strong proponent for one vs. the other, but it would be good to make sure you're being consistent.

2/32: The Spence et al. 2015 reference is incorrect here. Perhaps a mis-placement?

3/15: add "cm" to specification of active layer depth.

3/17 and elsewhere: better specified as "C: N"

4/5: Mean July temperature? A bit confusing if not specified.

4/6: "with baseflow re-establishing"?

4/21: Manual outlet samples taken at what frequency?

4/22: Can you clarify this sampling design? How many sampling points along this transect? Was there a pre-determined distance between points? Adding a reference to Fig. 1 would help here.

4/27: Herschel bottles were also triple-rinsed? It might be useful to start with a general sampling scheme at the top of this section. Also see comments above on clarifying the sampling scheme.

5/23: One technique to deal with particles is to subtract the average 700-800 nm baseline. This is a good practice for all samples, to correct for interference from colloids, etc., that might not be easily visible to the eye. See also 5/27 below.

5/27: This subtraction will correct for scatter (see above) but not for drift in instrument output across the range of wavelengths measured over hours of instrument use. The latter can be corrected for by measuring blanks (or, other standards) at specified time periods, and correcting measurements to this change.

5/31: Were SUVA corrected for Fe? Substantial Fe could present challenges to your ability to interpret these results. Other studies have found fairly high Fe levels in the western Canadian Arctic, and Fe can also be one instigator of DOM flocculation. Some of your higher SUVA values do suggest possible interference from Fe. Reading on, I see you have some text on this below; see my later comments on this issue.

Section 3.2: Specify statistical packages used?

6/16: Here and elsewhere, please specify what your +/- values indicate. Standard error? 95

6/27: Difference in slopes is not significant, given your error bounds? In addition, you might find it useful to express your slopes as values x $10^3$. $This is not uncommon in the literature, and might help with visualizing differences between sites, etc.$

7/8-9: Separation into two groups: In Figure 3c, however, it looks like you only provide statistics for a single slope. Why not calculate both slopes, and test – statistically – whether they are different?

7/13: This significant difference finding is interesting, because your error bounds overlap. It's difficult to assess this as a reader without knowing what's being presented as a metric for dispersion in the data. I'm not sure I would analyze the data in this way given that (from Figure 5) your slope values cover a similar (wide) range at each of the two sites. You also have substantially different n for your two sites, which could confound your statistical analyses.

Section 4.4 / Figure 6: What about using C-Q (i.e., hysteresis) plots to illustrate these responses. I think this would help elucidate better what is happening across the three events. For some of these events, I'm not sure I see much of a response, or at least – it's a bit difficult to tease out with the current presentation.

Section 5.1 / first paragraph: Again, this makes me concerned about interference from Fe.

Page 9 / line 25: Ah – yes! I see you get to Fe here. At the bottom of this paragraph (8/10): I'm really not sure you have eliminated all of the problematic samples. If you have extra water from these sites, it would be great to have this analyzed for Fe, if you haven't already. It seems you are using precipitation as a proxy for Fe interference? You may certainly have high Fe in non-precipitate samples. Particularly given the scatter in your data for a series of samples taken upstream of a lake (e.g., West River vs. East River in Figure 4c), I think you need to be concerned about whether some of the patterns you're observing are 'real'. For the Herschel Island samples, where it appears that Fe data are available, it would be great to correct for this.

5.2.1, first paragraph: I agree that you see higher DOM quantity at Herschel. From Figure 4 and 5, however, I don't think you can conclude there's any difference in quality. The quality values span across a similar range at both sites, with values from Cape Bounty showing a wider range. However, given your discussion on Fe, it seems possible that this difference in variation may be somewhat spurious.

5.2.1, second paragraph: it would be nice to introduce this difference in residence time earlier on, perhaps in the study site descriptions. This would put your results in context as they are being presented. In addition, this discussion on increased bacterial and photo-degradation in lakes and ponds could use a reference or two.

Section 5.2.2, first paragraph: it would be useful to consider and cite some of the literature investigating source limitation vs. transport limitation in this section. In addition, although I am not familiar with the earlier publication by Coch et al., it's unclear from

how this paragraph is written whether this analysis is contributing anything new. If there is something novel that's being presented in this particular ms, it would be good to structure the text in a way that really highlights that fact.

Section 5.2.2, third paragraph: it seems to me that this is perhaps the novel information that's being presented in this section; the second paragraph, as written, also seems to largely summarize findings from previous studies. Why not flesh this out a bit and focus here? For example – what is the evidence for increased permafrost DOM export with increasing baseflow; I may have missed this discussion above, but it would be nice to have this laid out very clearly. It would also be useful to cite the Spence et al. publication that seemed to be mis-placed above in this section.

12/23: See also the conceptual work on headwater to mainstem gradients by Drake et al.

12/25: IE – temporal variation? Within sampling dates, the SUVA and slopes are fairly consistent along the transect; the values are certainly much more variable across time than across space.

14/4: cDOM as a good proxy for DOC concentration. This is true (and quite well established) in cases where most DOM is terrestrial in origin, and not overly degraded. There are a few references you could cite here for studies that have made this point using pretty extensive datasets. It's not necessarily true universally, however; there's also some good papers showing a lack of relationship between DOC and colour for sites where DOM is highly reworked / photobleached – see for example work by Arts et al., Osburn et al. and others on prairie / great plains lakes.

14/7: What about the relationship between CDOM and soil organic carbon content in the catchment of study? Presumably there is a strong relationship here (see, for example Connolly et al. 2018, ERL). It seems likely that climate / latitude is a controlling variable in the sense that it has such an important influence on soils. Ah – I see you have this in the paragraph below. It would be good to look at the recent work by Connoly

et al, who also examine this relationship across a variety of watershed sizes.

---

## Author Comment (AC1) · 12 May 2019

-We are very grateful for the time invested and the constructive comments, which helped to improve the paper tremendously. We have carefully reviewed the comments and have revised the manuscript accordingly. Our responses are given in a line by line list below.

General Comments:

The manuscript, "Characterizing organic matter composition in small Low and High Arctic catchments using terrestrial colored dissolved organic matter (cDOM)," presents

a good body of work collected in vastly different Arctic catchments. The original data is strong and is mostly presented in a well-structured manner. Comparisons between the two sample locations show very different patterns with vegetation, latitude, rainfall events, and permafrost disturbance. Where the work requires attention is in the language used, sentence structure, some figure reorganizations/enhancements, and section reorganization. Following the major and minor revision suggestions below will greatly strengthen the manuscript.

Scientific significance: Does the manuscript represent a substantial contribution to scientific progress within the scope of Biogeosciences (substantial new concepts, ideas, methods, or data)? EXCELLENT Scientific quality: Are the scientific approach and applied methods valid? Are the results discussed in an appropriate and balanced way (consideration of related work, including appropriate references)? BETWEEN GOOD AND FAIR The scientific approach and applied methods are valid GOOD. The results are not discussed in a very balanced way FAIR Presentation quality: Are the scientific results and conclusions presented in a clear, concise, and well-structured way (number and quality of figures/tables, appropriate use of English language)? GOOD AND FAIR See comments regarding strong language, reevaluation, and reorganization that will improve the results and discussion sections, and the conclusion section should be reevaluated upon the completion of the rest of the edited sections. The number of Figures should be reevaluated based on the restructuring of the discussion section.

Major revisions points include: Adjusting weak language to strong scientific language. Examples are provided in the Line by Line revision points. The introduction is written well, but the results and discussion sections are written in a different style, with a narrative tone, that reads a bit too casually. Narrative writing styles are being encouraged in a great many manuscripts as long as the main messages of each sentence, section, and manuscript aren't lost. The recommendation under this point is to adjust the sentence structure to improve clarity, remove redundancy, and provide stronger scientific language. Briefly, language such as "There was, there are, we saw an initial drop in

values. . ." should be replaced with less wordy narrative components where stating what was observed can lead to more clearer understanding. See comments below. -We adjusted the language according to the line by line revision points. Many thanks for giving such thorough feedback.

Some figures require extra attention to improve readability and understanding. Those comments can be found below the main text comments. Some figures are included and not very well discussed in the text. Check through your figure list, decide which are very important for this work (including the possibility of moving figure S2 into the main text), and write appropriately about them. Figures that are included with little to no discussion should be deleted or moved to the supplemental section. Colors/symbols on most figures need to be improved (detailed comments below). Also Figures and Tables require all terms to be defined in the captions to avoid confusion. All acronyms and abbreviations should be written out as standalone text in the manuscript. -We revised the figures according to the detailed suggestions below. We decided to move figures 7 and 8 to the supplementary material.

Some results regarding understanding the composition of DOM from SUVA, etc. are written inconsistently. Please re-read the results and discussion section to make sure this information is accurate and not just typos. Also, the flow of the discussion section will benefit from reorganizing the sections. -We revised the respective sections according to the detailed comments below.

Some important sections are listed last with figures as well, which doesn't strengthen the work. Think about the main message of the manuscript, adjust the title and flow of ideas throughout the manuscript to match the main message. Important points should be made up front (earlier in the discussion section) and even within paragraphs, not at the end. Consider making the important points first in the text, and then support the findings or contrast the findings with the literature information afterward. -Please find the comments in the sections below.

Title What is terrestrial colored dissolved organic matter? Using the word terrestrial in the title is misleading. Consider a revision that highlights the strength of the conclusions. Rainfall events? Permafrost disturbances? Suggestion: Comparisons of chromophoric dissolved organic matter composition in small low and high Arctic catchments OR Comparisons of cDOM composition with permafrost disturbance in small low and high Arctic catchments -This is a very good suggestion. We revised the title to: "Comparisons of chromophoric dissolved organic matter composition in small low and high Arctic catchments"

Line by Line revision points (major and minor included).

The comments are organized by sections of the manuscript, including Figures, Tables, and Supplemental material. Page numbers and Line number are provided. Note: Check the manuscript for fluctuating usage of colored and coloured. -We checked for consistency.

Abstract

Line 20 Please define SOCC -We defined SOCC in the text.

Line 28-29 How are permafrost-derived DOM vs fresh derived DOM being defined in this study? -Permafrost-derived DOM is sourced from deeper in the active layer whereas fresh derived DOM is sourced at the surface of the active layer. We added a definition for clarification.

Line 30 What does fresher DOM prone to degradation mean? Photo? Microbial? Combination? The abstract does not describe the composition of the DOM. Stronger color of DOM does not describe more aromatic and/or lignin-type constituents. What are the absorption results besides "things change downstream"? Consider a more specific details. -"Fresh" DOM means near-surface derived and therefore prone to degradation. We added more specific details.

Line 31 "This work shows that optical properties of DOM will be a useful tool for understanding DOM sources and quality at a pan-Arctic scale" Yes, the work does, but the abstract doesn't. Consider blending the ideas together so that the abstract matches the measurements made, chemical interpretations, and conclusions from the work. -We edited the abstract according to the suggestions.

Introduction The introduction is nicely written and sets the stage very well. Consider a stronger ending so that it will tie in well with the discussion points and the relevance of small watershed importance with global carbon budgets and vulnerable environments with climate change. -This was changed accordingly.

Page 2 Line 21 Typo CDOM instead of cDOM. Please check.  -This was changed accordingly.

Page 2 Line 25 What is cDOM-DOC? Concentration? -They refer to ratios. The text is changed accordingly to "Previous studies have focused on characterizing cDOM-DOC ratios for the large Arctic rivers and shelf areas"

Page 3 Line 6-7 This sentence could be improved by describing the importance of this contribution to global carbon budgets as the climate warms. Consider ending with a stronger contribution statement. -This was changed accordingly.

Study Area

Page 3 Line 15 Add SOCC here. -We added SOCC.

Methods

Page 5 Line 28 Typo CDOM instead of cDOM. Please check. -It was changed according to the suggestion.

Results

Page 6 Line 23 Consider revising the subheading to DOC concentration and cDOM absorption characteristics.  Usually with a heading that lists specific items, they then appear in that order in the text. Think about this heading and whether it makes more

sense to report DOC concentration before the absorbance data. Page 6 Line 23, 24, and 27 Typo CDOM instead of cDOM. Please check. -It was changed according to the suggestion.

Page 6 Line 27 Will CDOM slope or spectral slope be used? Also, the spectral slope of both are within the same boundaries if accounting for the standard deviation. Will this similarity be discussed? The sentence as currently written suggests that they are significantly different. Please clarify. -During the study we use spectral slopes of cDOM for the wavelength ranges 275 to 295nm (S275-295) and 350 to 400nm (S350-400). Further the ratio of both is reported as slope ratio SR (S275-295 : S350-400). In case of line 27, we decided to report the slope ratio. The sentence is changed accordingly. Differences in cDOM spectral slopes and slope ratio are discussed later on.   Page 7 Line 2 Revise the sentence to list concentration at the beginning of the sentence for improved sentence structure. Then that word can be deleted in the next line. -It was changed according to the suggestion.

Page 7 Line 3 It appears as though the lowercase L and the number one are very near identical or identical looking to read. Consider using a capital L for Liters. -Although it would indeed improve readability, we decided to stay with the SI units as suggested by the journal.

Page 7 Line 5 Consider using the word significantly in this results section when a significance value has been calculated. In this sentence it makes sense and it also makes sense in Line 1, however, this word is used in every sentence thus far on this page. Edit the results section accordingly to use the word significantly or significant when it is appropriate. Also, in this line, an open parenthesis is missing. -We carefully went through the results section to check the use of the word "significant". In most cases, significance values were calculated, making the use of the word appropriate (see Table 2). We changed it where possible (lines 14,

Page 7 Line 9 refers to different slopes in Figure 3c. Might adding the slope line/trend

line or some kind of calculated slope help readers visualize this difference? This relationship is not clear from the data in Figure 3c with overlapping flowing water and standing water symbols. The overlapping data is at low DOC concentration and low absorbance at 350nm. When those values are increased, there may be a change in the grouping. Can that be reported and highlighted in the figure more clearly? Consider a reevaluation of the data and how those results will be reported. -We considered adding a slope/trend line to this figure. However, the data is not normally distributed, hence, the use of Spearman's rho. Using a straight line would suggest a regression curve and that the data is normally distributed. We added ellipses to show the "grouping".

Page 7 Line 13-14 This information is already reported in the first paragraph and commented on above. -We deleted this sentence.

Page 7 Line 14-15 Consider reporting in the text that this is a negative relationship. -The sentence was changed according to the suggestion.

Page 7 Line 16 We jumped from Figure 3c to Figure 4b. Please correct. -A reference to Figure 4a was inserted.

Page 7 Line 16-17 Redundant sentence, please delete. What outliers? -The sentence was changed according to the suggestion.

Page 7 Line 21 Same comment as the subheading for 4.2 Consider inserting the word concentration after DOC and a descriptive term for the cDOM measurement reported. This also comes up in Section 4.4 and the reason why it's misleading is because the DOC measurement is a quantitative value and the cDOM measurement involves both qualitative information and some quantitative normalization. Is the usage of cDOM all the time in these headings the best idea? What about using DOM and then describing the quantitative and qualitative information below? For example, 4.3 DOM patterns along longitudinal transects AND 4.4 DOM temporal trends with rainfall -This is a valid point. We changed headings 4.2, 4.3 and 4.4 referring to DOM instead of DOC and

cDOM individually.

Page 7 Line 23-24 This information seems to be in correct based on the figure for both the characterization of DOC concentration in Ice Creek East and West. -We changed the wording to make the meaning clearer.

Page 7 Line 26-27 The usage of the word "low" in this sentence is misleading. Please describe the data more accurately. Yes, it is lower than Herschel Island, which is what it is assumed to be compared to, but the wording is weak. Describe the trends of the DOC concentration in Cape Bounty first, then make comparisons. Plenty of streams and rivers have DOC concentrations below 1 mg/L, so think about specific word usage when reporting the results. -We edited the sentences according to the suggestions.

Page 7 Line 27 Consider this revision to improve clarity, ". . .levels of DOC concentration compared to other Cape Bounty rivers. . ." Also, this is the same trend as the other West River DOC concentration data (without the rainfall event) and that is important to note. -The sentence was changed accordingly.

Page 7 Line 28-29 How is that information supported from the figure shown? It looks like three data points are right on top of each other, which suggests they are not longitudinally or hydrologically separated. This information should be clarified. -The points seem on top of each other, because they are only a few metres away. They are indeed hydrologically connected. We

Page 7 Line 30 No clear pattern was detected in Boundary River? The figure shows two data points here which suggests that a pattern would be tough to determine. Perhaps report the similarities between concentrations of Boundary River and Robin Creek? -We edited the sentences according to the suggestions.

Page 8 Line 1 Good – we should hope so given the positive relationship. Consider strengthening this sentence by noting the strong relationship between these two parameters. For example, "This confirms the strong positive relationship between both

parameters." -We added "strong" to the sentence.

Page 8 Line 2-3 Same comment regarding the usage of the word low. Describe the data as remaining constant or with very little variation. Using the word low assumes a comparison. If the intention is to make a comparison, then describe it clearly. -We edited the sentence according to the suggestions.

Page 8 Line 4 Didn't DOC and absorbance also show different trends between these river systems? Consider revising this sentence to flow better with the previous text. -Less datapoints are available for the absorption measurement than there were for DOC. This is why no further trends can be described here.

Page 8 Line 4-6 Why isn't the increasing trend at âĹij1300m reported and discussed for DOC concentration, absorbance, and SUVA in the Herschel Island system? -We added this description. This is due to another

Page 8 Line 5-6 This is a clear sentence highlighting a comparison between rainfall events. The manuscript can be strengthened by making this point clearer throughout the results and discussion sections with these types of comparisons highlighted on the figures. Use this as a strength moving forward.

Page 8 Line 8 Certainly this could be due to some inputs? -Yes, we think so too.

Page 8 Line 9-14 Slope values? Spectral slopes? Or flow gradients? A notation of which figure is being discussed here should be included. -We inserted the reference to the figure.

Page 8 Line 16 "Electrical conductivity was found to increase. . ." This is weak scientific writing. Consider using less words to be clearer and strengthen the main message, e.g., "Electrical conductivity increased from. . ." -We changed it accordingly.

Page 8 Line 16-18 A notation of which figure is being discussed here should be included. -We inserted the reference to the figure.

Page 8 Line 22-24 Please reference the specific part of the figure. -We inserted the reference to the figure.

Page 8 Line 24 The word "drop" is weak scientific language. Consider using "decrease" in this sentence. -We changed the wording accordingly.

Page 8 Line 20-26 This section describes the results organized in Figure 6 a-f, yet the results are written a bit out of order ending with information seen in 6a. Annotate the text with the specific parts of the figure that is being discussed. -We referenced the figure parts accordingly.

Page 8 Line 27 "The hydrochemical response to the following rainfall event (Event-3, 12.7 mm) was different to the previous one." This is a weak opening sentence. Be more specific to hold the reader's attention. The response of Event 3 was different than the response of Event 2, correct? State that using stronger scientific language. This type of writing continues on in "Here, we saw an initial drop in. . .". DOC, absorbance, and Spectral slope decreased after the event, followed by a sharp increase. . . This section is difficult to follow with the events only listed on one part of the figure. Consider marking all a-f figures with a vertical line highlighting the rainfall events. -We changed the wording accordingly.

Page 8 Line 28-29 What does this mean? "SUVA shows an increase with two positive peaks." All the values are positive, so please describe increases and spikes in the data to higher values using stronger scientific language. -We changed the wording accordingly.

Page 8 Line 30 "No continuous slope records are available for this event as two outliers occurred in this event" This can't be evaluated without seeing the data or reading about how outliers were determined. Consider showing the data in the SI or discussing how outliers were calculated and extracted. -We adjusted the scale to capture the full variability. We reworded the sentence to make this clear.

  Discussion

Page 9 Line 3 Typo measurement should be measurements. Also, limitations in the measurement itself or the sample? The next sentence discusses precipitates. Clarify the limitation because certainly there are limitations in absorbance measurements to infer biogeochemical relationships. -We changed the wording as suggested.

Page 9 Line 3-5 Redundant language and weak writing. Consider stronger language, for example, "Some samples formed small precipitates, which partly remained in suspension or accumulated at the bottom of the bottles." -We changed the sentence structure accordingly.

Page 9 Line 5-6 Consider being more specific with the end of this sentence. Precipitation occurred after filtration during storage, correct? Note the time of storage and any other conditions that are relevant. The way the sentence currently reads assumes immediate precipitation, which probably did not happen. -This is true. We added the storage time to the methods description.

Page 9 Line 6 "In the absorption spectra, these samples showed extraordinarily high acDOM values. . ." This is redundant. Consider this revision, "These samples had very high absorbance values at 350nm. . ." and consider reporting those values. None of this data can be evaluated so "extraordinarily high" holds no water for the reader. A comparison to the DOC concentration level – what is meant by this? Were the samples settled before running the absorbance measurements? Or were the precipitates blocking, filtering, or absorbing some of the light? -We added clarifications to the paragraph.

Page 9 Line 6-8 "As described in the methods section (3.1), they were therefore excluded from the study based on the laboratory notes." This type of writing is redundant and without understanding what the values were before exclusion or any of these laboratory notes, the reader cannot evaluate or confirm any of this information. -We added specific values and a figure to the supplementary material.

Page 9 Line 8 "At Cape Bounty, this was the case for 25 out of 55 samples." This is very disappointing. Was no redissolution or shaking attempted? This is practically half the data set! -Unfortunately not.

Page 9 Line 13 Meaning absorbance interferences due to the sample and not the method? -This was added to the sentence.

Page 9 Line 14 "The cut off between solid and dissolved fraction in a solution is normally made. . ." Use caution here. Dissolved organic matter is operationally defined as material that can pass through a $1.0\mu$m filter poresize. What is listed here is just a few examples of filter poresize used commonly in the DOM aquatic community. Please revise this language. -We edited the sentence accordingly.

Page 9 Line 15 Please add a comma after e.g. -Comma added.

Page 9 Line 18 Please add a reference to this statement. We added the reference. Page 9 Line 21-22 For what environments? 12% cannot be evaluated without an environmental reference and ties to comparisons of the percentage range or difference in the outlier values. -We added that it was a terrestrial water body.

Page 9 Line 24 There is no filter difference in this study, correct? What is meant with this statement? -We revised the sentence for clarification: We therefore assume that colloid complexes between 22 $\mu$m and 0.7 $\mu$m have a minor influence on cDOM absorption in our samples.

Page 9 Line 26-27 "Dissolved iron in terrestrially dominated waters is dominantly complexed with humic and fulvic acids" Wouldn't this suggest that the "outliers" could also have been influenced by this effect? Was iron measured in this study? Are there any references to iron concentrations in this region? -We revised this section for clarification. Iron data from a previous study only shows total iron (Fe(II) and Fe(III)). Correction coefficients by Poulin et al. 2014 are based on Fe(III) values.

With regard to the "outliers": In comparison to the Herschel site, the Cape Bounty

site indeed shows a larger range of values. We found that the range in SUVA and slopes at the sampling sites is due to the different nature of the sites themselves (e.g. influenced by permafrost degradation, pulse of rainfall delivering fresh DOM). We found different water types with different transparency, which regulate the photodegradation of cDOM. Thus, changes in absorption, SUVA and cDOM slope can be explained by catchment properties and/or rainfall events (see Figure 3). It might also be interesting to note that catchments at Herschel cover an area of 3 km2 in total, whereas the sampled area at Cape Bounty covered about 30 km2. This naturally results in a greater heterogeneity (and range) of optical parameters. We added that information to the study site description.

We are very confident that discarding samples based on flocculation notes actually did ameliorate the issue. To support this argument, we added a figure to the supplementary material showing DOC vs. acDOM350 for all included and excluded samples across the sites. At Cape Bounty many of the samples had SUVA values above 6, meaning that the cDOM values were too high for the low DOC concentrations. The maximum SUVA recorded in the excluded samples amounted to 59.5 L mg-1 m-1.

Furthermore, the relationship between cDOM350 and DOC of all included samples from both study sites are within the error range of other published samples from similar arctic aquatic environments (Fig. S3). If cDOM absorption data used in this study had been strongly interfered by iron colloids, the goodness regression of the relationship would be significantly lower.

Page 9 Line 27-28 Did pH and temp change? The reference to Table 2 this late in the manuscript seems a bit out of place. This is good information that should be known before the discussion. Consider moving this table to the results section. -We referenced the table in the results section.

Page 10 Line 5-6 This is the first mention of iron concentrations being measured in this study. Please revise Table 1 or Table 2 to include this important information. Add a

methods section describing these measurements. Also, the iron concentration figure in the supplemental (S2) is great and should be added to the main text. -Measurements of iron are available from a previous study. We inserted the correct reference. See the detailed comment above.

Page 10 Line 9 "Therefore, all problematic samples were removed from this study." Understandable, but the work would be strengthened if the reader could see all these data and relationships, and then this discussion section would make a lot more sense. This section defines limitations regarding data that isn't presented. -We added the data to the manuscript and expanded the discussion.

Page 10 Line 12 Is this a typo? "Our both study sites. . ."? Use Our or Both our to start this sentence. -Thanks for pointing it out. It was indeed a typo.

Page 10 Line 17-20 Is the 195 a typo? -Thanks for pointing it out. It was indeed a typo.

Page 10 Line 20 Please insert a reference. -The references for the entire paragraph are found in the end.

Page 11 Line 4 What is a full response of a rainfall event? This sentence is very confusing. -We edited the sentence for clarity.

Page 11 Line 6-7 Consider revising this sentence to improve clarity. This indicates a decrease in aromaticity and a shift to lower molecular weight, which suggests. . . Also, please define what is meant by labile material. Labile from a microbial perspective? -We revised the sentence for clarity.

Page 11 Line 8 What is a clear increase? -The sentence was edited for clarity.

Page 11 Line 10-11 Confusing sentence. The meaning is meant to be about the rain itself or the river? During the event or after? -It meant the runoff during the event. We edited the sentence to make this clear.

Page 11 Line 16 The duration of the rainfall event seems very important. This point

should be included earlier in the text, added into a table, or gray shading can indicate the duration in Figure 6. -We added the onset of the rainfall events to the figure.

Page 11 Line 23-24 A tremendous increase? Compared to what? -Compared to the pre-rainfall conditions - Sentence was edited to make this clear.

Page 12 Line 16 Redundant portion of the sentence - delete "across the Arctic" -This was done according to the suggestion

Page 13 Line 1 ". . .constant proportion of bioavailable DOC. . ." Meaning concentration or qualitative nature? The meaning of DOC is dissolved organic carbon and doesn't inherently imply concentration so the usage of DOC in this manuscript should be clarified where appropriate and this section of the discussion needs to include more descriptive qualitative or quantitative language. -This is a very good point. We edited the sentence.

Page 13 Line 4-6 Example of weak language and very confusing ideas. How was the influence of ice wedge polygons assessed? The information is provided after the confusing sentence. Please reorganize and use concise language. -We reorganized the paragraph.

Page 13 Line 6-7 But not upstream? This sentence does not make sense as written. -This section was also reorganized and rewritten.

Page 13 Line 8-9 How does this make sense from the previous statements? The flow of this paragraph is very confusing. -This section was also reorganized and rewritten.

Page 13 Line 10-11 Why is rainfall discussed again in this permafrost impact section? Is that the disturbance? Clearer ideas need to be presented. -This pattern only becomes apparent after the rainfall event.

Page 13 Line 13 "SUVA and S275-295 do not show strong differences downstream in the West River." This is a result. Why is this? Is this discussed? We edited the section. Page 13 Line 18 What does a shallow S275-295 mean? -We replaced it with "low".

Page 13 Line 20 In this sentence, low aromaticity is linked with SUVA increases, yet a few sentences ago it is linked with decreases in SUVA. This is very confusing. Greater SUVA values mean??? -This was a typo. Greater SUVA and low S275-295 mean high aromacity and higher molecular weight (and vice versa).

Page 13 Line 20-30 This section was difficult to read and understand the flow of the main discussion points. Please reorganize and put main discussion points up front in the section, then provide supporting evidence throughout the paragraph. -We revised the paragraph

Page 13 Line 31 What is cDOM-DOC? And this section seems really important. Can it be reorganized earlier in the discussion section. If the figures are being kept in this section, then they will appear earlier. A reference to Figure1 might also assist in the terrestrial/nature argument of the different catchments. -We reorganized the section.

Page 13 Line 32-33 This is another example of weak language. Consider this revision to improve scientific language and flow of ideas. "Strong positive correlations between DOC and acDOM350 were previously reported in similar Arctic rivers and globally (insert references). -We revised the sentence accordingly.

Page 13-14 Line 1-2. This information stops the flow of the discussion. Consider removing the sentence, keep the references, and reorganize the next sentence to include them. -This was revised.

Page 14 Line 4 "This means that. . .." is an example of weak language. Consider revising these two thoughts into one sentence with a connecting word like "indicating" so that unnecessary words are removed, and the main messages are clear. -It was revised as suggested.

Page 14 Line 4-6 Why is the point of stating this? -They were removed.

Page 14 Line 1-6 Is the point of this section to state the good correlation and proxy for DOC concentration using absorbance? Figure inclusion and discussion should be

an important component of the manuscript. Why is it important in this work? Think about the distributions of the data and the relationships to the other work. Does the other comparative work have similar geographical features? Ice-wedges? Etc.? -We expanded here. The figure was moved to the supplementary material.

Page 14 Line 7-8 Very confusing sentence. Another example of weak language. Is this referring to concentration and a directional trend? -We changed the wording.

Page 14 Line 9 Delete "where a large range of absorption values is covered". This is redundant. Check each sentence for repetition and redundant ideas. -We deleted the phrase.

Page 14 Line 11 Going back to Figure 7? Consider keeping Figure 7 discussion in the same section. -We edited the section accordingly.

Page 14 Line 17 "higher aromaticity, which suggests that the material is fresh and prone to degradation" and fresh material? Fresh from what? Fresh as considered by what? Light? Microbes? Terrestrial soils? -"Fresh" is used as "less altered" permafrost DOC (Vonk et al. 2015 - Biodegradability of dissolved organic carbon in permafrost soils and aquatic systems: a meta-analysis

Page 14 Line 18-23 This seems like important information to put in the results section. Then it can be discussed in this section. Consider reorganizing this section. -We moved this section earlier into the discussion.

Page 14 Line 26-27 This point was just made in the discussion section and not fully developed to be included yet in this conclusion section. How are the linkages supported? -We expanded on this topic in the discussion section 5.2.1 Page 14 Line 29-31 These points needs to be clearer in the results and discussion section. Please reorganize.

Page 15 Line 2-3 Redundant sentence. Please delete. -We deleted the sentence.

Page 15 Line 4 Fresher DOM prone to degradation means what? How is fresh defined? What type of degradation? -We added clarification to this sentence.

Page 15 Line 7 This idea needs further development in this work and cannot be a standalone conclusion. The same comment can be applied for the remaining conclusion statements. -We extended the discussion on rainfall event impacts to support this conclusion.

Figure 1 In (a) it is a little confusing that ocean and glaciers are white? Is that correct? Where are the glaciers? Consider using line and dotted line symbols in the legend for catchment and subcatchment areas so that readers don't look for boxed regions. Can river flow direction be added to these (b) and (c) figures? The legend is written well and was easy to read. Consider two revisions to include the word "concentration" when referring to DOC measurements and define CAVM in the caption. -Thanks for pointing this out. In fact, the glaciers are not visible / existent at this scale or in this area respectively. We therefore decided, to remove them from the map. We changed the symbology of the catchments and subcatchments as suggested. Adding the flow direction made the figures appear very crowded and covered too much of the image. Instead, we added the flow direction to the text of the caption.

Figure 2 Very aesthetically pleasing, well done. In the caption, please revise the opening statement to "Dissolved organic matter (DOM) absorption characteristics from Herschel. . ." so that all the terms are defined. -We revised the caption as suggested, and also removed the gridlines from the figure.

Figure 3 Great ideas here, just need slight improvements to enhance understanding and readability. Define the terms in the caption, DOM, DOC, ICE, ICW, etc. Next, the symbols of circles and triangles indicating flowing and standing water are good, but too small in all these figures. Also, triangles and circles overlapping each other look like blobs. Consider open and closed symbols to improve readability. The data blobs are hardest to read in (a). The choice of pink and red or purple and pink colors are too close together to visualize clearly in (b) and (c). Consider using light green and dark green (or some similar color tone gradient) for upstream and downstream to keep that data grouped together aesthetically. Add trend lines for (c) to show the different

slopes or box/circle the two different groups to help visualize the differences discussed in the text. -Many thanks for these detailed suggestions. We improved the caption, increased the size of the symbols, changed the pink and purple symbols and added an outline to all data points. We further removed the grid lines. The color scheme as it stands enables people with colour-blindness to see the differences. We therefore decided against using a color-tone gradient for grouping data together. We, however, circled the different groups to visualize the differences.

Figure 4 Same comments as Figure 3 with caption definitions, data point size, circles and triangles, and color tones. Also, is the variability of Cape Bounty discussed in the text? These figures should really tight groupings for Herschel but not Cape Bounty. -We revised the figure accordingly.

Figure 5 Similar comments to Figure 3 and 4. Same sites should use colors that fall in the same family with different gradients so they can be linked visually on the figures. Shades of green IC East on different sample dates will help. Keep acronyms similar among figures, e.g., IC East vs. ICE. These figures have a lot of gridlines on them which makes the dotted line hard to follow. Consider removing the gridlines or thickening the dotted lines. Also, consider using different symbols for different river samples. Define the terms in the caption. -Also in this case, the colour scheme was selected to make them accessible to people with colour blindness. Instead of changing the colours, we decided to change the symbology to group same sites together. We removed the gridlines as suggested and defined the terms in the caption.

Figure 6 The gridlines wash out the green data points and lines. Consider changing the color scheme and increasing the size of the connected lines and data points. Adding a vertical line through all figures for each rain event will improve these figures. Missing data should be notated in the caption. If all the data was collected in 2016, please remove the 2016 date indicator on the x-axis because it is very crowded. The legend also includes IC west and IC East. Please make this consistent with the other notations in the previous figures and define all the terms. -We removed the grid lines, defined

all terms, increased the size of the connecting lines, and added vertical lines at the beginning of each rainfall event. We also removed the "missing data" label and added this information to the caption.

Figure 7 Good figure. If it is needed still in the manuscript, since only two to three sentences discuss parts of it, then keep it with some improvements. Keep the reference on the figure but put the regions for the samples it is referring to in the figure caption and remove these words from the figure (it crowds the data). Define the terms in the figure caption. There are multiple data sets with the same color assigned to them. Please select different colors to see the different groups represented on this figure. Also, consider including the slope calculated from this work as a comparison to the literature calculated slope. -As this Figure is not needed anymore, we decided to move it to the Supplementary Material.

Figure 8 Good figure. If it is needed still in the manuscript, since only two to three sentences discuss parts of it, then keep it with some improvements. Define the terms in the caption. Is it possible to put black outlines around the Permafrost extent legend colors? The isolated patches color is very difficult to read in the legend. Also, mark the color of the ocean, since it is nearly identical to the isolated patches color, or change the ocean color to something darker? -As this Figure is not needed anymore, we decided to move it to the Supplementary Material.

Table 1 Define the term CAVM in the caption. Some formatting of this table is confusing like the dark thick line near the top and then a defining line combining ICW, ICE, and CB. Consider using indents for the sample names under the low and high Arctic categories, using another horizontal row divider (as in Table 2), or separate columns. -We edited the table and caption as suggested.

Table 2 Define the terms in the caption (all abbreviations and acronyms) and provide an explanation for underlining as a useful tool for these statistical comparisons. Typo at the bottom line "He" should read "HE" -We edited the table and caption as suggested.

Table 3 Define the terms in the caption. Supplemental Information Define the terms in the figure and table captions. -We edited the table and caption as suggested.

Consider moving S2 to the main manuscript. -We decided to leave it in the supplementary material as it only contributes to the discussion of limitations and not to the story itself.

––––––––––––––––––––––––––––––

---

## Author Comment (AC2) · 12 May 2019

We appreciate and are encouraged by the detailed feedback of Reviewer 2. We carefully revised and addressed the comments below. Our responses are given in a line by line list below.

Reviewer #2 General Comments:

In this study, Coch et al. undertake a comparative assessment of dissolved organic carbon and optical (as a350, SUVA, and spectral slopes) measurements. The authors use measurements from catchments on Herschel Island and Cape Bounty, using a

transect design to explore changes in DOC concentration and DOM character across regions and with movement downstream. As described below (and, as discussed by the authors) there are some issues with the optical data that appear to be still outstanding. As described in greater detail in the overarching comments, it would also be nice to see the authors more clearly elucidate how their study represents a step forward in DOM dynamics in sub-Arctic and high Arctic regions.

Overarching comments My most significant comment on the manuscript is the concerns related to Fe interference. Given the large scatter in SUVA and slope results for Cape Bounty, even across the stream sites that appear to not be affected by particularly long residence times, I think that this is an issue that must be dealt with before these data can be interpreted soundly. I don't have great confidence that the authors' approach of discarding samples that had evidence of flocculation was able to fully ameliorate this issue. Perhaps there is some residual sample that could be analyzed for Fe? A high level of confidence in the optical measurements is quite critical for the integrity of the manuscript.

-We agree with the reviewer that the confidence in the optical measurements is essential for this work. We unfortunately do not have any residual sample for additional measurements.

In comparison to the Herschel site, the Cape Bounty site indeed shows a larger range of values. As the reviewer correctly notes, this is not due to different residence times. We found that the range in SUVA and slopes at the sampling sites is due to the different nature of the sites themselves (e.g. influenced by permafrost degradation, pulse of rainfall delivering fresh DOM). We found different water types with different transparency, which regulate the photodegradation of cDOM. Thus, changes in absorption, SUVA and cDOM slope can be explained by catchment properties and/or rainfall events (see Figure 3). It might also be interesting to note that catchments at Herschel cover an area of 3 km2 in total, whereas the sampled area at Cape Bounty covered about 30 km2. This naturally results in a greater heterogeneity (and range) of optical parameters.

We looked again carefully on the raw absorption data from all samples to check for elevated absorption in long wavelengths which can be a result of high scattering by particles (in our case e.g. iron colloids). The Figure (attached, Fig. 1) shows the relationship between cDOM350 and DOC and the colors indicate the raw absorption at 700nm. Samples which we excluded from this study show high absorptions (>1 m-1) caused by particle scattering in the cuvette. This result supports the lab-notes which was used as a basis to exclude samples from this dataset. We are very confident that discarding samples based on flocculation notes actually did ameliorate the issue. To support this argument, we added a figure to the supplementary material showing DOC vs. acDOM350 for all included and excluded samples across the sites. At Cape Bounty many of the samples had SUVA values above 6, meaning that the cDOM values were too high for the low DOC concentrations. The maximum SUVA recorded in the excluded samples amounted to 59.5 L mg-1 m-1.

Furthermore, the relationship between cDOM350 and DOC of all included samples from both study sites are within the error range of other published samples from similar arctic aquatic environments (Fig. S3). If cDOM absorption data used in this study had been strongly interfered by iron colloids, the goodness regression of the relationship would be significantly lower.

A second high level comment is that I would like to see the authors do a better job of putting their work in the context of what has been done previously in the arena of DOC and cDOM in Arctic stream networks and elsewhere. At some points (see specific comments below) it seemed as if the text was focusing more on re-iterating previous findings, and less on carving out how the results from this study advance our knowledge. Ideally, a revised manuscript would have a much clearer emphasis on the latter.

-We followed the recommendations and revised the manuscript accordingly. Please

see detailed responses to the comments below.

Some editing for English grammar is also needed throughout the manuscript. I certainly have sympathy for non-native English speakers who are having to write in a second language! Perhaps some of the co-authors could assist with this sort of an edit.

-We edited the manuscript accordingly.

Figure quality could be improved, particularly for figures 3 – 6. Lake residence time: there is some discussion on effects of lakes vs. streams on cDOM in the two regions. Presumably photobleaching is more prevalent at Cape Bounty. Knowing something about the residence time (or, even rough volume / mean depth of these systems) early on in the manuscript would help greatly with this interpretation. My understanding is that the 'lakes' on Cape Bounty are relatively large: perhaps the Herschel systems have a very low residence time by comparison? I see that this information is provided towards the end of the paper, but it would be helpful to have it

-We added this information to the study area description.

Specific comments (as page / line number): 1/16, 2/13: What is small? The catchments being studied are very small indeed, for catchments discharging straight to the Arctic Ocean. It is not correct to state that direct export catchments of this size cover 40

-Unfortunately, there is no study available showing the actual size distribution of "small" catchments. In this case "small" means "smaller than the large Arctic rivers". We appreciate that the catchments studied here are not representative for all of the remaining 47% of the drainage area. To clarify this, we added that the actual size distribution remains unknow in the introduction.

1/20: you don't test for variation with SOCC and vegetation cover: "consistent with variation in vegetation cover and SOCC between the two sites"?

-We edited the sentence to "can be explained by differences in vegetation cover and SOCC…"

1/21: I would keep lignin out of the abstract, seeing as you don't measure this at all

-We edited the sentence according to the suggestion.

2/20-21: cDOM, or CDOM? I'm not a strong proponent for one vs. the other, but it would be good to make sure you're being consistent. We checked the manuscript for consistency.

-We changed it to cDOM in all cases.

2/32: The Spence et al. 2015 reference is incorrect here. Perhaps a mis-placement?

-This is correct. Thanks for noticing this misplacement.

3/15: add "cm" to specification of active layer depth.

-We added the unit.

3/17 and elsewhere: better specified as "C: N"

-We followed the recommendation of the reviewer.

4/5: Mean July temperature? A bit confusing if not specified.

-Edited for clarification.

4/6: "with baseflow re-establishing"?

-Good suggestion.

4/21: Manual outlet samples taken at what frequency?

-We added the frequency when specifying the manual sampling.

4/22: Can you clarify this sampling design? How many sampling points along this transect? Was there a pre-determined distance between points? Adding a reference to Fig. 1 would help here.

-We clarified the sampling design.

4/27: Herschel bottles were also triple-rinsed? It might be useful to start with a general sampling scheme at the top of this section. Also see comments above on clarifying the sampling scheme.

-We added a general sampling scheme introduction to the paragraph.

5/23: One technique to deal with particles is to subtract the average 700-800 nm base. This is a good practice for all samples, to correct for interference from colloids, etc., that might not be easily visible to the eye. See also 5/27 below.

-To our knowledge, subtracting the 700 nm base will have the same effect. We have done this as specified in the methods description.

5/27: This subtraction will correct for scatter (see above) but not for drift in instrument output across the range of wavelengths measured over hours of instrument use. The latter can be corrected for by measuring blanks (or, other standards) at specified time periods, and correcting measurements to this change.

-We measured blanks to monitor the drift of the instrument output. We clarified this in the methods description.

5/31: Were SUVA corrected for Fe? Substantial Fe could present challenges to your ability to interpret these results. Other studies have found fairly high Fe levels in the western Canadian Arctic, and Fe can also be one instigator of DOM flocculation. Some of your higher SUVA values do suggest possible interference from Fe. Reading on, I see you have some text on this below; see my later comments on this issue.

-Please find our comments below.

Section 3.2: Specify statistical packages used?

-We used the basic functions in R, so no packages need to be specified.

6/16: Here and elsewhere, please specify what your +/- values indicate. Standard error? 95

-It indicated mean +/- standard deviation. We indicated it in the methods section.

6/27: Difference in slopes is not significant, given your error bounds? In addition, you might find it useful to express your slopes as values x 103 .This is not uncommon in the literature, and might help with visualizing differences between sites, etc.

-We decided to express slopes as values x 10ˆ-3 as suggested. We changed it throughout the manuscript.

7/8-9: Separation into two groups: In Figure 3c, however, it looks like you only provide statistics for a single slope. Why not calculate both slopes, and test – statistically – whether they are different?

-We conducted a one-way ANCOVA (F-statistic and p) to test whether they are statistically different. We added "statistically" to the text.

7/13: This significant difference finding is interesting, because your error bounds overlap. It's difficult to assess this as a reader without knowing what's being presented as a metric for dispersion in the data. I'm not sure I would analyze the data in this way given that (from Figure 5) your slope values cover a similar (wide) range at each of the two sites. You also have substantially different n for your two sites, which could confound your statistical analyses.

-We agree that the current presentation of the data does not make sense in this way. We put the emphasis rather on the wider range that is covered by Cape Bounty samples in comparison to Herschel.

Section 4.4 / Figure 6: What about using C-Q (i.e., hysteresis) plots to illustrate these responses. I think this would help elucidate better what is happening across the three events. For some of these events, I'm not sure I see much of a response, or at least – it's a bit difficult to tease out with the current presentation.

-We have tried presenting the data as hysteresis in an earlier version. Unfortunately, the sampling frequency in Ice Creek East is not high enough to detect a response. We

edited the figure for improved readability.

Section 5.1 / first paragraph: Again, this makes me concerned about interference from Fe.

-Please see above.

Page 9 / line 25: Ah – yes! I see you get to Fe here. At the bottom of this paragraph (8/10): I'm really not sure you have eliminated all of the problematic samples. If you have extra water from these sites, it would be great to have this analyzed for Fe, if you haven't already. It seems you are using precipitation as a proxy for Fe interference? You may certainly have high Fe in non-precipitate samples. Particularly given the scatter in your data for a series of samples taken upstream of a lake (e.g., West River vs. East River in Figure 4c), I think you need to be concerned about whether some of the patterns you're observing are 'real'. For the Herschel Island samples, where it appears that Fe data are available, it would be great to correct for this.

-Please see above.

5.2.1, first paragraph: I agree that you see higher DOM quantity at Herschel. From Figure 4 and 5, however, I don't think you can conclude there's any difference in quality. The quality values span across a similar range at both sites, with values from Cape Bounty showing a wider range. However how this paragraph is written whether this analysis is contributing anything new. If there is something novel that's being presented in this particular ms, it would be good to structure the text in a way that really highlights that fact.

-We changed the first paragraph accordingly.

Section 5.2.2, third paragraph: it seems to me that this is perhaps the novel information that's being presented in this section; the second paragraph, as written, also seems to largely summarize findings from previous studies. Why not flesh this out a bit and focus here? For example – what is the evidence for increased permafrost DOM export

with increasing baseflow; I may have missed this discussion above, but it would be nice to have this laid out very clearly. It would also be useful to cite the Spence et al. publication that seemed to be mis-placed above in this section.

-The section above leads to this conclusion. We put more emphasis on this section and also incorporated the studies by Spence et al.

12/23: See also the conceptual work on headwater to mainstem gradients by Drake et al.

-Thanks for pointing this out.

12/25: IE – temporal variation? Within sampling dates, the SUVA and slopes are fairly consistent along the transect; the values are certainly much more variable across time than across space.

-This paragraph focuses on the upstream to downstream patterns. However, it is correct that the temporal variation was not sufficiently discussed. We added this into the "rainfall events" discussion.

14/4: cDOM as a good proxy for DOC concentration. This is true (and quite well established) in cases where most DOM is terrestrial in origin, and not overly degraded. There are a few references you could cite here for studies that have made this point using pretty extensive datasets. It's not necessarily true universally, however; there's also some good papers showing a lack of relationship between DOC and colour for sites where DOM is highly reworked / photobleached – see for example work by Arts et al., Osburn et al. and others on prairie / great plains lakes.

-Thanks for pointing this out. We added the reference.

14/7: What about the relationship between CDOM and soil organic carbon content in the catchment of study? Presumably there is a strong relationship here (see, for example Connolly et al. 2018, ERL). It seems likely that climate / latitude is a controlling variable in the sense that it has such an important influence on soils. Ah – I see you

have this in the paragraph below. It would be good to look at the recent work by Connoly et al, who also examine this relationship across a variety of watershed sizes.

-This is indeed a very interesting study, especially with regard to upscaling results to the circumarctic region. Since they linked both, SOCC and DOC to slope, and not directly to each other, we decided not using the reference here.

———————————————

[Figure]

**Fig. 1.** Relationship between aCDOM350 and DOC concentration for all samples (included and excluded). Absorption at 700nm is color coded.

---

## Author Comment (AC3) · 12 May 2019

Please find the tracked changes manuscript attached.

Please also note the supplement to this comment:
https://www.biogeosciences-discuss.net/bg-2019-9/bg-2019-9-AC3-supplement.pdf

---

## Referee Report (RR1)

While this manuscript is greatly improved from the first iteration, it still requires considerable changes to support the main message of the work. In my opinion, the authors have not presented a cohesive message throughout the work and would benefit from more time to revise and clarify the points being suggested. If there was another step in between major and minor revisions, that would be the best suggestion for this work. Certainly, it has improved, but requires some major reorganizing and some minor tweaks. The best advice I can provide to the authors is to use the introduction to set the stage for the novelty of the work (which did improve) and use that platform throughout to tell a cohesive story. Reading this work generated three main themes, small catchments are important, standing and flowing water is important, and comparisons between high arctic and low arctic environments are important. This is in light of the disturbance regimes being tested from the perspective of C quality. Clarify the main theme, support your theme with strong topical statements at the beginning of the results and discussion sections. The results section still requires considerable work to support the main theme of this research project. The data is there, even some data that was eliminated. Report and discuss it all in a way that strengthens your argument for the importance of small catchments AND low to high arctic comparisons. A good deal of explanation is provided for the removal of the data that had precipitates, but I was left wondering whether or not that was one of the defining differences between high arctic and low arctic environments. Would it make sense to have those different C signals in high arctic environments comparatively after the environment was disturbed? Can we consider reporting that data to prepare other researchers for data that might not be outliers for future work? Were any considerations of contamination discussed? Some of that data seemed to be the most interesting and certainly showed the largest differences between low and high arctic environments. Good improvements so far. Consider these revisions to strengthen the work even further.

Page 1 Line 12: "Climate change is an important control of carbon cycling" reads funny. The first line of the introduction reads much smoother. Consider this edit: Climate change is affecting the rate of carbon cycling, particularly in the Arctic.

Page 5 Line 11: Typo "where" should be "were"

Page 5 Line 29: What does "This was noted down in the lab" mean? Please edit to "This information was documented…" And maybe something that says – these samples were removed from analyses?

Page 7 Line 16: Please add Fig. 3c to the end of this sentence and remove it from the next one. Also, how is this claimed when there are circles and triangles circled together in these groups on Figure 3c? This is very confusing.

Page 7 Line 17: DOC and SUVA data is on a different figure? This sentence seems out of place if it's referring to the next figure and the next thought is back to A350.

Page 7 Line 17-18: Wasn't it just stated that there was a difference between these two water types?

Page 7 Line 23-24: "The headwaters in both rivers showed slightly smaller slopes than the samples taken downstream." What does smaller slopes mean? Which rivers? Is this referring to Fig. 4b? Maybe clarify the use of "slopes" and "S275-295" notation or use "slope values" or "S". A reader might be looking for the slope of the relationship.

Page 7 Line 25-26: "Standing water samples showed significantly larger slopes (p < 0.05) and significantly smaller SUVA (p < 0.05) than flowing water samples." Yes, but I had to look for it, since there are low and high values for standing water. It seems like the standing water samples have data across broad SUVA and S ranges. Perhaps this should report its broad nature, because the text doesn't match the figure very well. Flowing water looks like a broad distribution too, but less so than standing water.

Page 7 Line 29-30: How can this be stated when it clearly increases between 1500 and 1000 m from the outflow? And again for ICW 30 Jul near the outflow? Consider language like "generally decreased" and edit the sentence accordingly. Maybe report the percent of decrease if that is important?

Page 7 Line 30-32: This is true of most of your other time points. Why is it important to highlight this point for just these two?

Page 7-8 Line 32 and 1: Correct, but it contradicts earlier text. Please correct the earlier statement and move this sentence earlier to improve cohesive reporting.

Page 8 Line 1-2: Please add the date in the text so it matches the figure.

Page 8 Line 6: Add Fig. 5b at the end of this sentence.

Page 8 Line 8: Revise the wording in this sentence to remove "remained with" and insert "had"

Page 8 Line 10-11: This information is not evident by looking at the figure. Can some of this information about between rainfall and post rainfall conditions be made evident?

Page 8 Line 11: This section compares a lot of trends back to DOC concentration. Is that important? Or can a general comment be stated more succinctly about DOC and optical properties generally decreasing longitudinally? Certainly, the values of SUVA for Herschel overlap for ICE and ICW for the first time showing similar character? Would it be helpful to have the data discussed in terms of dates? In general, will any values of AcDOM350 and SUVA be discussed to understand what this character might mean or is it just about reporting increases and decreases?

Page 8 Line 14: This kind of language "values were variable…" could be said for all your measured data points. What is the most important thing to report about the S values? An increase near headwaters and then…? This section should be setting the stage for why this information best informs us about this catchment.

Page 8 Line 15: Add the date in for the first rainfall event in the text to match the figure. It will improve clarity. The remaining part of the sentence is clear and is the first mention of seasonal relationships. Consider this type of language throughout this section.

Page 8 Line 16: Why is this important? This sentence structure "We found lows…" is poor scientific writing. Consider editing this sentence to: "The lowest S values were reported for…"

Page 8: There is a lot going on in this manuscript – different catchments (east and west) and seasonal aspects tracked longitudinally, as well as low and high arctic catchment comparisons. If the main message of the manuscript is to include a never before low to high arctic comparison, then a strong point can be made about the differences of each – individually in their catchments (Do east and west really have different influences and therefore different character? And do different rivers in the high arctic behave similarly?) – and then also as a comparison on low arctic and high arctic scales. The DOC concentration, A350, and EC are all quite different when comparing low and high arctic catchments. The other measured variables are not. Some reporting on this would strengthen the message of the work.

Page 8 Line 25: Please add a comma after composition

Page 8 Line 26: Please use rivers in this sentence for consistency. Delete streams, unless they are streams. Same comment again – or just edit for consistency in Lines 28 and 29.

Page 8 Line 30: Please add a comma after 350

Page 8 Line 31: This dynamic was not captured in ICE? It looks like the same trends are there in the figure. Sure, it was sampled at a longer time interval, but some of those trends seem reasonable, just not as highly resolved as ICW.

Page 8 Line 31: Please include a result of the EC data after rain event 2.

Page 8 Line 32: Delete the word "had". "Baseflow increased after this rainfall event (Fig. 6a)."

Page 9 Line 2: Please add a comma after 350

Page 9 Line 2-3: "SUVA increases with two spikes in the data." An example of poor scientific writing. What is important about this result? Please consider revising this to complement the work accomplished, such as, "SUVA increased sharply on August X and Y, describing a shift in DOM composition, followed by a general decreasing trend until August Z." That way, your readers will associate your measurements change after rainfall events and what's important about the disturbance of C in your system.

Page 9 Line 3: Please use stronger scientific language. "..a drop in SUVA" can be edited to "decreased SUVA values". This section should be edited for consistent tense, i.e. past or present.

Page 9 Line 4-5: The scale captures the overall variability in the data for a reader, but can it be stated which direction the data went in the gaps? Those two gaps are right after a rainfall event?

Shouldn't those trends be reported as well? Increased S or decreased S values? Consider describing that information in the text and putting the full scale of those points in the caption, so the figure doesn't eliminate any information completely.

Page 9 Line 12: Fluctuating between AcDOM and AcDOM350. Please check.

Page 9 Line 14: These aren't realistic for natural surface waters, so what could it have been? Could it have been related to disturbed permafrost? Or some kind of contamination? Does Cape Bounty represent something unique about the high Arctic? This is very interesting and I'm curious as to why secondary filtration wasn't attempted? It is still a great deal of samples removed from the data set – 25 out of 55! What would have happened if they were incorporated into the study, but marked appropriately?

Page 9 Line 19 and Figure 7: Suggest plotting SUVA next to AcDOM350 to add to this figure.

Page 10 Line 1: Subheading suggestion: Nature of the DOM concentration and composition relationship across the terrestrial Arctic OR just add the word concentration into the title

Page 10 Line 2: Delete "as found" in this sentence.

Page 10 Line 7-9: Revise for stronger wording: "However, this relationship is not always strong for ecosystems where DOM is strongly altered…" Here's a question: Can't a photodegradation argument be made for your sites during the summer?

Page 10 Line 10: Is this insinuating that the DOM you are tracking may be directly a result of leaching or disturbance from 0-30cm or 100cm depths? Are these permafrost links?

Page 10 Line 26: Is fresh (less altered) referring to less microbially and photochemically altered? So freshly produced? Higher aromaticity = fresh material? And prone to degradation? Higher aromatic freshly produced material – is coming from what? And prone to what kind of degradation? This is an interesting point and should be clarified.

Page 10 Line 28-29: A great point to make about sampling smaller catchments and describing their impact in a changing Arctic climate. This point should be made up front and supported throughout.

Page 11 Line 1 and section: This section seems to be more important up front before the current 5.1 and 5.2 sections. Consider reorganizing the order of these discussion points.

Page 11 Line 11: Fresh DOM is high SUVA? An explanation should be discussed here. Is this freshly produced? Or freshly released? And the next sentence describes fresh as low autochthonous production. Fresh as in – newly introduced?

Page 11 Line 16-20: Check with figure. Both symbols are circled in these groups and a clear relationship is not apparent.

Page 11 Line 22: Please add a comma after "intensity"

Page 11 Line 27: What kind of degradation?

Page 11 Line 28: Deeper in the active layer? Of what? Soil? Permafrost? These are important points to continue to tie together throughout the manuscript. And it suggests that rainfall mobilizes different types of C.

Page 11 Line 30: Please add the word concentration after DOC

Page 11 Line 31-32: Indicative of more decomposed material – meaning the mobilization of more decomposed material from???

Page 12 Line 4: What isotope?

Page 12 Line 6-7: Delete "after the first one" and add "later". Also, what's important about this timing to the DOM story?

Page 12 Line 9-10: Was that trend reflected in your data?

Page 12 Line 16: Please add the word concentration after DOC and delete the word "there".

Page 12 Line 18: The definition of fresh seems to be changing. Consider a usage of it to indicate mobilization.

Page 13 Line 1: Provide a definition of labile here to improve clarity.

Page 13 Line1-2: You showed that stormflow alters flow pathways? This seems like an overstatement.

Page 13 Line 5: What kind of degradation?

Section 5.3.2: Much improved. Again, the whole 5.3 section should be 5.1 and reordered.

Page 13 Line 14: What kind of degradability?

Page 13 Line 18-19: The objective was therefore to… doesn't make any sense. Can you elaborate here? Our objective was therefore to investigate the upstream to downstream patterns in smaller coastal catchments to understand…? Tie in the information from the previous sentences.

Page 13 Line 20: Please add a comma after SUVA

Page 13 Line 21: Please add a comma after 295

Page 13 Line 20-21: So what does that mean? What kinds of C are coming in, transforming, whatever?

Page 13 Line 23: This information of the tributary is really important. Add it to the figure.

Page 13 Line 33: This sentence looks cut off. What's the point to be made with these ideas?

Page 14 Line 2: Add the word concentration after DOC

Page 14 Line 5: Use the past tense here.

Page 14 Line 15: Consider ending this section with comparative low and high arctic themes and impact.

Page 14 Line 18: A note on the strength of the message. Permafrost disturbance is continuously happening but exacerbated with rainfall events? Are they connected or not? That message might be good up front and then supported here; it is lost throughout the text.

Page 14 Line 22: And C quality, right?

Page 14 Line 26: Please add a comma after 295

Page 14 Line27: How can your measurements describe C prone to degradation?

Page 14 Line 28: Streams or rivers? And why is this useful tool for assessing downstream patters important to your study? Drive the message home?

Page 15 Line 12: Please add a comma after H.L.

Page 15 Line 13 and 14: Typo. Analyzes should be analyses.
Page 15 Line 14: Please delete the comma after analyses and add the word and. Also, correct the word visualized to interpreted.

Page 15 Line 15: Please add a comma after S.L.

Page 15 and 16: The authors fluctuate with usage of lab and laboratory. Please correct the usage of lab to laboratory where appropriate.

Figure 1: Greatly improved. Are the subzones discussed in the text? Is it important in this figure? Please add a comma after West River in Line 6 Figure 1 caption.

Figure 2: What would CB's data look like if the eliminated samples were added to this figure? A few times in this manuscript the words absorbance and absorption are interchanged. Consider using only one version of this word.

Figure 3: In (b) it is difficult to see the data behind the blue dots of ICW downstream. The gray and orange data points are covered. Can they be overlaid for easier visualization? Define what the two groups are in (c) in the caption. As mentioned earlier, these two groups encompass both circles and triangles so what is special about these groups if not water type?

Figure 4: It might be worth noting that SUVA is calculated at A254nm. Please add a comma after the word downstream in Line 4. Use consistent language for water type – circles and triangles. In this caption, please edit dot to circle. In (b) it is difficult to see the data behind ICW downstream, similar to Figure 3.

Figure 5: Might it be helpful to mark tributaries on the Herschel Island figure? Didn't that feature change some of the measurements?

Figure 6: Annotate which direction the data goes for the gaps in S in (e) or describe the 6x increase or whatever amount of increase or decrease in the caption.

Figure 7: Delete the word "to" in Line 4 before the word "high". This part of the data still gives pause, since other values are around 40, but with higher DOC concentrations. Can these really be excluded? Consider adding SUVA as a second panel to this figure.

Table 1: Soil organic carbon content in the table can be abbreviated to SOCC. This table is a nice tie in with Figure 1, with annotations regarding the subzones. Can more of that be incorporated into the text so that the reader is reminded that the different rivers correspond to different vegetation types?

Table 2: Add a comma after pH in Line 3 and standing waters in Line 4. Are the significant differences highlighted in the table discussed in the main text?

---

## Author Response (AR2)

**Author responses to reviews and edits to Biogeosciences manuscript bg-2019-9 "Characterizing organic matter composition in small Low and High Arctic catchments using terrestrial colored dissolved organic matter (cDOM)".**

**Response to Associate Editor report from 18.07.2019**

Associate Editor comments and our responses are presented below.
Associate Editor comments are given in *italic font*, our response in green regular and the resulting change in the manuscript in *green italic*.

We are grateful for the detailed and constructive comments on our manuscript. We agree to the suggestions and comments and are confident that the review has contributed to improve the paper during our revisions.

**General Comments:**

*"While this manuscript is greatly improved from the first iteration, it still requires considerable changes to support the main message of the work. In my opinion, the authors have not presented a cohesive message throughout the work and would benefit from more time to revise and clarify the points being suggested. If there was another step in between major and minor revisions, that would be the best suggestion for this work. Certainly, it has improved, but requires some major reorganizing and some minor tweaks. The best advice I can provide to the authors is to use the introduction to set the stage for the novelty of the work (which did improve) and use that platform throughout to tell a cohesive story. Reading this work generated three main themes, small catchments are important, standing and flowing water is important, and comparisons between high arctic and low arctic environments are important. This is in light of the disturbance regimes being tested from the perspective of C quality. Clarify the main theme, support your theme with strong topical statements at the beginning of the results and discussion sections."*

Thank you for the advices on creating a more cohesive story throughout the manuscript. We adapted the objectives section in the introduction and used this to restructure the result and discussion section based on those objectives. We hope that this brought a clearer "red-line" into our manuscript.
We adapted the objectives (1), (2), (3) and adjusted results and discussion sections.
The objectives are now clearly separated into three main themes of the story:

*"The aim of this study is to (1) compare the variability and relation of DOC concentration and cDOM in High and Low Arctic surface water environments, (2) to investigate changes in DOM composition along longitudinal stream profiles with regard to permafrost disturbances and (3) examine changes in DOM concentration and composition throughout the summer season with occasional rainfall events."*

We changed the results and discussion sections structure to:

*"4. Results*

> *4.1 DOM characteristics and relationships in and across Low Arctic (Herschel Island) and High Arctic (Cape Bounty) catchments*

> *4.2 Hydrochemical and DOM patterns along longitudinal stream transects*

*4.3. Temporal changes of DOM under different meteorological conditions*

*5. Discussion*

*5.1. Catchment processes and DOM alteration*

> *5.1.1 Regional catchment properties of DOM*

> *5.1.2 Downstream patterns of DOM and impact of permafrost disturbance*

*5.1.3 Rainfall event impacts on DOM*

*5.2. DOM dynamics of small and large Arctic catchments*

*5.3. Limitations of cDOM measurements from terrestrial sources"*

Additionally, we extended Figure 2 showing differences in flowing and standing waters in each study area. The text in the results is adapted accordingly.

*"The results section still requires considerable work to support the main theme of this research project. The data is there, even some data that was eliminated. Report and discuss it all in a way that strengthens your argument for the importance of small catchments AND low to high arctic comparisons. A good deal of explanation is provided for the removal of the data that had precipitates, but I was left wondering whether or not that was one of the defining differences between high arctic and low arctic environments.*
We added a paragraph on potential differences between Herschel Island (Low Arctic) and Cape Bounty (High Arctic):

*"Poulin et al. (2014) also showed that in samples with low pH the dominant fraction of iron is Fe(II) which then potentially can precipitate as Fe(III) with increasing pH during transport and storage. Cape Bounty samples which showed a substantially lower pH, likely caused by low vegetation, are therefore more prone to precipitate Fe(III) colloids which affect the optical absorption measurements and lead to the high absorption values at 700 nm (Fig. S4). Herschel Island samples originally already had a higher pH compared to Cape Bounty. Thus, we expect that the dominant fraction of iron on Herschel Island was Fe(III) which leads to a lower potential of Fe(III) precipitation compared to Cape Bounty."*

*"Would it make sense to have those different C signals in high arctic environments comparatively after the environment was disturbed? Can we consider reporting that data to prepare other researchers for data that might not be outliers for future work? Were any considerations of contamination discussed? Some of that data seemed to be the most interesting and certainly showed the largest differences between low and high arctic environments. "*
We discuss this topic in point 5.3 - Limitations of cDOM measurements from terrestrial sources. The data was reported in Fig. 7. The data circled in Fig. 7 have disproportionality high absorption values in relation to the DOC concentration. High CDOM is always accompanied by high DOC concentrations. Surface waters can have high DOC concentration and at the same time low CDOM in case the CDOM is highly bleached. Surface waters cannot have high CDOM and low DOC concentration at the same time, as CDOM is always a part of DOC.
The cuvette measurement is an optical measurement of transmission of the water sample. The optical transmission is reduced by absorption and by scattering of water and the specific dissolved and particulate water constituents of the sample. The protocol for CDOM measurements assumes that all attenuation is due to absorption and not due to scattering on particles because samples are filtrates. If the colloid size in the samples is changing as it seems to be the case for our samples due to pH instability, and even form precipitates there is a higher attenuation for this sample due to more scattering sources in the sample. This additional scattering by newly formed colloids and precipitates leads to lower transmission measurement that is calculated as very high Optical Density = absorbance of the sample that is than calculated towards high CDOM absorption. The high absorption in the long wavelengths are a clear indication for particle contamination in the excluded samples because cDOM does not absorb at these longer wavelengths. High absorption values at longer wavelengths are caused by this additional scattering. There is a standard procedure to subtract the onset of absorption at NIR wavelengths from all samples, but usually these group values are of very low magnitudes of absorption coefficient only and are just some noise in the data. In case of the contaminated samples, absorption in the NIR wavelengths was also exceptionally high, that gave another indicator for removal of samples. Thus, these samples are not natural outliers, but mainly occurred due to changes in the scattering properties in the samples.

We show the value range of low DOC < 5 mg/l and high $a_{CDOM}350$ in figure 7 to prepare other researchers for such type of data. We also highlight that there is a difference in pH of surface waters from Cape Bounty compared to Herschel Island and refer to the lithology of the catchment: last text paragraph in 5.3. Limitations of cDOM measurements from terrestrial sources: *"Catchment properties that influence riverine pH such as the local lithology may play an important role. In case of alpine and high Arctic catchments with thin or no soil cover, a bed rock composition of acid rocks in the catchments will lead to lower pH values in surface waters such as it is the case for Cape Bounty. Whereas surface waters from Herschel Island catchments on glacial moraines and marine sediments are characterized by higher alkalinity."*

*"Good improvements so far. Consider these revisions to strengthen the work even further."*
Thank you for the helpful general comments about our manuscript. We used these to restructure our manuscript and extended unclear/missing sections which were requested in the review.
Below we comment on specific comments

Page 1 Line 12: *"Climate change is an important control of carbon cycling" reads funny. The first line of the introduction reads much smoother. Consider this edit: Climate change is affecting the rate of carbon cycling, particularly in the Arctic. "*
The sentence was change according to the suggestion:
*"Climate change is affecting the rate of carbon cycling, particularly in the Arctic"*

Page 5 Line 11: *"Typo "where" should be "were" "*
Thanks for noticing. We changed this accordingly.

Page 5 Line 29: *"What does "This was noted down in the lab" mean? Please edit to "This information was documented…" And maybe something that says – these samples were removed from analyses? "*
We edited the sentence for clarity.

Page 7 Line 16: *"Please add Fig. 3c to the end of this sentence and remove it from the next one. Also, how is this claimed when there are circles and triangles circled together in these groups on Figure 3c? This is very confusing. "*
We moved Fig. 3c. Generally, the low number of samples in Cape Bounty makes a clear interpretation difficult. However, the groups we describe in this section show trends where the majority of flowing waters forming one group. Outliers, as the triangle in the group of flowing water, can occur in nature. It might have happened that this sample is not representative for the mid-stream water but is influenced by a very local source of DOM.
We adapted the text accordingly:
*"Whereas on Herschel Island, the relationship between acDOM350 and DOC follows one linear trend (rho = 0.72, p < 0.05) the relationship at Cape Bounty is broadly separated into two groups, namely flowing and standing water. Correlations for both groups are significant (< 0.05) and show different slopes of the regression. One sample identified as standing water falls into the group of flowing water. We identify this sample as an outlier that may was affected by very local DOM sources."*

Page 7 Line 17: *"DOC and SUVA data is on a different figure? This sentence seems out of place if it's referring to the next figure and the next thought is back to A350. "*
This refers to Table 2. It is added to the text for clarification.

Page 7 Line 17-18: *"Wasn't it just stated that there was a difference between these two water types? "*
In these lines, we reported no significant difference between the mean value for both water types (standing and flowing water). However, the mean value might be misleading since e.g. for Cape Bounty, the number of samples in low concentration range is substantially higher than in the high concentration range, thus leading to an overall low mean. We added a sub-boxplot presenting DOC, CDOM, SUVA and S275-295 values for standing versus flowing waters. Reporting only the mean

and the standard deviation (table 2) is likely not representative for the low number of samples (especially for Cape Bounty). We changed the text accordingly:

*"Mean S275-295 on Herschel Island is generally higher (16.4 ± 1.5 x 10-1 nm-1) compared to Cape Bounty (14.8 ± 3.2 x 10-1 nm-1), whereas SUVA values show a broader range on Cape Bounty (from 1.35 to 5.16 mg $L^{-1}$ $m^{-1}$) compared to Herschel Island (from 2.0 to 4.3 mg $L^{-1}$ $m^{-1}$) (Table 2)"*

Page 7 Line 23-24: *""The headwaters in both rivers showed slightly smaller slopes than the samples taken downstream." What does smaller slopes mean? Which rivers? Is this referring to Fig. 4b? Maybe clarify the use of "slopes" and "S275-295" notation or use "slope values" or "S". A reader might be looking for the slope of the relationship. "*

We removed this sentence since changes from headwater to downstream are described in the following section. We standardized the use of "spectral slope" or "S275-295" throughout the whole manuscript.

Page 7 Line 25-26: *"""Standing water samples showed significantly larger slopes (p < 0.05) and significantly smaller SUVA (p < 0.05) than flowing water samples." Yes, but I had to look for it, since there are low and high values for standing water. It seems like the standing water samples have data across broad SUVA and S ranges. Perhaps this should report its broad nature, because the text doesn't match the figure very well. Flowing water looks like a broad distribution too, but less so than standing water. "*

We changed this paragraph, see comment above.

Page 7 Line 29-30: *"How can this be stated when it clearly increases between 1500 and 1000 m from the outflow? And again for ICW 30 Jul near the outflow? Consider language like "generally decreased" and edit the sentence accordingly. Maybe report the percent of decrease if that is important? "*

Sentence was edited to:

*"At Herschel Island, overall, DOC concentration and $a_{cDOM}$350 (Fig. 5a) decreased downstream at all sampling periods. However, we observed a stronger decrease in Ice Creek West compared to Ice Creek East. Ice Creek West shows an increase in DOC and $a_{cDOM}$350 concentration at ~1300 m, where a tributary joins the main stem."*

Page 7 Line 30-32: *"This is true of most of your other time points. Why is it important to highlight this point for just these two? "*

See above extension of the sentence to clarify the importance of highlighting these two points.

Page 7-8 Line 32 and 1: *"Correct, but it contradicts earlier text. Please correct the earlier statement and move this sentence earlier to improve cohesive reporting. "*

We changed this accordingly.

Page 8 Line 1-2: *"Please add the date in the text so it matches the figure. "*

We changed this accordingly.

Page 8 Line 6: *"Add Fig. 5b at the end of this sentence".*

We changed this accordingly.

Page 8 Line 8: *"Revise the wording in this sentence to remove "remained with" and insert "had""*

We revised this accordingly.

Page 8 Line 10-11: *"This information is not evident by looking at the figure. Can some of this information about between rainfall and post rainfall conditions be made evident? "*

We tried different approaches of changing the figure. Displaying the onset of the rainfall event by vertical blue lines proved to be the best option. No further changes were made.

Page 8 Line 11: *"This section compares a lot of trends back to DOC concentration. Is that important? Or can a general comment be stated more succinctly about DOC and optical properties generally decreasing longitudinally? Certainly, the values of SUVA for Herschel overlap for ICE and ICW for the first time showing similar character? Would it be helpful to have the data discussed in terms of dates? In general, will any values of AcDOM350 and SUVA be discussed to understand what this character might mean or is it just about reporting increases and decreases? "*

We restructured this section. We mostly combined DOC and aCDOM350 for the descriptive part of the results.

Furthermore, we added a section to the discussion where we discuss the high variability and ranges of DOC and aCDOM350 in small Arctic catchments. Thus, we believe it is an important result which needs to be mentioned in the manuscript.

aCDOM350 and SUVA values are discussed in section 5.1.2

Page 8 Line 14: *"This kind of language "values were variable…" could be said for all your measured data points. What is the most important thing to report about the S values? An increase near headwaters and then…? This section should be setting the stage for why this information best informs us about this catchment. "*

We changed and extended this paragraph accordingly:

*"Spectral slope values (S275-295) at Herschel Island showed an increase downstream (Fig. 5d). When sampled on the same day, Ice Creek West showed only slightly smaller spectral slope values along the stream profile compared to Ice Creek East on 30 July 2016. Significant differences were observed between different sampling periods in the Ice Cree West. They were smallest after the first rainfall event and increase progressively over the course of the season. The rivers on Cape Bounty showed highest spectral slopes for East River (16.1 ± 1.6 x 10-3 nm-1) and the lowest for West River (11.9 ± 0.8 x 10-3 nm-1A slight downstream increase in spectral slope was recorded in West River."*

Page 8 Line 15: *"Add the date in for the first rainfall event in the text to match the figure. It will improve clarity. The remaining part of the sentence is clear and is the first mention of seasonal relationships. Consider this type of language throughout this section. "*

Dates were added to text and figure.

Page 8 Line 16: *"Why is this important? This sentence structure "We found lows…" is poor scientific writing. Consider editing this sentence to: "The lowest S values were reported for…" "*

See change of the paragraph above

Page 8: There is a lot going on in this manuscript – different catchments (east and west) and seasonal aspects tracked longitudinally, as well as low and high arctic catchment comparisons. If the main message of the manuscript is to include a never before low to high arctic comparison, then a strong point can be made about the differences of each – individually in their catchments (Do east and west really have different influences and therefore different character? And do different rivers in the high arctic behave similarly?) – and then also as a comparison on low arctic and high arctic scales. The DOC concentration, A350, and EC are all quite different when comparing low and high arctic catchments. The other measured variables are not. Some reporting on this would strengthen the message of the work.

We restructured the results sections to three main sections according to the objectives stated in the objectives:

1) DOM characteristics and relationships across low Arctic (Herschel Island) and high Arctic (Cape Bounty)
2) Hydrochemical and DOM patterns and modifications along longitudinal river transects
3) Temporal trends of DOM with changing meteorological conditions

We hope that this structure helps transfer a synthetic story.

Similar or different behaviors of rivers in and across high and low Arctic are discussion in sections 5.1.2. and 5.1.3.

Generally, results and discussion have been restructured as mentions above in this document.

Page 8 Line 25: *"Please add a comma after composition "*
Added

Page 8 Line 26: *"Please use rivers in this sentence for consistency. Delete streams, unless they are streams. Same comment again – or just edit for consistency in Lines 28 and 29. "*
We changed this accordingly.

Page 8 Line 30: *"Please add a comma after 350 "*
changed

Page 8 Line 31: *"This dynamic was not captured in ICE? It looks like the same trends are there in the figure. Sure, it was sampled at a longer time interval, but some of those trends seem reasonable, just not as highly resolved as ICW. "*
We agree and changed the text to:
*"This dynamic was only captured to some extent in Ice Creek East, which was sampled at a longer time interval. Baseflow had increased after this rainfall event (Fig. 6a)."*

Page 8 Line 31: *"Please include a result of the EC data after rain event 2. "*
Description added.

Page 8 Line 32: *"Delete the word "had". "Baseflow increased after this rainfall event (Fig. 6a).""*
done

Page 9 Line 2: *"Please add a comma after 350 "*
Changed

Page 9 Line 2-3: *""SUVA increases with two spikes in the data." An example of poor scientific writing. What is important about this result? Please consider revising this to complement the work accomplished, such as, "SUVA increased sharply on August X and Y, describing a shift in DOM composition, followed by a general decreasing trend until August Z." That way, your readers will associate your measurements change after rainfall events and what's important about the disturbance of C in your system. "*
We changed the text accordingly. However, we don't intend to mix discussion and results since "shift in DOM composition" would rather more belong to the discussion section

Page 9 Line 3: *"Please use stronger scientific language. "..a drop in SUVA" can be edited to "decreased SUVA values". This section should be edited for consistent tense, i.e. past or present."*
We changed the sentence accordingly and edited the tenses.

Page 9 Line 4-5: *"The scale captures the overall variability in the data for a reader, but can it be stated which direction the data went in the gaps? Those two gaps are right after a rainfall event? Shouldn't those trends be reported as well? Increased S or decreased S values? Consider describing that information in the text and putting the full scale of those points in the caption, so the figure doesn't eliminate any information completely. "*
We adjusted Figure 2 to show all data including the outliers. The text is also edited for clarification.

Page 9 Line 12: *"Fluctuating between AcDOM and AcDOM350. Please check. "*
We changed the text accordingly

Page 9 Line 14: *"These aren't realistic for natural surface waters, so what could it have been? Could it have been related to disturbed permafrost? Or some kind of contamination? Does Cape Bounty represent something unique about the high Arctic? This is very interesting and I'm curious as to why secondary filtration wasn't attempted? It is still a great deal of samples removed from the data set –*

*25 out of 55! What would have happened if they were incorporated into the study, but marked appropriately? "*
Please see explanation following line 22. High CDOM is always accompanied by high DOC concentrations. Surface waters can have high DOC concentration and at the same time low CDOM in case the CDOM is highly bleached. Surface waters cannot have high CDOM and low DOC concentration at the same time, as CDOM is always a part of DOC.
The cuvette measurement is an optical measurement of transmission of the water sample. The optical transmission is reduced by absorption and by scattering of water and the specific dissolved and particulate water constituents of the sample. The protocol for CDOM measurements assumes that all attenuation (attenuation=absorption + scattering) is due to absorption and not due to scattering on particles because the samples are technically filtrates. If the colloid size in the samples is enlarging as it seems to be the case for our samples due to pH instability, and even form precipitates there is a higher attenuation for this sample due to more scattering sources in the sample. This additional scattering by newly formed colloids and precipitates leads to the lower transmission measurement that is calculated as absorbance. The high absorption in the long wavelengths are a clear indication for particle contamination in the excluded samples because of additional scattering.

Page 9 Line 19 and Figure 7: *"Suggest plotting SUVA next to AcDOM350 to add to this figure. "*
SUVA, as described in the methods, is a parameter which is the DOC normalized absorbance, thus it is directly dependent on the absorption/ absorbance. The Absorbance is unitless [0-1]. The absorption is the Absorbance normalized with the length of the cuvette used (the optical path length L), absorption = absorbance / L with the most common units [m-1] or [cm-1] and used as Decadal absorption and Naperian absorption (Naperian absorption = Decadal absorption * 2.303). Thus, we think it is not helpful to plot the SUVA because if absorbance measurement is contaminated SUVA will be as well.

Page 10 Line 1: *"Subheading suggestion: Nature of the DOM concentration and composition relationship across the terrestrial Arctic OR just add the word concentration into the title"*
Subheadings were changed as previously mentioned.

Page 10 Line 2: *"Delete "as found" in this sentence. "*
Done

Page 10 Line 7-9: *"Revise for stronger wording: "However, this relationship is not always strong for ecosystems where DOM is strongly altered…" Here's a question: Can't a photodegradation argument be made for your sites during the summer? "*
This paragraph was changed according to your suggestions:
*"However, compared to other reported studies with DOC and cDOM in the same range, cDOM350 is slightly depleted. This can be a result of stronger photodegradation compared to other sites. Furthermore, it is reported that the DOC to cDOM relationship can strongly vary throughout the season and regions (Mannino et al., 2008; Vantrepotte et al., 2015)."*

Page 10 Line 10: *"Is this insinuating that the DOM you are tracking may be directly a result of leaching or disturbance from 0-30cm or 100cm depths? Are these permafrost links?"*
These are indeed links with SOCC and vegetation coverage. We added this into the paragraph.

Page 10 Line 26: *"Is fresh (less altered) referring to less microbially and photochemically altered? So freshly produced? Higher aromaticity = fresh material? And prone to degradation? Higher aromatic freshly produced material – is coming from what? And prone to what kind of degradation? This is an interesting point and should be clarified. "*
Changed to:
*"This confirms the hypothesis proposed by Vonk et al. (2015b), that DOM exported from smaller rivers has a higher aromaticity, which suggests that the material is fresh (less altered by different degradation processes)."*

Furthermore, as described throughout the whole manuscript we refer mostly to photodegradation since it is indicated by S275-295. The reason why S275-295 is changing with increasing photodegradation is that no photons are naturally available at 275nm. However, at 295 naturally available light spectrum starts and photons are available which can "bleach" DOM and 295nm. Keeping this is mind aCDOM275 will be stable but aCDOM295 will change which ultimately changes the aCDOM slope of the range between 275 and 295nm

There are no reliable optical characteristics which can point towards microbial degradation. Indications about "fresh" plant litter with high SUVA and thus higher aromaticity weight and higher molecular weight with low S275-295 were reported in Helms et al., 2008; Neff et al., 2006; Spencer et al., 2009; Striegl et al., 2005; Weishaar et al., 2003, and Stedmon et al., 2011.

Page 10 Line 28-29: *"A great point to make about sampling smaller catchments and describing their impact in a changing Arctic climate. This point should be made up front and supported throughout. "*
We moved this paragraph to the beginning of the section.

Page 11 Line 1 and section: *"This section seems to be more important up front before the current 5.1 and 5.2 sections. Consider reorganizing the order of these discussion points. "*
As described earlier in this document, we changed the structure of the discussion section according to the suggestions. This section is now the first part in the discussion.

Page 11 Line 11: *"Fresh DOM is high SUVA? An explanation should be discussed here. Is this freshly produced? Or freshly released? And the next sentence describes fresh as low autochthonous production. Fresh as in – newly introduced? "*
We changed this paragraph to:
*"This includes degraded (higher S275-295), which was prone to mobilized or remobilized from permafrost since longer time already, and older (lower SUVA) organic matter which was freshly mobilized from lower soil horizons in the permafrost."*

Page 11 Line 16-20: *"Check with figure. Both symbols are circled in these groups and a clear relationship is not apparent. "*
See explanation in results

Page 11 Line 22: *"Please add a comma after "intensity" "*
done

Page 11 Line 27: *"What kind of degradation? "*
This is valid for microbial and photodegradational processes.
We clarified sentence:
*"...degradation processes both, microbial and photodegradation"*

Page 11 Line 28: *"Deeper in the active layer? Of what? Soil? Permafrost? These are important points to continue to tie together throughout the manuscript. And it suggests that rainfall mobilizes different types of C. "*
"Active layer" is a term which is commonly used for permafrost soils, thus, yes – this refers to permafrost.
*"Of what? Soil? Permafrost? These are important points to continue to tie together throughout the manuscript."*
In this context "deeper in the active layer" can only mean deeper into the top layer of soil that thaws during the summer and freezes again during the autumn. This is a common terminology in permafrost research.
The type of soil is described in the section study area.
We further modified the sentence:
*"...the active layer containing potentially older carbon (decreasing SUVA and increasing S275-295 at the outflow and throughout the profile) than surface soils with mostly recently fixed carbon from modern plants."*

Page 11 Line 30: *"Please add the word concentration after DOC"*
done

Page 11 Line 31-32: *"Indicative of more decomposed material – meaning the mobilization of more decomposed material from???"*
This is discussed in Coch et al. (in review) as indicated in the following sentence.

Page 12 Line 4: *"What isotope? "*
See comment above. We added the reference:
*"...the isotopic signature of rain (Coch et al. (in review)."*

Page 12 Line 6-7: *"Delete "after the first one" and add "later". Also, what's important about this timing to the DOM story? "*
*Done. The timing between rainfall events control soil moisture and flow pathways. If the soil is saturated when a rainfall event occurs (i.e. short duration between rainfall events), we see a mobilisation of surface OM.*

Page 12 Line 9-10: *"Was that trend reflected in your data? "*
No, this is a discussion based on reported literature which we think might explain the characteristics of our data.

Page 12 Line 16: *"Please add the word concentration after DOC and delete the word "there". "*
done
Page 12 Line 18: *"The definition of fresh seems to be changing. Consider a usage of it to indicate mobilization. "*
We changed the text accordingly

Page 13 Line 1: *"Provide a definition of labile here to improve clarity."*
We added an explanation to the sentence:
*"These studies found that permafrost-derived DOM is more labile and thus easily used by bacteria compared to surface (organic mat) DOM"*

Page 13 Line1-2: *"You showed that stormflow alters flow pathways? This seems like an overstatement. "*
We changed the sentence to*: "...we show that different flow pathways are activated during stormflow conditions at the Low and High Arctic locations, which influence the quality of DOM exported."* In fact, flow pathways are different during baseflow and rainfall conditions, see also Coch et al. (2018).

Page 13 Line 5: *"What kind of degradation? "*
We changed this paragraph to:
*"Based on the optical properties, this material shows low molecular weight and aromaticity, i.e. it is already altered. In contrast, rainfall events of high magnitude and intensity that act on saturated soil lead to shorter residence time in the flow path and thus export more fresh (less altered due to different degradation processes) near-surface-derived DOM (higher SUVA and lower S275-295)."*

Section 5.3.2: *"Much improved. Again, the whole 5.3 section should be 5.1 and reordered. "*
Done as described earlier!

Page 13 Line 14: *"What kind of degradability? "*
Changed to:
*"Vonk et al. (2015b) showed that the microbial and photo-degradability decreased from small streams towards larger rivers within the continuous permafrost zone."*

Page 13 Line 18-19: *"The objective was therefore to… doesn't make any sense. Can you elaborate here? Our objective was therefore to investigate the upstream to downstream patterns in smaller coastal catchments to understand…? Tie in the information from the previous sentences."*
*Changed to:*
*"Our sampling strategy along the rivers in combination with detailed mapping of the catchments with a focus on permafrost disturbances, provide insights into upstream to downstream patterns in small coastal catchments in the Low and High Arctic."*

Page 13 Line 20: *"Please add a comma after SUVA "*
done

Page 13 Line 21: *"Please add a comma after 295"*

changed

Page 13 Line 20-21: *"So what does that mean? What kinds of C are coming in, transforming, whatever? "*
The interpretation of this signals is given at the end of the paragraph which was adapted following:

*"The locations at 2000 m and 1300 m distance from the outflow show distinct high values of DOC and acDOM350 compared to the other locations downstream of them. These high concentrations are a result of degrading ice-wedge polygons, which heavily influence DOM in the headwaters of the stream (Coch et al. in review). The location at 1300 m marks the inflow of another headwater tributary impacted by degrading ice-wedge polygons. Thus, main expected sources for fresh mobilized DOM, from deeper permafrost soil horizons, are headwaters and water from the tributary. This is supported by high SUVA and low S275-295 indicating high molecular weight."*

Page 13 Line 23: *"This information of the tributary is really important. Add it to the figure. "*

We added this to the figure.

Page 13 Line 33: *"This sentence looks cut off. What's the point to be made with these ideas? "*

We deleted this part.

Page 14 Line 2: *"Add the word concentration after DOC "*

Done

Page 14 Line 5: *"Use the past tense here. "*

Done

Page 14 Line 15: *"Consider ending this section with comparative low and high arctic themes and impact. "*

An additional paragraph was added:

*"Overall, DOM characteristics in both study areas are affected by local permafrost disturbances. In sampling transects which are not affected by permafrost disturbances, gradual degradation was observed."*

Page 14 Line 18: *"A note on the strength of the message. Permafrost disturbance is continuously happening but exacerbated with rainfall events? Are they connected or not? That message might be good up front and then supported here; it is lost throughout the text. "*

We inserted the sentence: "Degrading ice-wedge polygons and retrogressive thaw slumps impact DOM quantity and quality in the catchments.". We did not directly study the connection between disturbance and rainfall events, thus, there is no clear message here.

Page 14 Line 22: *"And C quality, right? "*
*C quality differences are described later in the Conclusion.*

Page 14 Line 26: *"Please add a comma after 295 "*
Done

Page 14 Line27: *"How can your measurements describe C prone to degradation? "*
changed

Page 14 Line 28: *"Streams or rivers? And why is this useful tool for assessing downstream patters important to your study? Drive the message home? "*
We think the "why" is explained in the following sentence

Page 15 Line 12: *"Please add a comma after H.L. "*
done

Page 15 Line 13 and 14: *"Typo. Analyzes should be analyses. "*
changed

Page 15 Line 14: *"Please delete the comma after analyses and add the word and. Also, correct the word visualized to interpreted. "*
done

Page 15 Line 15: *"Please add a comma after S.L. "*
Done

Page 15 and 16: *"The authors fluctuate with usage of lab and laboratory. Please correct the usage of lab to laboratory where appropriate. "*
We changed this accordingly.

Figure 1: *"Greatly improved. Are the subzones discussed in the text? Is it important in this figure? Please add a comma after West River in Line 6 Figure 1 caption. "*
Yes, they are discussed in section 2.

Figure 2: *"What would CB's data look like if the eliminated samples were added to this figure? A few times in this manuscript the words absorbance and absorption are interchanged. Consider using only one version of this word. "*
We checked the use of absorbance and absorption throughout the manuscript and each use we think is correct.
The Absorbance is unitless [0-1]. The absorption is the Absorbance normalized with the length of the cuvette used (the optical path length L), absorption = absorbance / L with the most common units [m-1] or [cm-1] and used as Decadal absorption and Naperian absorption (Naperian absorption = Decadal absorption * 2.303).In methods we introduce the use of both words.

Figure 3: *"In (b) it is difficult to see the data behind the blue dots of ICW downstream. The gray and orange data points are covered. Can they be overlaid for easier visualization? Define what the two groups are in (c) in the caption. As mentioned earlier, these two groups encompass both circles and triangles so what is special about these groups if not water type? "*
We added transparency to the points and decreased the size.
We added explanation for the "outlier" earlier in this document as well as in the manuscript.
Figure 4: *"It might be worth noting that SUVA is calculated at A254nm.*

This is described in the method section of the manuscript

*Please add a comma after the word downstream in Line 4.*

comma added.

*Use consistent language for water type – circles and triangles. In this caption, please edit dot to circle.*

*We edited this accordingly.*

*In (b) it is difficult to see the data behind ICW downstream, similar to Figure 3. "*

We added transparency to the points and decreased the size.

Figure 5: *"Might it be helpful to mark tributaries on the Herschel Island figure? Didn't that feature change some of the measurements? "*

We marked the tributary in the figure.

Figure 6: *"Annotate which direction the data goes for the gaps in S in (e) or describe the 6x increase or whatever amount of increase or decrease in the caption. "*

The outliers are now presented in Fig. 2.

Figure 7: *"Delete the word "to" in Line 4 before the word "high".*

Changed to:

*"The samples circled in black show disproportionately high absorption values in relation to the DOC concentration"*

*This part of the data still gives pause, since other values are around 40, but with higher DOC concentrations. Can these really be excluded? Consider adding SUVA as a second panel to this figure. "*

Yes, this data needs to be excluded as described carefully in the methods and in a whole section of the discussions. This extreme DOC to cDOM350 relationship that is encircled in black is unrealistic and cannot exist for natural surface waters. There is no mechanism that can explain high cDOM and at the same time very low DOC concentrations. These cuvette measurements had low transmission values due to additional scattering that is not assumed to occur in the cDOM protocol because samples are assumed to be filtrates and all reduction in transmission is per CDOM protocol measured absorbance. There are also more excluded samples showing some precipitates that are not outliers in terms of the DOC to cDOM350 relationship (yellow circles and triangles) and seem to be correct. We did not use these samples for analyses in this study as we had detected the group with the problematic DOC to cDOM350 relationship and wanted to be careful

Table 1: *"Soil organic carbon content in the table can be abbreviated to SOCC.*

Done

*This table is a nice tie in with Figure 1, with annotations regarding the subzones. Can more of that be incorporated into the text so that the reader is reminded that the different rivers correspond to different vegetation types? "*

This description can be found in the Study Site section and along the discussion.

Table 2: *"Add a comma after pH in Line 3 and standing waters in Line 4.*

Done

*Are the significant differences highlighted in the table discussed in the main text?"*

Yes, described in the result section and discussed

[revised manuscript text omitted]

Moved up [17]: Herschel Island (Low Arctic) shows on average
Moved up [34]: setting by Balcarczyk et al. (2009). The
Moved up [32]: Fouché et al. (2017) conducted an extensive
Moved up [1]: , a higher relative contribution of autochthonous
Moved up [26]: At Herschel Island, we captured the res… [23]
Moved up [29]: A change in water sources for these two rainfall
Moved up [33]: Fouché et al. (2017) explain this pattern by a
Moved up [28]: The contrasting response of Ice Creek West to
Moved up [30]: Thus, the DOM was first sourced from the
Moved up [31]: In addition to the antecedent conditions, the
Moved up [35]: At the Low Arctic setting our data suggests that
Moved up [36]: towards a nival-pluvial flow regime leading to
Moved up [18]: . Studies show the importance of headwater
Moved up [19]: degrading ice-wedge polygons, which heavily
Moved up [21]: downstream (3 August 2017), which is also
Moved up [22]: West River is characterized by a downstream
Moved up [23]: Abbott et al. (2014) found that DOM is most
Moved up [24]: High S275-295 and SR were observed in
Moved up [25]: . The authors attribute low SUVA and high

[revised manuscript text omitted]

As described in the methods section, some samples formed precipitates inside the bottles in the form of small thin flakes, which partly remained in suspension or accumulated at the bottom. All samples were filtered in the field through 0.7 µm glass fiber filters, and the precipitation occurred after filtration during storage.

| Page 15: [40] Moved from page 9 (Move #9) | Caroline Coch | 01/09/2019 18:00:00 |
| --- | --- | --- |

As described in the methods section, some samples formed precipitates inside the bottles in the form of small thin flakes, which partly remained in suspension or accumulated at the bottom. All samples were filtered in the field through 0.7 µm glass fiber filters, and the precipitation occurred after filtration during storage.

| Page 15: [41] Moved from page 9 (Move #11) | Caroline Coch | 01/09/2019 18:00:00 |
| --- | --- | --- |

As described in the methods section (3.1), samples showing precipitates in the laboratory were excluded from the study, even if the absorption values were plausible when compared to the corresponding DOC concentration (Fig 7). At Cape Bounty, this was the case for 25 out of 55 samples.

| Page 15: [41] Moved from page 9 (Move #11) | Caroline Coch | 01/09/2019 18:00:00 |
| --- | --- | --- |

As described in the methods section (3.1), samples showing precipitates in the laboratory were excluded from the study, even if the absorption values were plausible when compared to the corresponding DOC concentration (Fig 7). At Cape Bounty, this was the case for 25 out of 55 samples.

**Page 15: [42] Moved from page 9 (Move #13)**      **Caroline Coch**                    **01/09/2019 18:00:00**

. Dissolved iron in terrestrially dominated waters is dominantly complexed with humic and fulvic acids.